# ECG-QA: A Comprehensive Question Answering Dataset Combined With Electrocardiogram

**Jungwoo Oh[1], Gyubok Lee[1], Seongsu Bae[1], Joon-myoung Kwon[2], Edward Choi[1]**
KAIST, Daejeon[1] Medical AI Inc., Seoul[2]
`{ojw0123, gyubok.lee, seongsu, edwardchoi}@kaist.ac.kr`[1]
`cto@medicalai.com`[2]

## Abstract

Question answering (QA) in the field of healthcare has received much attention due to significant advancements in natural language processing. However, existing healthcare QA datasets primarily focus on medical images, clinical notes, or structured electronic health record tables. This leaves the vast potential of combining electrocardiogram (ECG) data with these systems largely untapped. To address this gap, we present ECG-QA, the first QA dataset specifically designed for ECG analysis. The dataset comprises a total of 70 question templates that cover a wide range of clinically relevant ECG topics, each validated by an ECG expert to ensure their clinical utility. As a result, our dataset includes diverse ECG interpretation questions, including those that require a comparative analysis of two different ECGs. In addition, we have conducted numerous experiments to provide valuable insights for future research directions. We believe that ECG-QA will serve as a valuable resource for the development of intelligent QA systems capable of assisting clinicians in ECG interpretations.

## 1 Introduction

In recent years, significant advancements in natural language processing have revolutionized the field of question answering (QA) in a wide range of domains. Previous works have demonstrated the great potential of QA systems in various domains, where they have been combined with different modalities such as images [1, 32, 15, 8] or tables with images [24, 13]. Concurrently, QA systems have also been explored in the healthcare domain, including visual QA with chest X-ray [11, 14], clinical-note-based QA [20], and QA over structured electronic health record (EHR) data [28, 12]. These pioneering efforts have successfully bridged the gap between general-domain QA and the medical field, unlocking new possibilities to improve healthcare outcomes and enhance medical decision-making processes.

Despite this remarkable progress, there is a noticeable absence of datasets that combine electrocardiogram (ECG) data with question answering. As a fundamental diagnostic tool in cardiology, ECG provides critical insights into the electrical activity of the heart and plays an important role in detecting various cardiac conditions [3, 22, 33]. Consequently, integrating ECG data with QA systems holds tremendous potential to improve the interpretation of cardiac data, leading to more accurate diagnoses and personalized treatment plans.

To this end, we present ECG-QA[1], a novel QA dataset that incorporates ECG data for question answering tasks. To the best of our knowledge, ECG-QA is the first dataset that combines QA and ECG, opening up new avenues for integrating multi-modal machine learning with cardiac healthcare.

The main contributions of this work are threefold:

---

[1]The dataset is available at `https://github.com/Jwoo5/ecg-qa`, licensed under CC-BY-4.0 license.

37th Conference on Neural Information Processing Systems (NeurIPS 2023) Track on Datasets and Benchmarks.

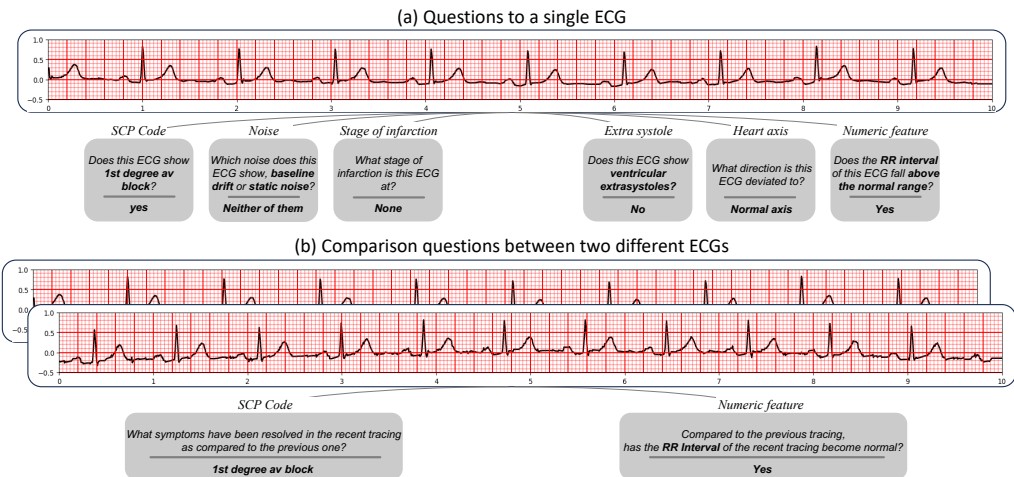

Figure 1: Sample question-answer pairs in ECG-QA. (a) Questions to a single ECG with various types of attributes. (b) Comparison questions between two different ECGs. Refer to Section 3.1.1 for more details about each attribute type.

- We propose the ECG-QA dataset, a diverse collection of questions focused on ECG interpretation and analysis (See Figure 1). This dataset introduces the novel concept of incorporating question answering into the realm of ECG analysis, making it a valuable resource for developing and evaluating QA systems in the context of cardiology.

- To cover more complex yet clinically critical questions, we include questions that require comparative analysis of two ECGs (See Figure 1 (b)). This inclusion brings a new degree of complexity, as addressing these comparison questions extends beyond the conventional scope of ECG analysis using machine learning. By incorporating these types of questions, we not only address the real-world needs of medical professionals but also broaden the potential applications of machine learning in ECG analysis.

- We provide a benchmark for QA models, including recent large language models (LLMs), on the ECG-QA dataset, promoting further research and encouraging the development of novel methods to leverage ECG signals for question answering tasks. We believe that ECG-QA will serve as a valuable resource in advancing machine learning applications in cardiology and improving medical decision-making processes.

## 2 Related works

**Medical QA datasets**   QA systems have been extensively explored in the healthcare domain, catering to the specific needs and challenges of medical data. However, most existing medical QA datasets are primarily based on clinical texts, medical images, or structured EHR tables. For example, Pampari et al. [20] proposed the emrQA dataset, consisting of question-answer pairs derived from unstructured clinical notes. In addition, Kovaleva et al. [11] and Liu et al. [14] proposed datasets for visual QA using X-ray images, aiming to investigate intelligent interactive systems for radiology. Meanwhile, in the field of QA over structured EHR data, Wang et al. [28] and Lee et al. [12] have developed datasets called MIMICSQL and EHRSQL respectively, which consist of questions and their corresponding SQL queries. While these healthcare QA systems demonstrate the potential of leveraging medical data with QA to improve healthcare outcomes, there is currently no dedicated QA dataset specifically designed for ECG data despite its widespread use in diagnosing cardiovascular conditions and monitoring patients' heart health.

**Electrocardiogram**   Previous studies in the field of ECG have predominantly focused on using ECG data for diagnostic purposes such as identifying various heart diseases. For instance, Nejedly et al. [18] proposed an ensemble of residual networks with attention modules to classify cardiac diseases using ECGs, which won first place in the *PhysioNet/Computing in Cardiology Challenge 2021* [21]. At the same challenge, Han et al. [6] achieved second place by utilizing SE-WRN [31], which is a

Table 1: Sample template questions for different question & attribute types in ECG-QA.

| Question type | Attribute type | Example template question |
|---|---|---|
| Single-Verify | SCP Code | Does this ECG show symptoms of **non-specific ST changes**? |
| | Noise | Does this ECG show **baseline drift** in **lead I**? |
| | Stage of infarction | Does this ECG show **early stage of myocaridal infarction**? |
| | Extra systole | Does this ECG show **ventricular extrasystoles**? |
| | Heart axis | Does this ECG show **left axis deviation**? |
| | Numeric feature | Does the **RR interval** of this ECG fall **within the normal range**? |
| Single-Choose | SCP Code | Which symptom does this ECG show, **conduction disturbance** or **hypertrophy**? |
| | Noise | Which noise does this ECG show, **baseline drift** or **static noise**? |
| | Stage of infarction | Which stage of infarction is this ECG at, **early stage of myocardial infarction** or **late stage of myocardial infarction**? |
| | Extra systole | Which kind of extra systoles does this ECG show, **ventricular extrasystoles** or **supraventricular extrasystoles**? |
| | Heart axis | Which cardiac axis does this ECG show, **left axis deviation** or **right axis deviation**? |
| | Numeric feature | Which range does the **RR interval** of this ECG fall in, **below the normal range** or **within the normal range**? |
| Single-Query | SCP Code | What form-related symptoms does this ECG show? |
| | Noise | What kind of noises does this ECG show in **lead I**? |
| | Stage of infarction | What stage of infarction is this ECG at? |
| | Extra systole | What kind of extra systoles does this ECG show? |
| | Heart axis | What direction is this ECG deviated to? |
| | Numeric feature | What range does the **RR interval** of this ECG fall in? |
| Comparison-Consecutive-Verify | SCP Code | Compared to the previous tracing, has **left ventricular hypertrophy** been resolved in the recent tracing? |
| | Numeric feature | Compared to the previous tracing, has the **PR interval** of the recent tracing become normal? |
| Comparison-Consecutive-Query | SCP Code | What symptoms have been resolved in the recent tracing as compared to the previous one? |
| | Numeric feature | What numeric features of the recent tracing now have become normal compared to the previous one? |
| Comparison-Irrelevant-Verify | SCP Code | Compared to the first ECG, has **atrial fibrillation** been newly detected in the second ECG? |
| | Numeric feature | Compared to the first ECG, has the **P duration** of the second ECG changed to an abnormal value? |
| Comparison-Irrelevant-Query | SCP Code | What symptoms still remain in the second ECG as compared to the first ECG? |
| | Numeric feature | What numeric features of the second ECG are now considered abnormal values as compared to the first ECG? |

combination of wide residual network [30] and squeeze-and-excitation modules [7]. Furthermore, several works [4, 10, 19] studied self-supervised learning with ECGs to improve performances on the cardiac arrhythmia classification task. These works concentrate on classifying diagnoses based on a single ECG, and do not consider the significance of comparing two ECGs, despite its importance in clinical contexts. For example, by detecting resolved symptoms after some treatments, the medical practitioners can assess the effectiveness of the treatments and evaluate the progress of the patient's condition. To reflect this clinical reality, we have included questions that involve the comparison of two ECGs within the ECG-QA dataset, making our dataset unique and valuable.

## 3 Dataset Construction

We constructed the ECG-QA dataset upon the PTB-XL dataset [26][2], which offers comprehensive metadata regarding ECGs annotated by expert cardiologists. This metadata covers a wide range of information including ECG reports, diagnostic statements, diagnosis likelihoods, and signal-specific properties. To ensure the high quality of our dataset, we performed additional filtering on the original

---

[2]This dataset is licensed under CC-BY-4.0 license.

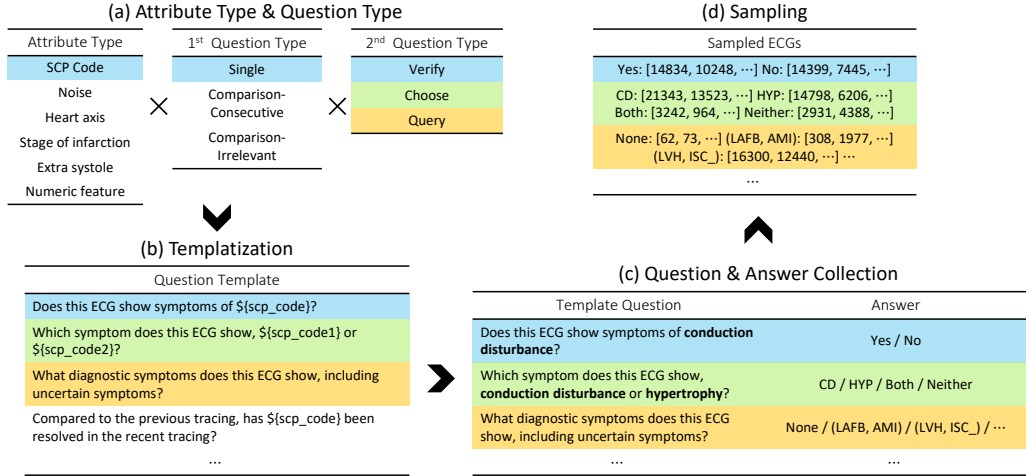

Figure 2: Visualization of the ECG-QA sample generation pipeline. The numbers in the sampling stage (d) stand for the ECG IDs in the PTB-XL dataset. In the sampling process, we also convert the template questions into pre-defined paraphrases for each sample.

PTB-XL dataset. Specifically, we selected ECGs that were marked with a `validated_by_human` tag set to True, which indicates the validation by a human cardiologist, and excluded ECGs that had empty reports. As a result, the ECG-QA dataset was constructed using $16,054$ samples of 10-second ECGs from the PTB-XL dataset. In addition, we split the samples into training and test sets according to a 8:2 ratio based on their patient IDs before generating QA samples to prevent the overlapping of ECGs between training and test sets. We again split the training samples into training and validation sets by a 9:1 ratio, yielding 7.2:0.8:2.0 training-validation-test distribution.

## 3.1 Question template

To generate QA samples, we start by creating the question templates to collect questions, answers, and their corresponding ECGs. Because the questions are fully derived from these templates, it is important to define templates that are not only diverse but also clinically meaningful. To achieve this goal, we extracted the relevant attributes from the PTB-XL metadata to determine the content of the questions (*i.e., attribute types*) and categorized the questions into several question types. Then, we combined these types to construct the template questions, and additionally generated template paraphrases to add lexical diversity to our dataset. As a result, we defined a total of 70 templates as shown in Supplementary A.2, and we also provide the example questions derived from these templates in Table 1. All the processes of designing question templates have been validated by a board-certified medical expert from the Department of Critical Care and Emergency Medicine in terms of clinical utility. The detailed processes of each step are described in the following subsections, as well as visualized in Figure 2 (a) and (b).

### 3.1.1 Attribute type extraction

**SCP code** The PTB-XL dataset provides SCP codes for each ECG sample, consisting of 71 different ECG symptoms that adhere to the SCP-ECG v0.4 that preceded the current SCP-ECG standard [23]. These attributes are composed of form-related (*e.g., inverted T-waves*), rhythm-related (*e.g., sinus arrhythmia*), and diagnostic symptoms (*e.g., non-specific ischemic*) along with additional 5 superclasses for diagnostic labels. Given that detecting cardiac symptoms is a primary objective in many ECG studies [18, 6, 4, 10, 19], we included questions that inquire about various ECG symptoms in the ECG-QA dataset. To ensure the dataset quality, we excluded attributes with a low number of positive ECG samples in the test split, such as `WPW` (wolf-parkinson-white syndrome), resulting in a final selection of 64 attributes, including the 5 superclasses. Furthermore, we developed a regular expression parser to extract the grounded lead position of form-related symptoms from the ECG reports (See Supplementary B.1). This enables us to include questions in the ECG-QA dataset that

specifically address the leads in which symptoms are detected (*e.g., Does this ECG show symptoms of **inverted T-waves** in **lead I**?*), making the ECG-QA dataset more comprehensive.

**Noise**    Considering that ECG measurements involve placing electrodes on specific body surfaces, it is inevitable to encounter various signal interferences such as baseline drift, which can be caused by patient movement or machine issues. Therefore, it becomes crucial to differentiate these interferences from the original ECG signals during analysis. To reflect this aspect in the ECG-QA dataset, we leveraged the signal noise information available in the PTB-XL metadata. This information is provided as a string indicating the specific lead positions where each noise is detected (*e.g.,* "v1-v6" or "i-iii"). We parsed these strings to identify the exact lead positions associated with four different types of noises: *Baseline drift*, *Static noise*, *Burst noise*, and *Electrode problems*.

**Stage of infarction**    Since identifying the stage of myocardial infarction (MI) helps healthcare professionals determine the most appropriate management strategies by assessing the risk profile for the patient, we also have considered the stage of infarction as an important attribute in the ECG-QA dataset. In the PTB-XL dataset, it distinguishes the stage of MI into six levels including intermediate stages: "I," "I-II," "II," "II-III," "III," and "Unknown." In addition, because there are two fields indicating the stage of infarction (`infarction_stadium1` and `infarction_stadium2`), the statements could be potentially multiple. For the sake of simplicity, we simplified the stages into 4 levels by regarding the intermediate stages ("I-II" and "II-III") as their "lower" stage ("I" and "II"). Then, we used the second statement if there are multiple entries at a time. After defining an additional stage called "None" for those who do not have MI, we could derive five attributes for the stage of infarction.

**Extra systole**    Since extra systoles can be a sign of underlying cardiac conditions or abnormalities, it is also important to detect them to evaluate the patient's heart health. To address the presence of extra systoles in the ECG-QA dataset, we utilized the relevant annotations provided in the PTB-XL metadata, which includes information about the occurrence of different types of extra systoles: *Extrasystoles*, *Ventricular extrasystoles*, and *Supraventricular extrasystoles*.

**Heart axis**    The heart axis provides valuable information about the direction of the heart's electrical activity during each cardiac cycle. In the ECG-QA dataset, we have considered the heart axis as another crucial attribute since it is an important parameter that can help in diagnosing certain cardiac conditions. Although the PTB-XL metadata includes heart axis information, we did not utilize it because it does not specify the actual numerical values of the heart axis. Instead, we manually calculated the heart axis degrees by employing an external tool, NeuroKit2 [16] [3]. Then, we classified them into four categories following the conventional standards: *Normal heart axis*, *Left axis deviation*, *Right axis deviation*, and *Extreme axis deviation*.

**Numeric feature**    The ECG-QA dataset also incorporates numeric features that provide further insights into the cardiac signals. Similar to the heart axis, we used NeuroKit2 to calculate the numeric values for these features since the PTB-XL dataset does not explicitly provide such information. Specifically, we extracted the locations of P, Q, R, S, and T waves for each beat present in lead II, and computed six different numeric features: *RR interval*, *P duration*, *PR interval*, *QRS duration*, *QT interval*, and *QT corrected*. Given that a 10-second ECG recording typically contains multiple beats and thus multiple numeric values for each feature, we represented each feature using its median value. This approach helps to minimize the impact of abnormal contractions, such as ventricular premature contractions, on the calculated values. Additionally, we categorized each numeric value as below, within, or above the normal range, where the normal range criteria are described in Supplementary B.2, derived from a previous study [27]. These ranges serve as a useful reference for assessing the numerical measurements and identifying potential abnormalities in the cardiac signals.

### 3.1.2   Question type definition

We have defined two different types of questions. The first type pertains to the type of ECGs associated with a question, which can be categorized as follows: 1) *Single*, which refers to questions involving a single ECG; 2) *Comparison-Consecutive*, which involves comparison questions between two

---

[3]This software is licensed under MIT license.

Table 2: Test performances for different question types. We also provide 95% confidence interval across 3 random seeds. The best performances for each question type are highlighted with **boldface**.

| Question Type | per Q-type majority | M³AE[2] | | MedViLL[17] | | Fusion Transf. | | Blind Transf. | | Deaf Transf. | |
|---|---|---|---|---|---|---|---|---|---|---|---|
| | | EM Acc. | AUROC | EM Acc. | AUROC | EM Acc. | AUROC | EM Acc. | AUROC | EM Acc. | AUROC |
| S-Verify | 67.7 | $\mathbf{74.6_{\pm 0.4}}$ | $0.761_{\pm 0.002}$ | $73.9_{\pm 0.5}$ | $\mathbf{0.768_{\pm 0.011}}$ | $72.1_{\pm 0.5}$ | $0.725_{\pm 0.008}$ | $67.7_{\pm 0.0}$ | $0.629_{\pm 0.008}$ | $67.3_{\pm 0.2}$ | $0.613_{\pm 0.003}$ |
| S-Choose | 31.2 | $\mathbf{57.1_{\pm 0.8}}$ | $\mathbf{0.850_{\pm 0.002}}$ | $54.1_{\pm 0.8}$ | $0.839_{\pm 0.001}$ | $46.4_{\pm 0.4}$ | $0.797_{\pm 0.007}$ | $31.0_{\pm 0.1}$ | $0.529_{\pm 0.006}$ | $31.4_{\pm 0.0}$ | $0.786_{\pm 0.011}$ |
| S-Query | 23.2 | $\mathbf{41.0_{\pm 0.5}}$ | $\mathbf{0.836_{\pm 0.002}}$ | $40.4_{\pm 0.6}$ | $0.831_{\pm 0.004}$ | $37.4_{\pm 0.6}$ | $0.791_{\pm 0.011}$ | $24.0_{\pm 0.0}$ | $0.549_{\pm 0.006}$ | $27.0_{\pm 0.1}$ | $0.754_{\pm 0.006}$ |
| CC-Verify | 62.8 | $\mathbf{75.5_{\pm 0.2}}$ | $\mathbf{0.792_{\pm 0.002}}$ | $74.3_{\pm 2.6}$ | $0.778_{\pm 0.047}$ | $71.9_{\pm 0.6}$ | $0.760_{\pm 0.003}$ | $65.7_{\pm 0.8}$ | $0.610_{\pm 0.001}$ | $59.5_{\pm 0.6}$ | $0.510_{\pm 0.009}$ |
| CC-Query | 16.9 | $20.1_{\pm 1.6}$ | $0.808_{\pm 0.003}$ | $\mathbf{22.0_{\pm 1.3}}$ | $\mathbf{0.816_{\pm 0.003}}$ | $18.4_{\pm 1.3}$ | $0.781_{\pm 0.003}$ | $16.9_{\pm 0.0}$ | $0.568_{\pm 0.023}$ | $16.9_{\pm 0.1}$ | $0.693_{\pm 0.012}$ |
| CI-Verify | 66.1 | $75.3_{\pm 0.9}$ | $0.769_{\pm 0.010}$ | $\mathbf{77.5_{\pm 1.6}}$ | $\mathbf{0.823_{\pm 0.021}}$ | $68.1_{\pm 0.6}$ | $0.723_{\pm 0.010}$ | $66.2_{\pm 0.1}$ | $0.508_{\pm 0.004}$ | $61.1_{\pm 0.5}$ | $0.505_{\pm 0.004}$ |
| CI-Query | 1.10 | $\mathbf{4.19_{\pm 0.2}}$ | $0.741_{\pm 0.008}$ | $3.50_{\pm 0.2}$ | $\mathbf{0.758_{\pm 0.004}}$ | $2.19_{\pm 0.1}$ | $0.704_{\pm 0.004}$ | $0.95_{\pm 0.1}$ | $0.527_{\pm 0.008}$ | $1.11_{\pm 0.0}$ | $0.632_{\pm 0.016}$ |

S: Single, CC: Comparison-Consecutive, CI: Comparison-Irrelevant

consecutive ECGs from the same patient; and 3) *Comparison-Irrelevant*, which involves comparison questions between two irrelevant ECGs from different patients. Although *Comparison-Irrelevant* questions may not seem realistic in a clinical setting, we included these questions since they can help to reinforce a machine's comprehension ability and be utilized for model evaluation when comparing two different ECGs. In addition, inspired by GQA [8], the second type of question refers to the main function it should perform. These can be categorized as follows: 1) *Verify*, which corresponds to yes/no questions; 2) *Choose*, which applies to questions where the selection is made from two given options; and 3) *Query*, which are open-ended questions that seek to retrieve specific attributes.

By combining these two types, we could derive a total of 9 possible question types. However, we did not include the combinations of *Comparison* and *Choose* types since it seemed unnatural to select from two given options when comparing different ECGs. Similarly, with regards to the attribute types, we only considered **SCP code** and **Numeric feature** for comparison questions because these two attribute types are providing the most informative features when comparing two ECGs.

### 3.1.3  Paraphrase generation

To enhance the lexical diversity of the ECG-QA dataset, we manually curated paraphrases for each question template based on the machine-generated candidates by utilizing OpenAI's ChatGPT. We ensured that the questions in the test split were not included in the training set to evaluate the generalizability of the QA models on different lexical variations. The detailed procedure for generating paraphrases and its results are presented in Supplementary A.3.

### 3.2  QA sample collection

As shown in Figure 2 (b) and (c), we collected questions by plugging the corresponding attributes into the placeholder that existed in the question templates. For example, a question template *"Does this ECG show symptoms of* `${scp_code}`*?"* can be transformed into *"Does this ECG show symptoms of **conduction disturbance**?"* We further gathered the corresponding answers for each question and paired them to create (question, answer) pairs. Then, for each (question, answer) pair, we again randomly sampled the corresponding ECGs from the candidate ECGs. In each split, the candidate ECGs can be 1) all the single ECGs for *Single* questions; 2) all the ($ECG_1$, $ECG_2$) pairs where $ECG_1$ and $ECG_2$ are the consecutive ECGs from the same patient for *Comparison-Consecutive* questions; and 3) all the ($ECG_1$, $ECG_2$) combinations where $ECG_1$ and $ECG_2$ have different patient IDs for *Comparison-Irrelevant* questions. After we finally replaced the template question with randomly selected paraphrases that matched the corresponding question's template, the process of collecting QA samples was complete. The detailed sampling strategies for different question types are described in Supplementary B.3.

After all these processes, the ECG-QA dataset consists of 267,539 training samples, 64,663 validation samples, and 82,146 test samples, which cover various types of attributes and questions. More detailed statistics of the dataset are described in Supplementary B.4.

## 4  Experiments

**Task formulation**   We formulate QA task as a multi-label classification over all possible answer options that exist in the ECG-QA dataset. The answer labels are composed of 88 attributes from the six attribute sets, 12 lead positions (*i.e.,* lead I - lead V6), and 3 answers for *Verify* questions (yes, no, not sure), leading to a total of 103 answers. Note that we processed "None" answer as an empty label.

**Baselines**  We implemented the following QA baselines: $M^3AE^\dagger$ [2], MedViLL$^\dagger$ [17], Fusion Transformer, Blind Transformer (seeing questions only), and Deaf Transformer (seeing ECGs only). Because the original implementations of $M^3AE$ and MedViLL were intended to pre-train images with texts, we modified them to be applied to ECGs instead of images and pre-trained them using ECG data, as marked with $^\dagger$. Additionally, similar to the per Q-type prior in VQA [1], we include a prior model, per Q-type majority, which outputs only the most frequent answer for each question type in the test split. More details about each model implementation including training hyperparameters are described in Supplementary C.1.1.

## 4.1  Evaluation metrics

**Exact match accuracy**  To calculate the exact match accuracy, we applied a threshold value of 0.5 to each score in the model's output vector, $\hat{\mathbf{y}} \in \mathbb{R}^{103}$, which gives a multi-hot vector of length 103. Then, we compare the output vector with the ground truth answer vector. If the two vectors are exactly the same, we assign a score of 1; otherwise, we assign a score of 0. To obtain the overall accuracy, we sum the scores for all the test questions and divide the aggregated score by the total number of questions, yielding the percentage of questions that were answered exactly.

**AUROC**  While the exact match accuracy is a useful metric, it may not fully capture the model's performance since it does not consider partial credits, especially in the case of *Query* questions that require consideration of much more attributes. To provide a more comprehensive evaluation, we employ the area under ROC curve (AUROC) as another metric. When calculating AUROC, we adopt a cautious approach by only considering the "valid" answer candidates for each question. This approach aims to prevent overestimation, as the model might naturally assign lower scores to "invalid" answer options. For example, in a question like *"Which noise does this ECG show, **baseline drift** or **static noise**?"*, we exclusively consider the scores of the answer options **baseline drift** and **static noise**. We collect scores for each answer option over all the samples and compute macro-averaged AUROC among the answer options.

## 4.2  Upper bound experiments

In the field of clinical medicine, even experienced medical practitioners cannot be entirely certain when making crucial decisions, such as diagnosing a patient's condition. Similarly, the ECG-QA dataset can also suffer from this inherent uncertainty even though we extracted attributes from the existing annotations made by expert cardiologists. As 100% accuracy is unlikely to attain due to the inherent uncertainty, we aim to estimate the upper bound performance a model can achieve with our dataset, and use it as a reference when evaluating the model performance.

Within our dataset, we speculate that questions of the **Single-Verify** type necessitate basic perceptual abilities while other question types can be solved by logically combining these perceptual abilities. For example, **Single-Choose** questions can be answered by verifying the presence of each attribute in the given two options, and similarly, **Single-Query** questions can be solved by verifying the presence of each element within the specified attribute set. Consequently, we hypothesize that achieving high performance on the whole ECG-QA dataset is unlikely without a high level of perceptual ability. Based on this hypothesis, we can estimate the upper bound performances for the whole ECG-QA dataset by measuring the upper bound performances of the **Single-Verify** samples. To this end, we designed the following experiments.

We convert all the *Single* QA samples (**Single-Verify**, **Single-Choose**, **Single-Query**) in the training set into the format that ECG classification models can process, and train the classification models using the converted training samples. Similarly, after converting the **Single-Verify** samples in the test set, we estimate the upper bound performance by measuring performances on the converted **Single-Verify** test samples. Then, we compare this upper bound with **Single-Verify** performances of QA models to show how much the QA baselines can be improved in terms of their perceptual ability. The detailed process of converting QA samples into ECG classification format is described in Supplementary C.1.2.

For these experiments, we employ powerful ECG classification models classifying all the individual attributes present in the **Single-Verify** samples. The models we use include a Transformer-based model pre-trained with the W2V+CMSC+RLM [19] method, Resnet with Attention [18], and SE-

Table 3: Macro-averaged test performances of upper bound models over all attributes for **Single-Verify** questions

| Upper bound Model | Acc. | AUROC |
|---|---|---|
| W2V+CMSC+RLM [19] | $83.0_{\pm0.4}$ | $0.864_{\pm0.003}$ |
| Resnet-Attention [18] | $82.6_{\pm0.3}$ | $0.875_{\pm0.002}$ |
| SE-WRN [6] | $83.1_{\pm0.3}$ | $0.883_{\pm0.002}$ |
| MAX | $85.4_{\pm0.4}$ | $0.907_{\pm0.002}$ |

Table 4: Macro-averaged test performances of QA models over all attributes for **Single-Verify** questions.

| QA Model | Acc. | AUROC |
|---|---|---|
| $M^3AE^\dagger$ [2] | $80.8_{\pm0.3}$ | $0.808_{\pm0.006}$ |
| MedViLL$^\dagger$ [17] | $79.8_{\pm0.3}$ | $0.809_{\pm0.005}$ |
| Fusion Transf. | $76.4_{\pm0.6}$ | $0.764_{\pm0.010}$ |

WRN [6]. In addition, to present the maximized upper bound, we derive another model that takes only the maximum score among the three models for each attribute, which is denoted as MAX. Detailed model implementations and training configurations are presented in Supplementary C.1.2.

## 4.3 Modeling with LLMs

As for one of the future research directions with our dataset, we further investigated the possibility of leveraging LLMs for ECG-QA. Inspired by ChatCAD [29], for each QA sample, we transformed ECGs into text descriptions using the output from the trained upper bound model (SE-WRN) and forwarded them to several OpenAI's GPT models (gpt-4, gpt-3.5-turbo, text-davinci-003)[4] along with the corresponding question. Due to the restricted quota of OpenAI's api usage policy, we randomly sampled 10% from the ECG-QA test set and conducted experiments only once for each LLM model. The detailed processes including prompts that we used are described in Supplementary C.1.3.

## 4.4 Results

**QA results**  The baseline results are presented in Table 2. We also report test performances for different attribute types in Supplementary C.2. As expected, Blind and Deaf Transformer exhibit poor performance while other models all achieve higher scores compared to the prior model (per Q-type majority), indicating that our dataset cannot be solved by solely seeing each question and ECG separately. Furthermore, among the top three models ($M^3AE^\dagger$, MedViLL$^\dagger$, and Fusion Transformer), the pre-trained models ($M^3AE^\dagger$ and MedViLL$^\dagger$) outperform Fusion Transformer, which demonstrates the potential advantages of utilizing novel multi-modal pre-training methods for our dataset. Additionally, the lower performance of *Choose* or *Query* questions compared to *Verify* questions suggests that the primary challenges in our dataset lie on a model's ability to learn logical and set operations based on the basic perceptual abilities that can be acquired from *Verify* questions.

**Upper bound results**  The results of the upper bound experiments are reported in Table 3 and 4. When we compare SE-WRN, which is the best upper bound model, with the best QA model, $M^3AE^\dagger$, we can see that the perceptual ability of baseline models can be improved by 2.3%p and 7.5%p in terms of EM accuracy and AUROC, respectively. Moreover, when comparing with MAX, which is expected to show a higher upper bound, the differences are increased to 4.6%p in EM accuracy and 9.9%p in AUROC. We believe these upper bound results can serve as a useful yardstick for assessing the basic perceptual ability required for more complicated questions such as *Choose* or *Query*.

**LLM modeling results**  The results of the experiments with LLMs are presented in Table 5. Interestingly, the performance of all the GPT models did not surpass that of the QA baseline model. We speculate that this is due to two primary reasons: 1) the upper bound model (*i.e.* SE-WRN) fails to accurately extract necessary information, and 2) some questions were too complicated to be answered with a zero-shot prompt. Since LLMs fully rely on the ECG classification model for interpreting the ECGs, their performance inevitably depends on the capabilities of the ECG classification model. However, we cannot guarantee that SE-WRN is such a perfect model that always outputs accurate interpretations, because it has been trained with a limited set of QA training set to measure only the upper bound of the perceptual ability. Therefore, we expect significant performance gains if we have

---

[4]As all of these experiments were conducted prior to June 7th, 2023, the associated models were referencing the earlier legacy versions. More precisely, at the time of the experiments, gpt-4 corresponded to gpt-4-0314, and gpt-3.5-turbo matched with gpt-3.5-turbo-0301.

Table 5: Test EM accuracies for different question types. Note that we randomly sampled **10%** from the ECG-QA test set for each question type to test the models due to the restricted quota of OpenAI's api usage policy. The best performance for each question type are highlighted with **boldface**.

| Question Type | per Q-type majority | SE-WRN + gpt-4 | SE-WRN + gpt-3.5-turbo | SE-WRN + text-davinci-003 | $M^3AE^\dagger$ [2] |
|---|---|---|---|---|---|
| S-Verify | 69.1 | 71.0 | 69.3 | 75.0 | $\textbf{76.0}_{\pm\textbf{0.7}}$ |
| S-Choose | 30.7 | 48.1 | 36.1 | 37.8 | $\textbf{58.2}_{\pm\textbf{0.9}}$ |
| S-Query | 25.3 | 35.7 | 31.1 | 36.0 | $\textbf{40.0}_{\pm\textbf{1.6}}$ |
| CC-Verify | 58.4 | 54.9 | 58.2 | 56.3 | $\textbf{74.7}_{\pm\textbf{1.2}}$ |
| CC-Query | 17.3 | 13.0 | 10.5 | 15.4 | $\textbf{21.2}_{\pm\textbf{2.0}}$ |
| CI-Verify | 67.0 | 68.8 | 64.1 | 71.5 | $\textbf{75.2}_{\pm\textbf{1.7}}$ |
| CI-Query | 1.32 | 2.53 | 1.32 | 1.40 | $\textbf{4.36}_{\pm\textbf{0.7}}$ |

S: Single, CC: Comparison-Consecutive, CI: Comparison-Irrelevant

a strong classification model that can extract all the existing information from an ECG, and fine-tune LLMs with our QA training set (*i.e.,* applying instruction learning to LLMs).

## 5 Conclusion

In this work, we present ECG-QA, the first QA dataset that incorporates ECG data for question answering tasks. Our dataset is designed to ensure clinical relevance and has been validated by an ECG expert. We created carefully designed question templates, which leverage clinically meaningful attributes extracted from the PTB-XL dataset, to generate a diverse collection of questions, including those that require the comparison of two ECGs. We believe that our dataset has the potential to significantly advance the field of ECG question answering research and contribute to the improvement of clinical practice in analyzing ECG data.

As for the future research directions with ECG-QA, one of the promising avenues is the exploration of multi-modal LLMs that can simultaneously process both ECG signals and natural language. While there is extensive work on LLMs that combine vision and language, there has been limited research on models that integrate signal processing with natural language. We believe our dataset can serve as an excellent testbed for such models.

## 6 Limitation

Despite our best efforts to create the current version of the dataset, there are some limitations as follows.

**Small number of ECGs** Due to the limited number of ECGs available in the original dataset (PTB-XL), our dataset was constructed using a relatively small number of ECGs ($\sim$16k), which leads that questions involving too rare symptoms (*e.g., Wolf-Parkinson-White syndrome*) could not be included. To provide more diverse combinations of ECGs and questions by incorporating questions regarding very rare attributes, we are planning to employ another dataset that is larger than the PTB-XL dataset such as MIMIC-IV-ECG [5], which is planned to be released in late 2023.

**Upper-bound of the dataset** As mentioned in Section 4.2, given the intricacies of the medical field, even medical experts cannot provide 100% accurate diagnoses for all questions. Thus, the upper-bound of the dataset itself is not expected to be 100%. To address this, we conducted experiments demonstrating the estimated upper-bound for each question type and attribute.

**Old version of SCP-ECG standard** Despite SCP-ECG v3.0 being the latest version, the metadata of the original dataset, PTB-XL, follows the SCP-ECG v0.4 standard. Consequently, in ECG-QA, we were constrained to categorize various symptoms based on the SCP-ECG v0.4 standard. However, after investigating how the SCP codes in SCP-ECG v3.0 are categorized, we found that there is only a little difference between SCP-ECG v0.4 and v3.0 regarding the SCP codes used in PTB-XL. Among the SCP codes used in PTB-XL, only one SCP code (BIGU, bigeminal pattern - unknown origin,

SV or Ventricular) has a different representation, which has changed to "SVBIG" (supraventricular bigeminy BIGU bigeminal pattern - unkown origin, SV or Ventricular) in SCP-ECG v3.0. The rest of the SCP codes have maintained their codes and definitions intact in SCP-ECG v3.0. Therefore, we believe that the impact of the differences between the two versions will not be significant in the ECG-QA dataset.

**Automatic generation of paraphrases** Although the paraphrases were manually curated, the initial candidates were automatically generated by ChatGPT, which may not be an optimal strategy. We expect that paraphrases could have been more diverse if we had involved medical practitioners in manually generating paraphrases.

## Acknowledgments and Disclosure of Funding

This work was supported by Institute of Information & communications Technology Planning & Evaluation (IITP) grant (No.2019-0-00075), National Research Foundation of Korea (NRF) grant (NRF-2020H1D3A2A03100945), and Korea Medical Device Development Fund grant (Project Number: 1711138160, KMDF_PR_20200901_0097), funded by the Korea government (MSIT, MOTIE, MOHW, MFDS).

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
