# Supplementary Material

# A   Full list of templates

## A.1   Attribute type descriptions

Individual attributes belonging to each attribute set used in ECG-QA are described in Table 6. For a detailed description of the attributes in SCP code, which are presented as abbreviations, please refer to the PTB-XL dataset [1].

Table 6: Individual attributes belonging to each attribute set used in the ECG-QA dataset.

| Name | Description | Num. | Attributes |
|------|-------------|------|------------|
| SCP code | SCP-ECG statements (symptoms) | 64 | NORM, STTC, MI, HYP, CD, NDT, NST_, DIG, LNGQT, IMI, ASMI, LVH, LAFB, ISC_, IRBBB, 1AVB, IVCD, ISCAL, CRBBB, CLBBB, ILMI, LAO/LAE, AMI, ALMI, ISCIN, INJAS, LMI, ISCIL, LPFB, ISCAS, INJAL, ISCLA, RAO/RAE, ILBBB, IPLMI, ISCAN, IPMI, INJIN, INJLA, PMI, INJIL, ABQRS, PVC, STD_, VCLVH, QWAVE, LOWT, NT_, PAC, LPR, INVT, LVOLT, HVOLT, TAB_, STE_, SR, AFIB, STACH, SARRH, SBRAD, PACE, BIGU, AFLT, SVTAC |
| Noise | Signal artifacts | 4 | Baseline drift, Static noise, Burst noise, Electrode problems |
| Stage of infarction | Stage of myocardial infarction | 5 | None, Unknown, Early, Middle, Late |
| Extra systole | Extra systoles | 3 | Extrasystoles, Ventricular extrasystoles, Supraventricular extrasystoles |
| Heart axis | Direction of heart axis | 4 | Left axis deviation, Right axis deviation Extreme axis deviation, Normal heart axis |
| Numeric feature | Numeric features | 6 | RR interval, P duration, PR interval, QRS duration, QT interval, QT corrected |

## A.2   Question templates

A total of 70 question templates is reported in Table 19.

## A.3   Paraphrases

The overall procedure of generating paraphrases is as follows:

1. 20 candidate paraphrases per template question were automatically generated by OpenAI's ChatGPT with the following prompt:

> Please provide 20 paraphrases for this question. The paraphrased sentences should keep the placeholder which is marked with {}. The paraphrases should entail the original sentence.
>
> ```
> ${question_template}
> ```

2. Based on the machine-generated paraphrases, we manually refined them to ensure the high quality of paraphrases. Specifically, we filtered out the paraphrases that deviated too much

---

[1] https://physionet.org/content/ptb-xl/1.0.3/

from the original question and manually revised them if the specific medical term we were targeting had changed.

3. Then, we randomly selected seven paraphrases for training and validation splits, and three paraphrases for the test split.

As a result, the final paraphrases for each question template are presented in Table 20.

# B    Dataset construction details

## B.1    Regular expressions for parsing lead positions of form-related SCP codes

To utilize grounded lead information of form-related SCP codes, we defined a parser based on regular expressions, which can be shown in Table 16, 17, and 18. During the parsing process, we utilized the Google Cloud Translation API to translate the ECG reports into English, as the majority of the original reports were written in German.

## B.2    Numeric feature extraction

To extract numeric values including heart axis degrees, we utilized NeuroKit2 [16]. The specific method to extract these attributes are described in the following paragraphs.

**Heart axis**    To calculate heart axis degrees, we extracted the magnitude of R peaks from the lead I and lead aVF for each heartbeat existed in the ECG. Then, we computed heart axis degrees by the following equation:

$$x = \frac{1}{N} \sum_{k=1}^{N} \arctan \frac{R_{aVF}^{(k)}}{R_I^{(k)}}$$

where $N$ is the number of heartbeats in the ECG, and $R_l^k$ is the magnitude value of the $k$-th R peak in the lead $l$. To ensure accurate calculations, we did not process the samples with any noises in lead I or lead aVF and restricted them not to be sampled from the relevant questions regarding the heart axis. Based on the calculated heart axis degrees, we categorized them into four standard classes as follows:

$$(\text{Heart axis}) = \begin{cases} \text{Normal} & \text{if} \ -30 \leq x < 90 \\ \text{LAD} & \text{if} \ -90 \leq x < -30 \\ \text{RAD} & \text{if} \ \ \ \ 90 \leq x \leq 180 \\ \text{EAD} & \text{if} -180 \leq x \leq -90 \end{cases}$$

**Numeric feature**    For the numeric features such as *RR interval* or *PR interval*, we extracted the locations of P, Q, R, S, and T waves for each beat present in lead II, and computed the following six numeric features:

- RR interval: The interval seconds between consecutive R peaks.

- P duration: The interval seconds between P onset and P offset.

- PR interval: The interval seconds between P onset and R onset.

- QRS duration: The interval seconds between Q peak and S peak.

- QT interval: The interval seconds between Q peak and T offset.

- QT corrected: $\frac{\text{QT interval}}{\sqrt{\text{RR interval}}}$

Similar to **Heart axis**, we did not process neither the samples with any noises in lead II nor the samples that have less than seven R peaks detected in lead II. Again, we categorized the calculated values into below, within, or above the normal range where the normal range criteria are presented in Table 7.

Table 7: Normal range criteria for numeric features.

| Numeric Feature | Below the normal range | Within the normal range | Above the normal range |
|---|---|---|---|
| RR interval | $x < 0.6$ | $0.6 \leq x \leq 1.0$ | $1.0 < x$ |
| P duration | - | $x \leq 0.12$ | $0.12 < x$ |
| PR interval | $x < 0.12$ | $0.12 \leq x \leq 0.2$ | $0.2 < x$ |
| QRS duration | $x < 0.06$ | $0.06 \leq x \leq 0.11$ | $0.11 < x$ |
| QT interval | $x < 0.33$ | $0.33 \leq x \leq 0.43$ | $0.43 < x$ |
| QT corrected | $x < 0.33$ | $0.33 \leq x \leq 0.45$ | $0.45 < x$ |

### B.3 Sampling strategy

During the sampling process, we sampled more negative samples than positive samples to reflect the clinical reality where normal (negative) cases are much more frequent than abnormal (positive) cases. The sampling size for each question is presented in Table 8. To avoid excessively unbalanced sampling, we set a limit on the number of negative samples if there are too few positive samples, ensuring it does not exceed five times the number of positive samples. For example, if there are 30 positive samples for **Single-Verify** question in the training set, we sample 150 negative samples for that question instead of 200 negative samples. Different sampling strategies for each question type are described in the following paragraphs.

Table 8: Sampling size for each question. It varies depending on the question type.

| Question Type | Train (Pos / Neg) | Validation (Pos / Neg) | Test (Pos / Neg) |
|---|---|---|---|
| S-Verify | 100/200 | 20/40 | 20/40 |
| S-Choose | 10/10 | 2/2 | 2/2 |
| S-Query | 100/200 | 50/100 | 50/100 |
| CC-Verify | 50/100 | 10/20 | 10/20 |
| CC-Query | 50/100 | 25/50 | 25/50 |
| CI-Verify | 50/100 | 10/20 | 10/20 |
| CI-Query | 50/100 | 25/50 | 25/50 |

*Verify* **questions**   Typically there are two answer options (*e.g.,* yes/no), but we included an additional answer option "not sure" especially for diagnostic labels of SCP codes to utilize the likelihood information given by the PTB-XL metadata. Specifically, in the PTB-XL metadata, diagnostic SCP codes are annotated along with their likelihood, indicating the certainty of the diagnoses on a scale of [0, 15, 35, 50, 80, 100]. Considering that the likelihood was derived from keywords in the ECG report and set to zero where no relevant keyword was available (*i.e.,* statements with no adjectives such as "non-specific st-t wave changes" $\rightarrow$ set to 0 likelihood), we classified 0 and 100 as a "certain" diagnosis, and any other value as an "uncertain" diagnosis. Accordingly, when a question asks to verify the presence of a specific diagnostic SCP code in an ECG, the answer is "not sure" if the ECG has been labeled with that SCP code with a likelihood of [15, 35, 50, 80].

Additionally, for questions related to a specific grounded lead position, we defined two types of negative samples: "hard negative" and "soft negative" to add complexity to the dataset. Specifically, "hard negative" samples have the corresponding attribute in other leads but not in the inquired lead, while "soft negative" samples do not have the attribute at all. We sampled at most half of the negative samples from the "hard negative" samples, and the remaining from the "soft negative" samples. By doing so, we intended the QA models to be able to deduce the grounding lead information of specific attributes.

*Choose* **questions**   *Choose* questions involve two attributes, and the answer depends on whether each attribute exists in the ECG, which leads to four possible answer options for each question. For these questions, we considered the absence of both attributes in the ECG as a negative sample, and the presence of at least one attribute as a positive sample. We balanced out the negative samples by

adjusting their number to be no more than five times the maximum number of samples among the three positive options.

***Query* questions**    Similar to *Choose* questions, we regarded negative samples for *Query* questions as the samples that do not have any of the attributes in the inquired attribute set. For example, if a question asks *"What kind of noises does this ECG show?"*, the negative samples are defined as the ECGs that do not have any noises such as *Baseline drift* or *Static noise*. Then, as aforementioned, we restricted the number of negative samples to maintain balance.

Table 9: ECG-QA dataset statistics for different question types.

| Question type | Train | Validation | Test |
|---|---|---|---|
| S-Verify | 62,554 | 10,718 | 13,081 |
| S-Choose | 50,015 | 9,085 | 9,855 |
| S-Query | 46,737 | 11,334 | 18,157 |
| CC-Verify | 21,173 | 2,721 | 4,230 |
| CC-Query | 5,128 | 672 | 1,662 |
| CI-Verify | 39,880 | 7,318 | 7,718 |
| CI-Query | 42,052 | 22,815 | 27,443 |
| *Total* | 267,539 | 64,663 | 82,146 |

Table 10: ECG-QA dataset statistics for different attribute types.

| Attribute type | Train | Validation | Test |
|---|---|---|---|
| SCP code | 201,183 | 47,160 | 60,869 |
| Noise | 26,192 | 6,017 | 7,460 |
| Stage of infarction | 1,233 | 304 | 364 |
| Extra systole | 1,777 | 407 | 493 |
| Heart axis | 1,780 | 395 | 440 |
| Numeric feature | 35,374 | 10,380 | 12,520 |
| *Total* | 267,539 | 64,663 | 82,146 |

## B.4    Dataset statistics

ECG-QA dataset statistics for different question types and attribute types are described in Table 9 and 10.

# C    Experimental details

## C.1    Implementation details

### C.1.1    QA Baselines

The training configurations for QA baselines along with model configurations and resource information are reported in Table 11. We used Adam [9] optimizer for all the models.

1. **per Q-type majority**: This is a prior model that outputs only the most frequent answer in the test split for each question type.

2. **M$^3$AE$^\dagger$** [2]: A multi-modal architecture of ECGs and texts based on Transformer Encoder [25], pre-trained with Masked Language Modeling (MLM), Masked Image Modeling (MIM), and Image Text Matching (ITM) tasks. The architecture comprises three components: 1) separated uni-modal encoders, 2) a multi-modal fusion module, and 3) separated uni-modal decoders for pre-training tasks. Considering the characteristics of the signal data, we employed several 1-d convolutional layers to embed ECGs instead of using a linear layer in the original implementation. Otherwise, we followed the same configurations as the original paper including model architecture and pre-training hyperparameters, except for batch size and learning rate. We used 256 batch size, and 5e-5 learning rate for the pre-training. In the fine-tuning phase, for *Comparison* questions that need to see two ECGs, we concatenate the two ECGs and forwarded them to the uni-modal encoder (the 1st component) to get ECG embeddings.

3. **MedViLL$^\dagger$** [17]: Fusion Transformer (see below) pre-trained with MedViLL [17] methodology. This method implements multi-modal pre-training with ECGs and texts, consisting of MLM and ITM tasks. When pre-training, we followed the same configurations introduced in the original paper such as MLM ratio, except for batch size and learning rate. We used 256 batch size, and 5e-5 learning rate for the pre-training.

4. **Fusion Transformer**: Both the questions and (potentially two) ECGs are concatenated and forwarded into the Transformer Encoder after embedding ECGs with several convolutional layers. To encode questions, we tokenize questions to subword tokens following the BERT

Table 11: Training, model configurations for QA baselines along with resource information. Some model configurations are not reported if not applicable. For any other configurations that are not reported here, we followed the original paper.

| Name | M³AE[†] [2] | MedViLL[†] [17] | Fusion Transf. | Deaf Transf. | Blind Transf. |
|---|---|---|---|---|---|
| ***Model configurations*** | | | | | |
| Conv layers | $[(256, 2, 2)] \times 4$ | $[(256, 2, 2)] \times 4$ | $[(256, 2, 2)] \times 4$ | $[(256, 2, 2)] \times 4$ | N/A |
| Transformer layers | N/A | 12 | 12 | 12 | 12 |
| Hidden dimension | 768 | 768 | 768 | 768 | 768 |
| Attention heads | N/A | 12 | 12 | 12 | 12 |
| ***Training configurations*** | | | | | |
| Training step | 50,000 | 50,000 | 50,000 | 50,000 | 50,000 |
| Local batch size | 16 | 8 | 8 | 16 | 256 |
| Total batch size | 256 | 256 | 256 | 256 | 256 |
| Gradient accumulation step | 4 | 8 | 8 | 4 | 1 |
| Learning rate | 5e-5 | 1e-4 | 1e-4 | 1e-4 | 1e-4 |
| LR scheduler | Tri(0.1, 0.4, 0.5) | Tri(0.1, 0.4, 0.5) | Tri(0.1, 0.4, 0.5) | Tri(0.1, 0.4, 0.5) | Tri(0.1, 0.4, 0.5) |
| ***Resources*** | | | | | |
| GPU device | A6000 $\times$ 4 | RTX 3090 $\times$ 4 | RTX 3090 $\times$ 4 | RTX 3090 $\times$ 4 | RTX 3090 $\times$ 1 |
| VRAM | 48GB $\times$ 4 | 24GB $\times$ 4 | 24GB $\times$ 4 | 24GB $\times$ 4 | 24GB $\times$ 1 |
| Training time | 45 hours | 37 hours | 37 hours | 33 hours | 18 hours |

Conv layers: [(channel size, kernel size, stride)] $\times$ N
Total batch size: Local batch size $\times$ Gradient accumulation step $\times$ Number of GPU devices
Tri(x, y, z): warmup ratio (x), hold ratio (y), exponential decay ratio (z), final lr decaying scale=0.05
A6000: NVIDIA RTX A6000
RTX 3090: NVIDIA GeForce RTX 3090

tokenizer and encode them using the embedding layer of BERT. For Single questions which involve only one ECG, the second ECG is padded by zeros. Then, we separately average-pool each of the modalities, which yields two ECG vectors ($\mathbf{v_1}, \mathbf{v_2} \in \mathbb{R}^{768}$) and a single question vector ($\mathbf{v_3} \in \mathbb{R}^{768}$). For Single questions, $\mathbf{v_2}$ is set to $\mathbf{0}$. We concatenate these vectors and perform multi-label classification task after projecting them onto the final answer space ($[\mathbf{v_1}, \mathbf{v_2}, \mathbf{v_3}] \in \mathbb{R}^{2304} \to \hat{\mathbf{y}} \in \mathbb{R}^{103}$).

5. **Deaf Transformer**: Only the ECGs are forwarded to the Transformer Encoder after being embedded by several convolutional layers. We average-pool the output vectors for each of ECGs, which results in two ECG vectors ($\mathbf{v_1}, \mathbf{v_2} \in \mathbb{R}^{768}$). Similar to the Fusion Transformer, for Single questions, the second ECG is padded by zeros, and $\mathbf{v_2}$ is set to $\mathbf{0}$. Then, we concatenate two vectors and project them onto the final answer space ($[\mathbf{v_1}, \mathbf{v_2}] \in \mathbb{R}^{1536} \to \hat{\mathbf{y}} \in \mathbb{R}^{103}$).

6. **Blind Transformer**: Only the questions are forwarded to the Transformer Encoder. Similar to Fusion Transformer, we follow BERT to tokenize and encode questions. We average-pool the output vectors to get a single question vector, and project the vector onto the final answer space ($\mathbf{v} \in \mathbb{R}^{768} \to \hat{\mathbf{y}} \in \mathbb{R}^{103}$).

### C.1.2 Upper bound models

The procedure of training upper bound models with QA samples is illustrated in Figure 3. For a fair comparison with QA models, we need to train the upper bound models using the comparable dataset with the ECG-QA dataset, requiring converting QA samples to the format that the upper bound models can process. Accordingly, we convert each *Single* QA sample pair (*Question*, *ECG*, *Answer*) in the training set into 4-tuples (*ECG*, *Lead*, *Attribute*, *Label*), and collect all the corresponding (*Attribute*, *Label*) pairs for each unique (*ECG*, *Lead*) pair as shown in Figure 3 (a). Then, each (*ECG*, *Lead*) pair is fed to the upper bound (ECG classification) model along with its labels, and the model is trained by the binary cross entropy (BCE) losses calculated from the corresponding classification heads (See Figure 3 (b)). When training the upper bound model, although the model outputs the scores (*i.e., probabilities*) for all the possible attributes from the classification heads, we only calculate BCE losses from the corresponding attributes that have been labeled for each (*ECG*, *Lead*) pair so that the classification heads for unlabeled attributes are not trained. In addition, if *Lead* in each (*ECG*, *Lead*) pair indicates the specific lead (*e.g., Lead I*), not an entire ECG, we forward the ECG to the model after zero-padding the other leads.

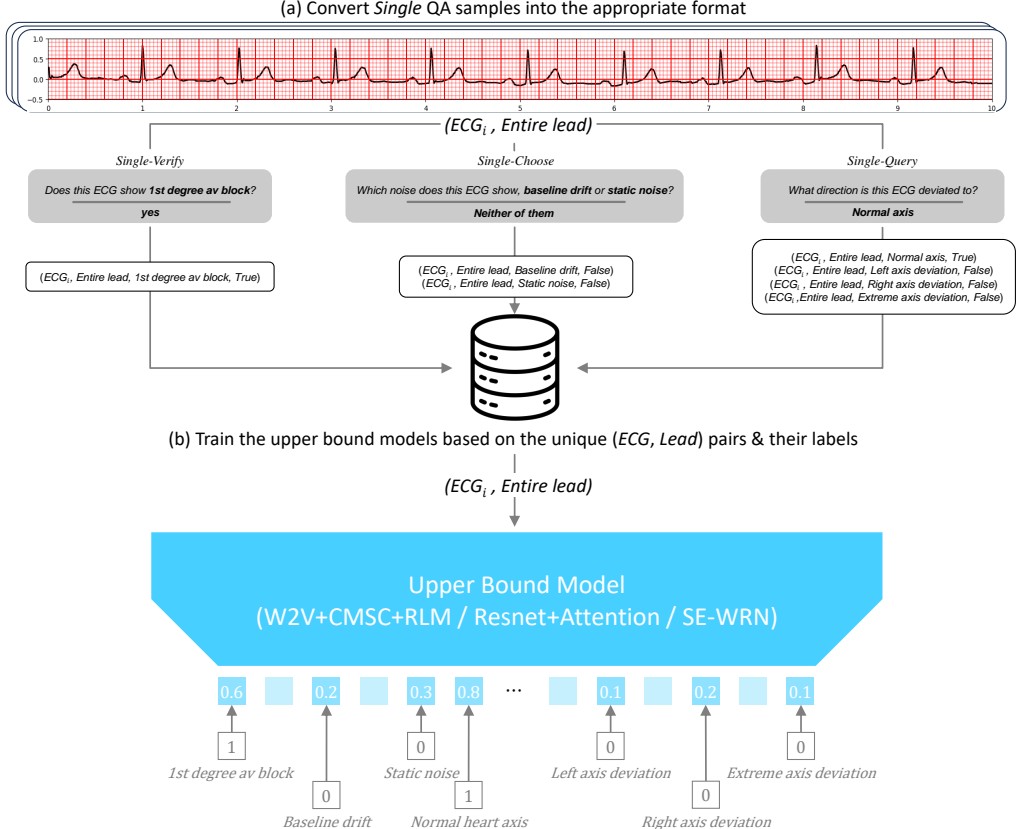

Figure 3: Illustration of the procedure for training upper bound models with QA samples. (a) It starts by converting all the *Single* QA samples into the appropriate format that the upper bound models can process. (b) Then, each unique (*ECG*, *Lead*) pair is fed into the upper bound model, and the binary cross entropy losses are calculated from the corresponding labels to train the model. Here, we do not calculate losses from the classification heads for unlabeled attributes, which are marked with blurred boxes in the final layer.

The strategies for converting QA samples into the 4-tuple format for different question types are described in the following paragraphs.

**Single-Verify samples**    These types of QA samples can be directly convertible to the 4-tuple format according to their answers. For example, if we have a QA sample (*"Does this ECG show **baseline drift** in **lead I**?"*, $ECG_A$, *"yes"*), then this QA sample is converted to ($ECG_A$, *Lead I*, *Baseline drift*, *True*), which means that "$ECG_A$ has *Baseline drift* in *Lead I*". On the other hand, if we have another QA sample where the answer is *"no"*, such as (*"Does this ECG show **baseline drift** in **Lead I**?"*, $ECG_B$, *"no"*), we can derive ($ECG_B$, *Lead I*, *Baseline drift*, *False*), which means that "$ECG_B$ does not have *Baseline drift* in *lead I*".

**Single-Choose samples**    Because there are two relevant attributes in these types of QA samples, we can extract two 4-tuples for each QA sample. For example, if we have a QA sample (*"Which noise does this ECG show in **Lead I**, **baseline drift** or **static noise**?"*, $ECG_A$, *"baseline drift"*), then we convert this QA sample into two 4-tuples: 1) ($ECG_A$, *Lead I*, *Baseline drift*, *True*) and 2) ($ECG_A$, *Lead I*, *Static noise*, *False*).

**Single-Query samples**    Similar to **Single-Choose** samples, we derive 4-tuples for each QA sample as many as the number of the relevant attributes. For example, if we have a QA sample (*"What kind of noises does this ECG show in **Lead I**?"*, $ECG_A$, *"baseline drift, burst noise"*), then we can derive four 4-tuples: 1) ($ECG_A$, *Lead I*, *Baseline drift*, *True*), 2) ($ECG_A$, *Lead I*, *Static noise*, *False*), 3) ($ECG_A$, *Lead I*, *Burst noise*, *True*), and 4) ($ECG_A$, *Lead I*, *Electrode problems*, *False*).

Table 12: Training configurations for upper bound models along with resource information.

| Name | W2V+CMSC+RLM [19] | Resnet+Attention [18] | SE-WRN [6] |
|---|---|---|---|
| ***Training configurations*** | | | |
| Training step | $100,000$ | $100,000$ | $100,000$ |
| Batch size | 64 | 128 | 128 |
| Learning rate | 5e-5 | 1e-4 | 1e-4 |
| LR scheduler | Tri(0.1, 0.4, 0.5) | Tri(0.1, 0.4, 0.5) | Tri(0.1, 0.4, 0.5) |
| ***Resources*** | | | |
| GPU device | RTX 3090 $\times$ 1 | RTX 3090 $\times$ 1 | RTX 3090 $\times$ 1 |
| VRAM | 24GB $\times$ 1 | 24GB $\times$ 1 | 24GB $\times$ 1 |
| Training time | 25 hours | 7 hours | 12 hours |

Tri(x, y, z): warmup ratio (x), hold ratio (y), exponential decay ratio (z), final lr decaying scale=0.05
RTX 3090: NVIDIA GeForce RTX 3090

The detailed upper bound model implementations are described in the following paragraphs. In addition, training configurations for upper bound models including resource information are presented in Table 12.

1. W2V+CMSC+RLM [19]: A Transformer-based model pre-trained with the W2V+CMSC+ RLM [19] method. This model comprises several convolutional layers to extract features from ECGs, followed by Transformer Encoder to contextualize the features. For the model configurations, we follow the original implementation. Specifically, we employ four convolutional layers, each of which has 256 channels with strides of two and kernel lengths of two, and 12 Transformer Encoder layers with 12 self-attention heads and 768 hidden dimensions.

2. Resnet-Attention [18]: This model was introduced in *PhysioNet/Computing in Cardiology Challenge 2021* [21] (CinC 2021), which won first place in the challenge. It is composed of several residual blocks followed by a multi-head attention layer, and two output layers: 1) a conventional BCE loss layer and 2) another loss layer specifically designed for optimizing the challenge score. Since we do not need to optimize the model to achieve a high challenge score, we omit the second loss layer and use only BCE loss to train the model. For any other model configurations such as the number of residual blocks or dropout rate, we follow the original implementation.

3. SE-WRN [6]: This model was introduced in the same challenge (CinC 2021), which won second place. It consists of a series of wide residual networks [30] combined with squeeze-and-excitation modules [7]. In the original paper, they additionally use demographic features such as age and sex, but we do not use those features. We follow the original implementation for the model configurations. However, because the original authors did not mention the kernel lengths for each convolutional layer, we selected kernel lengths of 11 after searching for the best options among 3, 5, 7, and 11.

### C.1.3 Modeling with LLMs

The procedure of modeling the ECG-QA dataset with LLMs is visualized in Figure 4. To provide the results of the ECG interpretations to the LLMs, we transform the outputs from the best upper bound model (SE-WRN [6]) into the text descriptions that LLMs can interpret. Specifically, we apply a threshold value of 0.5 to each score in the model's outputs to get only the attributes whose score is more than 0.5. Then, based on the selected attributes with their scores, we forward the following prompts to LLMs and measure the exact match accuracy. To enable the one-to-one comparison between LLM's answer and GT answer, we induce the LLMs to output the answers only from the valid answer options by giving the candidate options in the prompt for each question.

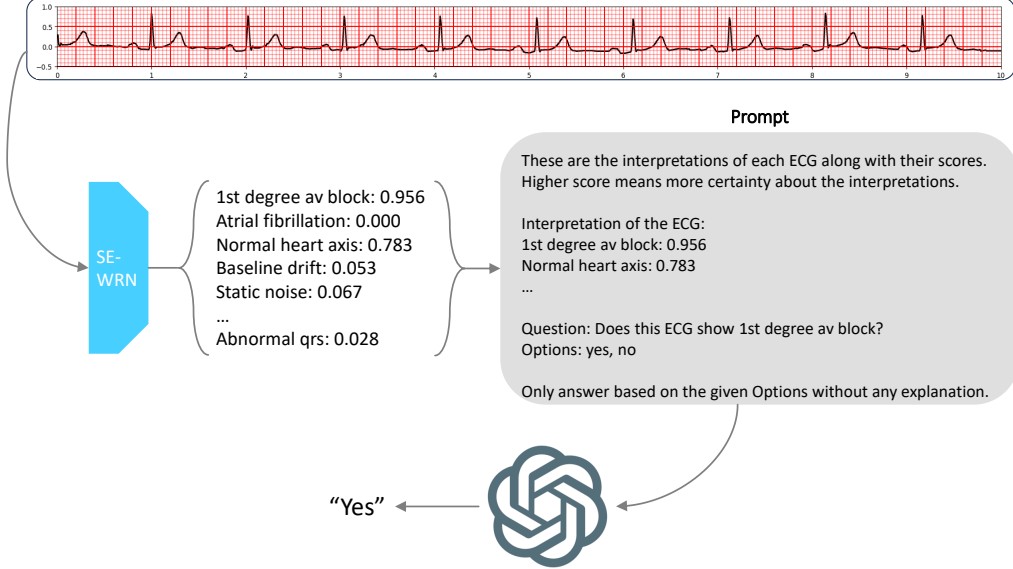

Figure 4: Illustration of modeling with LLMs for the ECG-QA dataset. The ECG is interpreted by the ECG classification model (SE-WRN), which is transformed into the text description for generating the prompt text for the LLMs. When converting the model's output to the text description, we only consider the attributes whose score is more than 0.5. For the prompts for other types of questions such as questions regarding specific lead positions or comparison questions, refer to Supplementary C.1.3.

Prompt for *Single* questions that address the specific leads in which attributes are detected. (*e.g., "Does this ECG show symptoms of **inverted T-waves** in **lead I**?"*)

These are the interpretations of each ECG along with their scores.
Higher score means more certainty about the interpretations.

Interpretation of the ECG in ${lead}:
${attribute#0}: ${score#0}
...
${attribute#N}: ${score#N}

Question: ${question}
Options: ${options}

Only answer based on the given Options without any explanation.

Prompt for *Single* questions that require retrieving specific leads in which attributes are detected.
(*e.g., "What leads are showing symptoms of **inverted T-waves**?"*)

These are the interpretations of each ECG along with their scores.
Higher score means more certainty about the interpretations.

Interpretation of the ECG in lead I:
`${attribute#0}`: `${score#0}`
`...`
`${attribute#N}`: `${score#N}`

`...`

Interpretation of the ECG in lead V6:
`${attribute#0}`: `${score#0}`
`...`
`${attribute#N}`: `${score#N}`

Question: `${question}`
Options: lead I, lead II, lead III, lead aVR, lead aVL, lead aVF, lead V1, lead V2, lead V3, lead V4, lead V5, lead V6

Only answer based on the given Options without any explanation.

Prompt for other *Single* questions.
(*e.g., "Does this ECG show symptoms of **non-diagnostic t abnormalities**?")*

These are the interpretations of each ECG along with their scores.
Higher score means more certainty about the interpretations.

Interpretation of the ECG:
`${attribute#0}`: `${score#0}`
`...`
`${attribute#N}`: `${score#N}`

Question: `${question}`
Options: `${options}`

Only answer based on the given Options without any explanation.

> **Prompt for Comparison-Consecutive questions.**
> *(e.g., Compared to the previous tracing, has the **non-specific st depression** been resolved in the recent tracing?)*

These are the interpretations of each ECG along with their scores.
Higher score means more certainty about the interpretations.

Interpretation of the previous ECG:
`${attribute#0}`: `${score#0}`
...
`${attribute#N}`: `${score#N}`

Interpretation of the recent ECG:
`${attribute#0}`: `${score#0}`
...
`${attribute#N}`: `${score#N}`

Question: `${question}`
Options: `${options}`

Only answer based on the given Options without any explanation.

> **Prompt for Comparison-Irrelevant questions.**
> *(e.g., Compared to the first ECG, has the **non-specific st depression** been resolved in the second ECG?)*

These are the interpretations of each ECG along with their scores.
Higher score means more certainty about the interpretations.

Interpretation of the first ECG:
`${attribute#0}`: `${score#0}`
...
`${attribute#N}`: `${score#N}`

Interpretation of the second ECG:
`${attribute#0}`: `${score#0}`
...
`${attribute#N}`: `${score#N}`

Question: `${question}`
Options: `${options}`

Only answer based on the given Options without any explanation.

### C.2 Detailed experimental results

**Detailed QA results**  The QA baseline results for different question types & attribute types are presented in Table 13.

**Detailed upper bound results**  Table 14 shows the test performances of the upper bound models for different attributes in **Single-Verify** questions, and Table 15 shows the test performances of the QA baselines.

## D  Author statement

We bear all responsibility in case of violation of rights, etc. associated with the ECG-QA dataset.

Table 13: Test performances for different question types & attribute types. We also provide 95% confidence interval across 3 random seeds. The best performances for each question type & attribute type are highlighted with **boldface**.

| Question & Attribute Type | per Q-type majority | M³AE[†] [2] | | MedViLL[†] [17] | | Fusion Transf. | |
|---|---|---|---|---|---|---|---|
| | | EM Acc. | AUROC | EM Acc. | AUROC | EM Acc. | AUROC |
| *S-Verify* | | | | | | | |
| SCP code | 68.0 | $\mathbf{75.3_{\pm0.8}}$ | $0.780_{\pm0.006}$ | $74.1_{\pm0.6}$ | $\mathbf{0.783_{\pm0.015}}$ | $71.6_{\pm0.7}$ | $0.727_{\pm0.011}$ |
| Noise | 67.1 | $68.0_{\pm0.7}$ | $0.595_{\pm0.008}$ | $68.9_{\pm0.5}$ | $0.633_{\pm0.009}$ | $\mathbf{69.7_{\pm0.2}}$ | $\mathbf{0.639_{\pm0.007}}$ |
| Stage of infarction | 71.4 | $81.5_{\pm0.7}$ | $0.834_{\pm0.019}$ | $\mathbf{82.7_{\pm0.7}}$ | $\mathbf{0.870_{\pm0.022}}$ | $76.4_{\pm4.5}$ | $0.805_{\pm0.069}$ |
| Extra systole | 69.6 | $\mathbf{88.7_{\pm1.0}}$ | $0.893_{\pm0.011}$ | $86.8_{\pm1.2}$ | $\mathbf{0.901_{\pm0.012}}$ | $80.4_{\pm2.1}$ | $0.816_{\pm0.022}$ |
| Heart axis | 67.5 | $\mathbf{89.5_{\pm1.0}}$ | $\mathbf{0.938_{\pm0.019}}$ | $87.9_{\pm1.5}$ | $0.921_{\pm0.011}$ | $85.4_{\pm2.2}$ | $0.894_{\pm0.007}$ |
| Numeric feature | 66.7 | $\mathbf{82.2_{\pm0.3}}$ | $\mathbf{0.843_{\pm0.007}}$ | $80.5_{\pm0.7}$ | $0.835_{\pm0.010}$ | $77.7_{\pm1.7}$ | $0.805_{\pm0.014}$ |
| *S-Choose* | | | | | | | |
| SCP code | 31.1 | $\mathbf{57.8_{\pm0.9}}$ | $\mathbf{0.866_{\pm0.004}}$ | $54.7_{\pm0.8}$ | $0.853_{\pm0.001}$ | $46.7_{\pm0.5}$ | $0.808_{\pm0.009}$ |
| Noise | 31.5 | $37.9_{\pm0.4}$ | $0.690_{\pm0.017}$ | $\mathbf{38.0_{\pm0.4}}$ | $\mathbf{0.718_{\pm0.011}}$ | $36.7_{\pm2.3}$ | $0.681_{\pm0.042}$ |
| Stage of infarction | 33.3 | $\mathbf{46.3_{\pm9.6}}$ | $\mathbf{0.792_{\pm0.068}}$ | $38.9_{\pm6.3}$ | $0.667_{\pm0.092}$ | $42.6_{\pm3.6}$ | $0.701_{\pm0.080}$ |
| Extra systole | 27.3 | $53.0_{\pm3.0}$ | $0.785_{\pm0.040}$ | $\mathbf{56.1_{\pm3.0}}$ | $\mathbf{0.804_{\pm0.035}}$ | $37.9_{\pm0.3}$ | $0.771_{\pm0.056}$ |
| Heart axis | 33.3 | $\mathbf{57.4_{\pm1.8}}$ | $0.884_{\pm0.004}$ | $57.4_{\pm4.8}$ | $\mathbf{0.887_{\pm0.036}}$ | $53.7_{\pm4.8}$ | $0.823_{\pm0.030}$ |
| Numeric feature | 34.6 | $54.8_{\pm1.1}$ | $0.826_{\pm0.020}$ | $\mathbf{55.4_{\pm4.4}}$ | $\mathbf{0.839_{\pm0.014}}$ | $52.2_{\pm1.7}$ | $0.815_{\pm0.011}$ |
| *S-Query* | | | | | | | |
| SCP code | 24.5 | $\mathbf{37.1_{\pm0.9}}$ | $\mathbf{0.838_{\pm0.003}}$ | $36.7_{\pm0.8}$ | $0.834_{\pm0.006}$ | $32.7_{\pm0.8}$ | $0.786_{\pm0.015}$ |
| Noise | 39.3 | $45.3_{\pm0.8}$ | $0.725_{\pm0.010}$ | $48.9_{\pm0.5}$ | $0.765_{\pm0.004}$ | $\mathbf{49.9_{\pm0.3}}$ | $\mathbf{0.775_{\pm0.004}}$ |
| Stage of infarction | 28.1 | $\mathbf{60.9_{\pm1.9}}$ | $\mathbf{0.805_{\pm0.017}}$ | $56.9_{\pm4.5}$ | $0.784_{\pm0.012}$ | $53.4_{\pm3.4}$ | $0.750_{\pm0.031}$ |
| Extra systole | 41.5 | $\mathbf{65.8_{\pm1.9}}$ | $0.821_{\pm0.017}$ | $59.3_{\pm4.0}$ | $\mathbf{0.828_{\pm0.011}}$ | $48.4_{\pm4.5}$ | $0.737_{\pm0.016}$ |
| Heart axis | 29.9 | $73.5_{\pm2.4}$ | $\mathbf{0.927_{\pm0.006}}$ | $\mathbf{75.7_{\pm3.2}}$ | $0.927_{\pm0.012}$ | $71.1_{\pm3.1}$ | $0.893_{\pm0.025}$ |
| Numeric feature | 6.8 | $\mathbf{41.9_{\pm1.0}}$ | $\mathbf{0.881_{\pm0.004}}$ | $37.2_{\pm1.0}$ | $0.860_{\pm0.003}$ | $33.5_{\pm0.8}$ | $0.840_{\pm0.005}$ |
| *CC-Verify* | | | | | | | |
| SCP code | 62.0 | $\mathbf{75.8_{\pm0.0}}$ | $\mathbf{0.798_{\pm0.004}}$ | $74.3_{\pm3.3}$ | $0.783_{\pm0.050}$ | $72.6_{\pm0.7}$ | $0.769_{\pm0.005}$ |
| Numeric feature | 66.7 | $73.7_{\pm1.2}$ | $0.754_{\pm0.023}$ | $\mathbf{74.4_{\pm0.5}}$ | $\mathbf{0.755_{\pm0.021}}$ | $68.8_{\pm0.8}$ | $0.710_{\pm0.014}$ |
| *CC-Query* | | | | | | | |
| SCP code | 15.4 | $18.8_{\pm1.1}$ | $0.817_{\pm0.003}$ | $\mathbf{19.5_{\pm1.0}}$ | $\mathbf{0.825_{\pm0.004}}$ | $16.7_{\pm0.6}$ | $0.789_{\pm0.005}$ |
| Numeric feature | 18.9 | $22.0_{\pm2.3}$ | $0.724_{\pm0.009}$ | $\mathbf{25.8_{\pm1.9}}$ | $\mathbf{0.741_{\pm0.005}}$ | $20.8_{\pm2.6}$ | $0.711_{\pm0.012}$ |
| *CI-Verify* | | | | | | | |
| SCP code | 66.0 | $75.6_{\pm1.0}$ | $0.772_{\pm0.012}$ | $\mathbf{78.2_{\pm1.8}}$ | $\mathbf{0.830_{\pm0.022}}$ | $68.3_{\pm0.5}$ | $0.726_{\pm0.010}$ |
| Numeric feature | 66.7 | $\mathbf{72.7_{\pm1.6}}$ | $\mathbf{0.741_{\pm0.014}}$ | $70.5_{\pm0.6}$ | $0.729_{\pm0.009}$ | $66.6_{\pm2.1}$ | $0.692_{\pm0.009}$ |
| *CI-Query* | | | | | | | |
| SCP code | 0.66 | $\mathbf{3.10_{\pm0.09}}$ | $0.743_{\pm0.009}$ | $2.32_{\pm0.12}$ | $\mathbf{0.760_{\pm0.006}}$ | $1.34_{\pm0.06}$ | $0.706_{\pm0.005}$ |
| Numeric feature | 3.08 | $\mathbf{9.33_{\pm0.70}}$ | $0.727_{\pm0.006}$ | $9.05_{\pm0.48}$ | $\mathbf{0.732_{\pm0.008}}$ | $6.23_{\pm0.32}$ | $0.684_{\pm0.002}$ |

S: Single, CC: Comparison-Consecutive, CI: Comparison-Irrelevant

Table 14: Test performances of upper bound models for different attributes in **Single-Verify** questions. Attributes are sorted in alphabetical order. We provide 95% confidence interval across 3 random seeds. Note that MAX takes the maximal score across the upper bound models for each random seed.

| Attribute name | W2V+CMSC+RLM [19] | | Resnet-Attention [18] | | SE-WRN [6] | | MAX | |
|---|---|---|---|---|---|---|---|---|
| | Acc. | AUROC | Acc. | AUROC | Acc. | AUROC | Acc. | AUROC |
| *Macro-average* | $83.0_{\pm0.4}$ | $0.864_{\pm0.003}$ | $82.6_{\pm0.3}$ | $0.875_{\pm0.002}$ | $83.1_{\pm0.3}$ | $0.883_{\pm0.002}$ | $85.4_{\pm0.4}$ | $0.907_{\pm0.002}$ |
| Abnormal qrs | $72.8_{\pm2.9}$ | $0.731_{\pm0.097}$ | $69.4_{\pm4.4}$ | $0.753_{\pm0.034}$ | $71.7_{\pm3.3}$ | $0.734_{\pm0.046}$ | $73.3_{\pm1.9}$ | $0.779_{\pm0.033}$ |
| Above the normal range of p duration | $85.6_{\pm2.2}$ | $0.910_{\pm0.013}$ | $87.8_{\pm3.9}$ | $0.938_{\pm0.038}$ | $86.7_{\pm0.0}$ | $0.939_{\pm0.014}$ | $88.3_{\pm3.3}$ | $0.948_{\pm0.018}$ |
| Above the normal range of pr interval | $88.3_{\pm0.0}$ | $0.962_{\pm0.010}$ | $86.1_{\pm2.9}$ | $0.954_{\pm0.029}$ | $91.1_{\pm2.2}$ | $0.973_{\pm0.015}$ | $91.1_{\pm2.2}$ | $0.976_{\pm0.010}$ |
| Above the normal range of qrs duration | $86.1_{\pm1.1}$ | $0.915_{\pm0.043}$ | $89.4_{\pm1.1}$ | $0.958_{\pm0.004}$ | $87.8_{\pm2.9}$ | $0.932_{\pm0.007}$ | $89.4_{\pm1.1}$ | $0.958_{\pm0.002}$ |
| Above the normal range of qt corrected | $71.1_{\pm2.2}$ | $0.813_{\pm0.034}$ | $74.4_{\pm2.2}$ | $0.839_{\pm0.003}$ | $76.1_{\pm2.9}$ | $0.862_{\pm0.010}$ | $77.2_{\pm1.1}$ | $0.862_{\pm0.010}$ |
| Above the normal range of qt interval | $87.8_{\pm5.4}$ | $0.940_{\pm0.043}$ | $85.0_{\pm3.8}$ | $0.949_{\pm0.022}$ | $85.0_{\pm5.7}$ | $0.940_{\pm0.038}$ | $90.6_{\pm2.9}$ | $0.966_{\pm0.011}$ |
| Above the normal range of rr interval | $98.3_{\pm0.0}$ | $1.000_{\pm0.000}$ | $98.3_{\pm1.9}$ | $0.999_{\pm0.002}$ | $97.2_{\pm1.1}$ | $0.999_{\pm0.001}$ | $98.9_{\pm1.1}$ | $1.000_{\pm0.000}$ |
| Any diagnostic symptoms | $85.6_{\pm2.2}$ | $0.903_{\pm0.031}$ | $81.7_{\pm1.9}$ | $0.878_{\pm0.016}$ | $81.7_{\pm3.3}$ | $0.865_{\pm0.043}$ | $85.6_{\pm2.2}$ | $0.914_{\pm0.012}$ |
| Any form-related symptoms | $81.1_{\pm4.4}$ | $0.820_{\pm0.023}$ | $76.7_{\pm1.9}$ | $0.805_{\pm0.021}$ | $74.4_{\pm3.9}$ | $0.746_{\pm0.031}$ | $81.1_{\pm4.4}$ | $0.820_{\pm0.023}$ |
| Any kind of abnormal symptoms | $76.7_{\pm1.9}$ | $0.847_{\pm0.009}$ | $77.2_{\pm9.7}$ | $0.863_{\pm0.044}$ | $81.7_{\pm1.9}$ | $0.864_{\pm0.011}$ | $82.8_{\pm3.9}$ | $0.877_{\pm0.030}$ |
| Any kind of extra systoles | $94.7_{\pm1.3}$ | $0.960_{\pm0.020}$ | $91.3_{\pm1.3}$ | $0.875_{\pm0.024}$ | $85.3_{\pm5.7}$ | $0.874_{\pm0.035}$ | $94.7_{\pm1.3}$ | $0.960_{\pm0.020}$ |
| Any kind of noises | $74.0_{\pm0.8}$ | $0.779_{\pm0.008}$ | $70.3_{\pm1.2}$ | $0.749_{\pm0.006}$ | $70.9_{\pm0.8}$ | $0.755_{\pm0.003}$ | $74.0_{\pm0.8}$ | $0.779_{\pm0.008}$ |
| Any rhythm-related symptoms | $75.0_{\pm9.8}$ | $0.794_{\pm0.122}$ | $82.2_{\pm6.6}$ | $0.837_{\pm0.050}$ | $83.9_{\pm4.7}$ | $0.882_{\pm0.043}$ | $85.6_{\pm5.4}$ | $0.891_{\pm0.038}$ |
| Atrial fibrillation | $94.4_{\pm1.1}$ | $0.936_{\pm0.054}$ | $90.6_{\pm1.1}$ | $0.922_{\pm0.016}$ | $87.2_{\pm2.9}$ | $0.954_{\pm0.002}$ | $94.4_{\pm1.1}$ | $0.961_{\pm0.010}$ |
| Atrial flutter | $94.4_{\pm0.0}$ | $0.907_{\pm0.067}$ | $94.4_{\pm0.0}$ | $0.907_{\pm0.098}$ | $94.4_{\pm0.0}$ | $0.987_{\pm0.020}$ | $94.4_{\pm0.0}$ | $0.991_{\pm0.013}$ |
| Atrial premature complex | $83.3_{\pm5.0}$ | $0.950_{\pm0.023}$ | $78.3_{\pm0.0}$ | $0.871_{\pm0.015}$ | $82.2_{\pm2.2}$ | $0.901_{\pm0.038}$ | $83.9_{\pm3.9}$ | $0.950_{\pm0.023}$ |
| Baseline drift | $78.2_{\pm0.7}$ | $0.820_{\pm0.007}$ | $78.9_{\pm0.6}$ | $0.839_{\pm0.014}$ | $80.5_{\pm1.0}$ | $0.889_{\pm0.003}$ | $80.5_{\pm1.0}$ | $0.889_{\pm0.003}$ |
| Below the normal range of pr interval | $91.7_{\pm1.9}$ | $0.968_{\pm0.008}$ | $84.4_{\pm1.1}$ | $0.931_{\pm0.007}$ | $91.1_{\pm3.9}$ | $0.951_{\pm0.025}$ | $92.8_{\pm2.9}$ | $0.968_{\pm0.008}$ |
| Below the normal range of qrs duration | $82.2_{\pm4.7}$ | $0.975_{\pm0.024}$ | $88.9_{\pm1.1}$ | $0.995_{\pm0.005}$ | $91.7_{\pm3.8}$ | $0.989_{\pm0.006}$ | $92.2_{\pm2.9}$ | $0.995_{\pm0.005}$ |
| Below the normal range of qt corrected | $90.0_{\pm0.0}$ | $0.953_{\pm0.016}$ | $92.2_{\pm1.1}$ | $0.988_{\pm0.010}$ | $86.1_{\pm2.9}$ | $0.973_{\pm0.011}$ | $92.2_{\pm1.1}$ | $0.988_{\pm0.010}$ |
| Below the normal range of qt interval | $81.7_{\pm3.8}$ | $0.922_{\pm0.009}$ | $86.1_{\pm4.4}$ | $0.935_{\pm0.019}$ | $83.9_{\pm3.9}$ | $0.940_{\pm0.026}$ | $87.8_{\pm1.1}$ | $0.954_{\pm0.004}$ |
| Below the normal range of rr interval | $98.3_{\pm0.0}$ | $0.998_{\pm0.003}$ | $98.3_{\pm1.9}$ | $0.999_{\pm0.002}$ | $100.0_{\pm0.0}$ | $1.000_{\pm0.000}$ | $100.0_{\pm0.0}$ | $1.000_{\pm0.000}$ |
| Bigeminal pattern (unknown origin, supraventricular, or ventricular) | $85.3_{\pm1.3}$ | $0.888_{\pm0.061}$ | $86.0_{\pm0.0}$ | $0.887_{\pm0.042}$ | $85.3_{\pm4.7}$ | $0.853_{\pm0.078}$ | $87.3_{\pm2.6}$ | $0.912_{\pm0.057}$ |
| Burst noise | $75.5_{\pm0.7}$ | $0.756_{\pm0.033}$ | $70.1_{\pm0.2}$ | $0.671_{\pm0.006}$ | $67.9_{\pm0.4}$ | $0.697_{\pm0.056}$ | $75.5_{\pm0.7}$ | $0.761_{\pm0.023}$ |
| Complete left bundle branch block | $96.1_{\pm2.2}$ | $0.993_{\pm0.004}$ | $95.0_{\pm0.0}$ | $0.977_{\pm0.009}$ | $94.4_{\pm1.1}$ | $0.992_{\pm0.003}$ | $96.1_{\pm2.2}$ | $0.994_{\pm0.003}$ |
| Complete right bundle branch block | $98.9_{\pm1.1}$ | $1.000_{\pm0.000}$ | $96.7_{\pm1.9}$ | $1.000_{\pm0.000}$ | $97.2_{\pm1.1}$ | $0.998_{\pm0.008}$ | $98.9_{\pm1.1}$ | $1.000_{\pm0.000}$ |
| Conduction disturbance | $90.0_{\pm0.0}$ | $0.926_{\pm0.042}$ | $88.9_{\pm1.1}$ | $0.946_{\pm0.013}$ | $86.1_{\pm2.2}$ | $0.938_{\pm0.016}$ | $90.0_{\pm0.0}$ | $0.952_{\pm0.006}$ |
| Digitalis effect | $80.7_{\pm0.0}$ | $0.932_{\pm0.013}$ | $81.9_{\pm2.3}$ | $0.950_{\pm0.012}$ | $83.6_{\pm1.1}$ | $0.947_{\pm0.019}$ | $83.6_{\pm1.1}$ | $0.953_{\pm0.013}$ |
| Early stage of myocardial infarction | $78.5_{\pm1.1}$ | $0.848_{\pm0.023}$ | $79.1_{\pm4.0}$ | $0.863_{\pm0.056}$ | $78.0_{\pm3.3}$ | $0.864_{\pm0.023}$ | $80.8_{\pm2.9}$ | $0.889_{\pm0.008}$ |
| Electrodes problems | $83.8_{\pm1.0}$ | $0.600_{\pm0.124}$ | $82.8_{\pm2.6}$ | $0.716_{\pm0.047}$ | $81.3_{\pm1.0}$ | $0.720_{\pm0.069}$ | $83.8_{\pm1.0}$ | $0.732_{\pm0.061}$ |
| Extrasystoles | $90.6_{\pm2.2}$ | $0.926_{\pm0.044}$ | $89.4_{\pm2.9}$ | $0.942_{\pm0.055}$ | $86.1_{\pm2.9}$ | $0.915_{\pm0.043}$ | $90.6_{\pm2.9}$ | $0.953_{\pm0.051}$ |
| Extreme axis deviation | $86.0_{\pm3.4}$ | $0.936_{\pm0.037}$ | $89.5_{\pm2.0}$ | $0.941_{\pm0.017}$ | $90.1_{\pm5.0}$ | $0.953_{\pm0.018}$ | $91.8_{\pm3.0}$ | $0.960_{\pm0.009}$ |
| First degree av block | $87.2_{\pm4.7}$ | $0.947_{\pm0.015}$ | $78.9_{\pm1.1}$ | $0.882_{\pm0.003}$ | $85.0_{\pm3.3}$ | $0.913_{\pm0.011}$ | $88.3_{\pm3.3}$ | $0.947_{\pm0.015}$ |
| High qrs voltage | $79.8_{\pm0.5}$ | $0.724_{\pm0.045}$ | $80.2_{\pm1.2}$ | $0.695_{\pm0.036}$ | $78.9_{\pm2.0}$ | $0.705_{\pm0.020}$ | $80.7_{\pm0.8}$ | $0.726_{\pm0.040}$ |
| Hypertrophy | $72.2_{\pm2.9}$ | $0.780_{\pm0.071}$ | $81.7_{\pm1.9}$ | $0.882_{\pm0.004}$ | $80.6_{\pm4.7}$ | $0.870_{\pm0.032}$ | $83.3_{\pm1.9}$ | $0.889_{\pm0.014}$ |
| Incomplete left bundle branch block | $81.0_{\pm1.3}$ | $0.900_{\pm0.041}$ | $79.7_{\pm1.3}$ | $0.891_{\pm0.081}$ | $81.0_{\pm1.3}$ | $0.954_{\pm0.018}$ | $81.7_{\pm1.3}$ | $0.954_{\pm0.018}$ |
| Incomplete right bundle branch block | $77.8_{\pm4.4}$ | $0.873_{\pm0.050}$ | $75.0_{\pm1.9}$ | $0.889_{\pm0.036}$ | $73.9_{\pm2.9}$ | $0.843_{\pm0.050}$ | $78.3_{\pm3.3}$ | $0.905_{\pm0.012}$ |
| Inverted t-waves | $85.2_{\pm1.1}$ | $0.936_{\pm0.007}$ | $84.5_{\pm0.6}$ | $0.944_{\pm0.003}$ | $87.3_{\pm0.5}$ | $0.949_{\pm0.006}$ | $87.3_{\pm0.5}$ | $0.950_{\pm0.005}$ |
| Ischemic | $72.1_{\pm0.9}$ | $0.765_{\pm0.008}$ | $71.7_{\pm1.1}$ | $0.803_{\pm0.012}$ | $71.3_{\pm0.8}$ | $0.835_{\pm0.017}$ | $72.1_{\pm0.9}$ | $0.835_{\pm0.017}$ |
| Late stage of myocardial infarction | $82.3_{\pm2.7}$ | $0.661_{\pm0.043}$ | $83.0_{\pm1.3}$ | $0.648_{\pm0.036}$ | $85.0_{\pm1.3}$ | $0.681_{\pm0.055}$ | $85.0_{\pm1.3}$ | $0.699_{\pm0.034}$ |
| Left anterior fascicular block | $96.1_{\pm1.1}$ | $1.000_{\pm0.000}$ | $96.1_{\pm2.2}$ | $0.998_{\pm0.001}$ | $98.3_{\pm1.9}$ | $0.995_{\pm0.005}$ | $98.9_{\pm1.1}$ | $1.000_{\pm0.000}$ |
| Left atrial overload/enlargement | $68.3_{\pm1.9}$ | $0.725_{\pm0.087}$ | $67.8_{\pm2.2}$ | $0.717_{\pm0.039}$ | $70.6_{\pm2.2}$ | $0.762_{\pm0.033}$ | $71.1_{\pm1.1}$ | $0.780_{\pm0.034}$ |
| Left axis deviation | $85.0_{\pm1.9}$ | $0.909_{\pm0.029}$ | $88.3_{\pm1.9}$ | $0.924_{\pm0.015}$ | $87.8_{\pm2.2}$ | $0.951_{\pm0.026}$ | $88.9_{\pm2.2}$ | $0.953_{\pm0.023}$ |
| Left posterior fascicular block | $80.6_{\pm2.9}$ | $0.972_{\pm0.005}$ | $79.4_{\pm2.2}$ | $0.986_{\pm0.009}$ | $84.4_{\pm1.1}$ | $0.996_{\pm0.004}$ | $84.4_{\pm1.1}$ | $0.996_{\pm0.004}$ |
| Left ventricular hypertrophy | $79.4_{\pm1.1}$ | $0.910_{\pm0.018}$ | $87.2_{\pm1.1}$ | $0.963_{\pm0.018}$ | $88.3_{\pm1.9}$ | $0.969_{\pm0.016}$ | $88.9_{\pm1.1}$ | $0.973_{\pm0.009}$ |
| Long qt-interval | $78.4_{\pm0.0}$ | $0.632_{\pm0.013}$ | $78.4_{\pm0.0}$ | $0.646_{\pm0.164}$ | $78.4_{\pm0.0}$ | $0.842_{\pm0.061}$ | $78.4_{\pm0.0}$ | $0.842_{\pm0.061}$ |
| Low amplitude t-wave | $72.3_{\pm0.7}$ | $0.786_{\pm0.019}$ | $73.4_{\pm0.6}$ | $0.816_{\pm0.018}$ | $74.5_{\pm1.1}$ | $0.858_{\pm0.004}$ | $74.5_{\pm1.1}$ | $0.858_{\pm0.004}$ |
| Low qrs voltages in the frontal and horizontal leads | $70.6_{\pm0.9}$ | $0.751_{\pm0.034}$ | $74.3_{\pm2.6}$ | $0.850_{\pm0.028}$ | $75.7_{\pm1.4}$ | $0.850_{\pm0.015}$ | $76.5_{\pm1.0}$ | $0.863_{\pm0.018}$ |
| Middle stage of myocardial infarction | $74.4_{\pm4.7}$ | $0.782_{\pm0.019}$ | $75.0_{\pm0.0}$ | $0.821_{\pm0.027}$ | $68.9_{\pm2.9}$ | $0.762_{\pm0.034}$ | $76.1_{\pm2.2}$ | $0.821_{\pm0.027}$ |
| Myocardial infarction | $77.9_{\pm1.1}$ | $0.817_{\pm0.017}$ | $80.2_{\pm1.1}$ | $0.873_{\pm0.014}$ | $77.5_{\pm1.2}$ | $0.844_{\pm0.007}$ | $80.2_{\pm1.1}$ | $0.873_{\pm0.014}$ |
| Non-diagnostic t abnormalities | $71.6_{\pm1.3}$ | $0.755_{\pm0.012}$ | $71.5_{\pm1.1}$ | $0.805_{\pm0.014}$ | $76.1_{\pm0.8}$ | $0.841_{\pm0.003}$ | $76.1_{\pm0.8}$ | $0.841_{\pm0.003}$ |
| Non-specific intraventricular conduction disturbance (block) | $71.7_{\pm0.0}$ | $0.819_{\pm0.027}$ | $67.8_{\pm1.1}$ | $0.650_{\pm0.065}$ | $68.3_{\pm3.3}$ | $0.620_{\pm0.043}$ | $71.7_{\pm0.0}$ | $0.819_{\pm0.027}$ |
| Non-specific ischemic | $83.9_{\pm1.1}$ | $0.917_{\pm0.039}$ | $84.4_{\pm1.1}$ | $0.945_{\pm0.005}$ | $83.9_{\pm3.9}$ | $0.947_{\pm0.026}$ | $85.6_{\pm1.1}$ | $0.957_{\pm0.013}$ |
| Non-specific st changes | $77.8_{\pm0.1}$ | $0.696_{\pm0.014}$ | $78.2_{\pm1.1}$ | $0.697_{\pm0.008}$ | $77.7_{\pm0.3}$ | $0.788_{\pm0.023}$ | $78.2_{\pm0.4}$ | $0.788_{\pm0.023}$ |
| Non-specific st depression | $84.5_{\pm0.4}$ | $0.901_{\pm0.009}$ | $83.1_{\pm0.1}$ | $0.904_{\pm0.006}$ | $84.6_{\pm0.7}$ | $0.918_{\pm0.006}$ | $84.9_{\pm0.4}$ | $0.918_{\pm0.006}$ |
| Non-specific st elevation | $83.3_{\pm0.0}$ | $0.467_{\pm0.021}$ | $83.3_{\pm0.0}$ | $0.850_{\pm0.112}$ | $83.3_{\pm0.0}$ | $0.713_{\pm0.171}$ | $83.3_{\pm0.0}$ | $0.850_{\pm0.112}$ |
| Non-specific t-wave changes | $75.3_{\pm0.6}$ | $0.782_{\pm0.031}$ | $73.7_{\pm1.5}$ | $0.816_{\pm0.014}$ | $77.4_{\pm1.2}$ | $0.850_{\pm0.011}$ | $77.4_{\pm1.2}$ | $0.850_{\pm0.011}$ |
| Normal ecg | $82.2_{\pm1.9}$ | $0.908_{\pm0.027}$ | $80.0_{\pm3.3}$ | $0.887_{\pm0.029}$ | $80.6_{\pm3.9}$ | $0.864_{\pm0.050}$ | $82.8_{\pm2.9}$ | $0.918_{\pm0.010}$ |
| Normal functioning artificial pacemaker | $100.0_{\pm0.0}$ | $1.000_{\pm0.000}$ | $97.9_{\pm0.0}$ | $0.991_{\pm0.006}$ | $97.2_{\pm1.4}$ | $0.997_{\pm0.006}$ | $100.0_{\pm0.0}$ | $1.000_{\pm0.000}$ |
| Normal heart axis | $94.4_{\pm2.9}$ | $0.982_{\pm0.020}$ | $94.4_{\pm1.1}$ | $0.983_{\pm0.014}$ | $95.6_{\pm2.2}$ | $0.992_{\pm0.009}$ | $96.1_{\pm1.1}$ | $0.997_{\pm0.001}$ |
| Prolonged pr interval | $90.0_{\pm1.9}$ | $0.960_{\pm0.011}$ | $87.8_{\pm2.2}$ | $0.974_{\pm0.024}$ | $86.1_{\pm2.9}$ | $0.988_{\pm0.012}$ | $90.6_{\pm1.1}$ | $0.992_{\pm0.013}$ |
| Q waves present | $83.3_{\pm0.4}$ | $0.892_{\pm0.010}$ | $83.1_{\pm0.9}$ | $0.889_{\pm0.005}$ | $83.4_{\pm2.2}$ | $0.911_{\pm0.005}$ | $84.3_{\pm0.5}$ | $0.911_{\pm0.005}$ |
| Right atrial overload/enlargement | $76.2_{\pm1.2}$ | $0.877_{\pm0.077}$ | $75.6_{\pm4.7}$ | $0.955_{\pm0.047}$ | $84.5_{\pm4.7}$ | $0.979_{\pm0.008}$ | $84.5_{\pm4.7}$ | $0.979_{\pm0.008}$ |
| Right axis deviation | $86.1_{\pm2.2}$ | $0.955_{\pm0.009}$ | $92.8_{\pm1.1}$ | $0.962_{\pm0.008}$ | $91.1_{\pm1.1}$ | $0.959_{\pm0.027}$ | $92.8_{\pm1.1}$ | $0.971_{\pm0.009}$ |
| Sinus arrhythmia | $77.8_{\pm2.2}$ | $0.847_{\pm0.079}$ | $73.3_{\pm3.8}$ | $0.689_{\pm0.023}$ | $78.9_{\pm4.7}$ | $0.795_{\pm0.097}$ | $80.6_{\pm2.9}$ | $0.869_{\pm0.038}$ |
| Sinus bradycardia | $86.7_{\pm1.9}$ | $0.944_{\pm0.016}$ | $86.7_{\pm5.0}$ | $0.958_{\pm0.006}$ | $87.8_{\pm1.1}$ | $0.960_{\pm0.018}$ | $89.4_{\pm2.2}$ | $0.965_{\pm0.009}$ |
| Sinus rhythm | $79.4_{\pm4.7}$ | $0.854_{\pm0.034}$ | $76.1_{\pm3.9}$ | $0.819_{\pm0.036}$ | $78.3_{\pm5.0}$ | $0.848_{\pm0.024}$ | $81.1_{\pm4.4}$ | $0.864_{\pm0.018}$ |
| Sinus tachycardia | $96.1_{\pm1.1}$ | $0.987_{\pm0.010}$ | $96.1_{\pm2.9}$ | $0.995_{\pm0.007}$ | $95.0_{\pm1.9}$ | $0.990_{\pm0.006}$ | $97.2_{\pm1.1}$ | $0.996_{\pm0.004}$ |
| St/t change | $78.9_{\pm1.1}$ | $0.857_{\pm0.009}$ | $80.0_{\pm3.8}$ | $0.868_{\pm0.008}$ | $78.9_{\pm1.1}$ | $0.833_{\pm0.022}$ | $81.1_{\pm2.2}$ | $0.868_{\pm0.008}$ |
| Static noise | $68.9_{\pm1.5}$ | $0.703_{\pm0.019}$ | $69.4_{\pm0.9}$ | $0.714_{\pm0.010}$ | $71.1_{\pm1.1}$ | $0.736_{\pm0.014}$ | $71.1_{\pm1.1}$ | $0.736_{\pm0.014}$ |
| Subendocardial injury | $77.4_{\pm0.8}$ | $0.932_{\pm0.036}$ | $75.9_{\pm2.2}$ | $0.951_{\pm0.024}$ | $75.4_{\pm2.1}$ | $0.968_{\pm0.006}$ | $77.9_{\pm0.5}$ | $0.969_{\pm0.007}$ |
| Supraventricular extrasystoles | $85.6_{\pm2.9}$ | $0.883_{\pm0.042}$ | $76.7_{\pm3.8}$ | $0.765_{\pm0.049}$ | $83.3_{\pm3.8}$ | $0.838_{\pm0.032}$ | $86.1_{\pm2.9}$ | $0.883_{\pm0.012}$ |
| Supraventricular tachycardia | $85.2_{\pm3.6}$ | $0.733_{\pm0.206}$ | $83.3_{\pm0.0}$ | $0.956_{\pm0.067}$ | $92.6_{\pm3.6}$ | $0.941_{\pm0.077}$ | $92.6_{\pm3.6}$ | $0.963_{\pm0.073}$ |
| T-wave abnormality | $83.3_{\pm0.0}$ | $0.482_{\pm0.054}$ | $83.3_{\pm0.0}$ | $0.515_{\pm0.050}$ | $83.3_{\pm0.0}$ | $0.461_{\pm0.107}$ | $83.3_{\pm0.0}$ | $0.532_{\pm0.024}$ |
| Ventricular extrasystoles | $80.6_{\pm1.1}$ | $0.897_{\pm0.024}$ | $73.3_{\pm0.0}$ | $0.870_{\pm0.042}$ | $79.4_{\pm2.9}$ | $0.894_{\pm0.034}$ | $81.1_{\pm1.1}$ | $0.910_{\pm0.027}$ |
| Ventricular premature complex | $92.2_{\pm1.1}$ | $0.974_{\pm0.007}$ | $92.8_{\pm2.9}$ | $0.957_{\pm0.009}$ | $91.1_{\pm1.1}$ | $0.974_{\pm0.009}$ | $93.3_{\pm1.9}$ | $0.978_{\pm0.004}$ |
| Voltage criteria (qrs) for left ventricular hypertrophy | $73.1_{\pm2.0}$ | $0.760_{\pm0.042}$ | $82.0_{\pm1.3}$ | $0.907_{\pm0.017}$ | $82.6_{\pm1.9}$ | $0.898_{\pm0.027}$ | $82.6_{\pm1.9}$ | $0.912_{\pm0.022}$ |
| Within the normal range of p duration | $84.4_{\pm5.8}$ | $0.948_{\pm0.027}$ | $78.3_{\pm3.8}$ | $0.889_{\pm0.029}$ | $76.1_{\pm2.9}$ | $0.875_{\pm0.032}$ | $85.0_{\pm5.0}$ | $0.948_{\pm0.027}$ |
| Within the normal range of pr interval | $85.0_{\pm1.9}$ | $0.916_{\pm0.011}$ | $73.3_{\pm1.9}$ | $0.854_{\pm0.020}$ | $79.4_{\pm2.9}$ | $0.876_{\pm0.009}$ | $85.0_{\pm1.9}$ | $0.916_{\pm0.011}$ |
| Within the normal range of qrs duration | $75.6_{\pm3.9}$ | $0.860_{\pm0.033}$ | $82.8_{\pm5.8}$ | $0.925_{\pm0.045}$ | $81.1_{\pm2.2}$ | $0.889_{\pm0.015}$ | $83.3_{\pm5.0}$ | $0.925_{\pm0.045}$ |
| Within the normal range of qt corrected | $80.0_{\pm5.0}$ | $0.886_{\pm0.020}$ | $79.4_{\pm4.8}$ | $0.874_{\pm0.021}$ | $76.7_{\pm8.6}$ | $0.873_{\pm0.057}$ | $82.8_{\pm4.4}$ | $0.903_{\pm0.020}$ |
| Within the normal range of qt interval | $82.8_{\pm2.2}$ | $0.891_{\pm0.029}$ | $85.6_{\pm3.9}$ | $0.922_{\pm0.026}$ | $78.3_{\pm5.7}$ | $0.876_{\pm0.023}$ | $85.6_{\pm3.9}$ | $0.922_{\pm0.026}$ |
| Within the normal range of rr interval | $96.7_{\pm1.9}$ | $0.997_{\pm0.004}$ | $93.9_{\pm2.2}$ | $0.985_{\pm0.001}$ | $88.9_{\pm3.9}$ | $0.962_{\pm0.020}$ | $96.7_{\pm1.9}$ | $0.997_{\pm0.004}$ |

Table 15: Test performances of QA baselines for different attributes in **Single-Verify** questions. Attributes are sorted in alphabetical order. We provide 95% confidence interval across 3 random seeds.

| Attribute name | M³AE[†] [2] | | MedViLL[†] [17] | | Fusion Transf. | |
|---|---|---|---|---|---|---|
| | Acc. | AUROC | Acc. | AUROC | Acc. | AUROC |
| *Macro-average* | $80.8_{\pm0.3}$ | $0.808_{\pm0.006}$ | $79.8_{\pm0.3}$ | $0.809_{\pm0.005}$ | $76.4_{\pm0.6}$ | $0.764_{\pm0.010}$ |
| Abnormal qrs | $76.7_{\pm1.9}$ | $0.791_{\pm0.036}$ | $72.2_{\pm2.9}$ | $0.734_{\pm0.064}$ | $75.0_{\pm5.0}$ | $0.751_{\pm0.075}$ |
| Above the normal range of p duration | $67.8_{\pm6.1}$ | $0.683_{\pm0.030}$ | $59.4_{\pm2.9}$ | $0.528_{\pm0.069}$ | $66.1_{\pm6.6}$ | $0.624_{\pm0.035}$ |
| Above the normal range of pr interval | $86.7_{\pm3.8}$ | $0.893_{\pm0.046}$ | $87.2_{\pm6.1}$ | $0.881_{\pm0.039}$ | $77.2_{\pm2.2}$ | $0.799_{\pm0.026}$ |
| Above the normal range of qrs duration | $86.7_{\pm5.0}$ | $0.901_{\pm0.073}$ | $81.1_{\pm2.9}$ | $0.782_{\pm0.018}$ | $80.6_{\pm5.8}$ | $0.848_{\pm0.066}$ |
| Above the normal range of qt corrected | $76.1_{\pm2.2}$ | $0.763_{\pm0.060}$ | $80.6_{\pm1.1}$ | $0.821_{\pm0.023}$ | $74.4_{\pm2.9}$ | $0.709_{\pm0.025}$ |
| Above the normal range of qt interval | $81.1_{\pm1.1}$ | $0.824_{\pm0.060}$ | $76.1_{\pm3.9}$ | $0.782_{\pm0.059}$ | $83.3_{\pm8.2}$ | $0.875_{\pm0.023}$ |
| Above the normal range of rr interval | $98.3_{\pm1.9}$ | $0.992_{\pm0.017}$ | $96.7_{\pm0.0}$ | $0.995_{\pm0.008}$ | $94.4_{\pm1.1}$ | $0.943_{\pm0.038}$ |
| Any diagnostic symptoms | $88.6_{\pm1.1}$ | $0.918_{\pm0.006}$ | $86.7_{\pm1.6}$ | $0.911_{\pm0.006}$ | $85.6_{\pm1.1}$ | $0.885_{\pm0.022}$ |
| Any form-related symptoms | $78.9_{\pm6.1}$ | $0.820_{\pm0.045}$ | $77.2_{\pm4.4}$ | $0.789_{\pm0.059}$ | $75.0_{\pm5.0}$ | $0.768_{\pm0.047}$ |
| Any kind of abnormal symptoms | $74.5_{\pm12.6}$ | $0.710_{\pm0.138}$ | $69.4_{\pm6.1}$ | $0.690_{\pm0.072}$ | $62.2_{\pm2.9}$ | $0.619_{\pm0.045}$ |
| Any kind of extra systoles | $92.0_{\pm3.9}$ | $0.932_{\pm0.047}$ | $84.0_{\pm3.9}$ | $0.859_{\pm0.089}$ | $78.7_{\pm2.6}$ | $0.743_{\pm0.036}$ |
| Any kind of noises | $68.0_{\pm1.5}$ | $0.612_{\pm0.018}$ | $68.9_{\pm0.9}$ | $0.645_{\pm0.017}$ | $70.6_{\pm1.8}$ | $0.652_{\pm0.024}$ |
| Any rhythm-related symptoms | $70.5_{\pm2.2}$ | $0.738_{\pm0.032}$ | $73.3_{\pm5.0}$ | $0.752_{\pm0.069}$ | $75.6_{\pm4.7}$ | $0.739_{\pm0.047}$ |
| Atrial fibrillation | $92.2_{\pm1.1}$ | $0.931_{\pm0.019}$ | $87.8_{\pm2.9}$ | $0.922_{\pm0.019}$ | $77.2_{\pm1.1}$ | $0.834_{\pm0.018}$ |
| Atrial flutter | $97.2_{\pm0.0}$ | $0.945_{\pm0.020}$ | $92.6_{\pm1.8}$ | $0.918_{\pm0.040}$ | $91.7_{\pm0.0}$ | $0.809_{\pm0.081}$ |
| Atrial premature complex | $82.8_{\pm5.8}$ | $0.861_{\pm0.048}$ | $75.0_{\pm1.9}$ | $0.777_{\pm0.059}$ | $68.3_{\pm3.3}$ | $0.657_{\pm0.025}$ |
| Baseline drift | $69.2_{\pm1.2}$ | $0.664_{\pm0.006}$ | $72.1_{\pm0.9}$ | $0.732_{\pm0.016}$ | $72.7_{\pm0.9}$ | $0.728_{\pm0.028}$ |
| Below the normal range of pr interval | $77.8_{\pm6.1}$ | $0.811_{\pm0.018}$ | $77.8_{\pm4.7}$ | $0.807_{\pm0.049}$ | $70.6_{\pm2.9}$ | $0.750_{\pm0.071}$ |
| Below the normal range of qrs duration | $88.9_{\pm1.1}$ | $0.876_{\pm0.014}$ | $85.0_{\pm3.8}$ | $0.885_{\pm0.035}$ | $76.7_{\pm3.3}$ | $0.787_{\pm0.045}$ |
| Below the normal range of qt corrected | $86.7_{\pm3.8}$ | $0.912_{\pm0.036}$ | $86.7_{\pm3.3}$ | $0.930_{\pm0.035}$ | $78.3_{\pm0.0}$ | $0.827_{\pm0.049}$ |
| Below the normal range of qt interval | $76.1_{\pm2.9}$ | $0.809_{\pm0.030}$ | $73.3_{\pm1.9}$ | $0.793_{\pm0.031}$ | $75.0_{\pm1.9}$ | $0.811_{\pm0.040}$ |
| Below the normal range of rr interval | $97.2_{\pm1.1}$ | $0.982_{\pm0.004}$ | $99.4_{\pm1.1}$ | $1.000_{\pm0.000}$ | $95.0_{\pm5.7}$ | $0.986_{\pm0.025}$ |
| Bigeminal pattern (unknown origin, supraventricular, or ventricular) | $88.0_{\pm2.3}$ | $0.862_{\pm0.021}$ | $85.3_{\pm5.2}$ | $0.827_{\pm0.044}$ | $81.3_{\pm4.7}$ | $0.660_{\pm0.076}$ |
| Burst noise | $66.6_{\pm0.2}$ | $0.523_{\pm0.033}$ | $66.9_{\pm0.0}$ | $0.522_{\pm0.022}$ | $66.7_{\pm0.1}$ | $0.515_{\pm0.025}$ |
| Complete left bundle branch block | $95.0_{\pm0.0}$ | $0.976_{\pm0.016}$ | $98.9_{\pm1.1}$ | $0.990_{\pm0.010}$ | $96.1_{\pm2.2}$ | $0.987_{\pm0.018}$ |
| Complete right bundle branch block | $97.3_{\pm1.1}$ | $0.984_{\pm0.026}$ | $97.3_{\pm1.1}$ | $0.907_{\pm0.134}$ | $95.6_{\pm2.1}$ | $0.966_{\pm0.030}$ |
| Conduction disturbance | $91.1_{\pm5.4}$ | $0.933_{\pm0.032}$ | $92.8_{\pm3.9}$ | $0.958_{\pm0.030}$ | $87.2_{\pm2.2}$ | $0.918_{\pm0.048}$ |
| Digitalis effect | $69.2_{\pm5.2}$ | $0.827_{\pm0.014}$ | $69.7_{\pm3.5}$ | $0.831_{\pm0.030}$ | $62.2_{\pm3.5}$ | $0.746_{\pm0.016}$ |
| Early stage of myocardial infarction | $86.3_{\pm1.9}$ | $0.877_{\pm0.024}$ | $84.2_{\pm1.1}$ | $0.911_{\pm0.044}$ | $76.3_{\pm1.9}$ | $0.809_{\pm0.072}$ |
| Electrodes problems | $84.3_{\pm1.0}$ | $0.599_{\pm0.051}$ | $84.3_{\pm1.0}$ | $0.686_{\pm0.033}$ | $83.3_{\pm1.7}$ | $0.615_{\pm0.029}$ |
| Extrasystoles | $92.2_{\pm2.9}$ | $0.935_{\pm0.020}$ | $91.1_{\pm1.1}$ | $0.947_{\pm0.021}$ | $87.8_{\pm4.4}$ | $0.923_{\pm0.026}$ |
| Extreme axis deviation | $90.1_{\pm3.0}$ | $0.913_{\pm0.028}$ | $87.7_{\pm2.0}$ | $0.913_{\pm0.014}$ | $87.1_{\pm3.0}$ | $0.903_{\pm0.033}$ |
| First degree av block | $83.6_{\pm2.7}$ | $0.861_{\pm0.029}$ | $84.1_{\pm1.8}$ | $0.827_{\pm0.073}$ | $73.0_{\pm0.0}$ | $0.722_{\pm0.104}$ |
| High qrs voltage | $79.3_{\pm0.0}$ | $0.602_{\pm0.082}$ | $79.1_{\pm0.5}$ | $0.583_{\pm0.041}$ | $79.3_{\pm0.0}$ | $0.562_{\pm0.015}$ |
| Hypertrophy | $72.8_{\pm2.9}$ | $0.765_{\pm0.027}$ | $75.6_{\pm5.8}$ | $0.767_{\pm0.062}$ | $73.9_{\pm2.9}$ | $0.777_{\pm0.052}$ |
| Incomplete left bundle branch block | $86.3_{\pm0.0}$ | $0.878_{\pm0.034}$ | $83.7_{\pm2.6}$ | $0.872_{\pm0.073}$ | $80.4_{\pm2.2}$ | $0.792_{\pm0.038}$ |
| Incomplete right bundle branch block | $69.6_{\pm2.8}$ | $0.808_{\pm0.024}$ | $71.5_{\pm2.5}$ | $0.815_{\pm0.006}$ | $64.7_{\pm2.5}$ | $0.808_{\pm0.046}$ |
| Inverted t-waves | $75.2_{\pm1.7}$ | $0.769_{\pm0.016}$ | $72.8_{\pm1.2}$ | $0.783_{\pm0.016}$ | $71.8_{\pm0.4}$ | $0.714_{\pm0.034}$ |
| Ischemic | $67.4_{\pm1.3}$ | $0.722_{\pm0.015}$ | $68.0_{\pm1.4}$ | $0.743_{\pm0.010}$ | $62.4_{\pm1.2}$ | $0.628_{\pm0.027}$ |
| Late stage of myocardial infarction | $83.7_{\pm2.3}$ | $0.777_{\pm0.046}$ | $84.4_{\pm1.3}$ | $0.790_{\pm0.070}$ | $83.0_{\pm3.5}$ | $0.836_{\pm0.108}$ |
| Left anterior fascicular block | $84.3_{\pm1.0}$ | $0.935_{\pm0.008}$ | $88.7_{\pm1.0}$ | $0.933_{\pm0.013}$ | $82.8_{\pm4.2}$ | $0.912_{\pm0.018}$ |
| Left atrial overload/enlargement | $52.1_{\pm6.7}$ | $0.618_{\pm0.036}$ | $54.2_{\pm1.6}$ | $0.642_{\pm0.019}$ | $52.1_{\pm1.6}$ | $0.592_{\pm0.041}$ |
| Left axis deviation | $82.8_{\pm5.8}$ | $0.902_{\pm0.050}$ | $86.1_{\pm4.7}$ | $0.909_{\pm0.029}$ | $77.2_{\pm4.7}$ | $0.839_{\pm0.033}$ |
| Left posterior fascicular block | $92.8_{\pm2.9}$ | $0.942_{\pm0.019}$ | $92.8_{\pm1.1}$ | $0.976_{\pm0.011}$ | $84.4_{\pm5.4}$ | $0.906_{\pm0.027}$ |
| Left ventricular hypertrophy | $62.1_{\pm4.3}$ | $0.734_{\pm0.025}$ | $62.9_{\pm2.2}$ | $0.774_{\pm0.038}$ | $56.7_{\pm2.2}$ | $0.758_{\pm0.033}$ |
| Long qt-interval | $78.8_{\pm2.2}$ | $0.562_{\pm0.138}$ | $78.2_{\pm2.5}$ | $0.736_{\pm0.074}$ | $76.9_{\pm0.0}$ | $0.681_{\pm0.068}$ |
| Low amplitude t-wave | $73.9_{\pm3.1}$ | $0.728_{\pm0.026}$ | $75.2_{\pm0.8}$ | $0.808_{\pm0.013}$ | $71.7_{\pm1.9}$ | $0.723_{\pm0.030}$ |
| Low qrs voltages in the frontal and horizontal leads | $69.8_{\pm0.9}$ | $0.650_{\pm0.056}$ | $73.0_{\pm4.7}$ | $0.683_{\pm0.128}$ | $68.5_{\pm1.0}$ | $0.574_{\pm0.051}$ |
| Middle stage of myocardial infarction | $75.0_{\pm1.9}$ | $0.817_{\pm0.020}$ | $80.0_{\pm1.9}$ | $0.864_{\pm0.014}$ | $71.1_{\pm10.7}$ | $0.786_{\pm0.083}$ |
| Myocardial infarction | $71.1_{\pm0.8}$ | $0.800_{\pm0.005}$ | $70.5_{\pm0.8}$ | $0.796_{\pm0.016}$ | $63.6_{\pm1.8}$ | $0.740_{\pm0.033}$ |
| Non-diagnostic t abnormalities | $70.7_{\pm1.6}$ | $0.778_{\pm0.040}$ | $68.2_{\pm1.4}$ | $0.795_{\pm0.024}$ | $68.2_{\pm1.4}$ | $0.735_{\pm0.066}$ |
| Non-specific intraventricular conduction disturbance (block) | $71.4_{\pm3.1}$ | $0.674_{\pm0.059}$ | $68.8_{\pm4.5}$ | $0.659_{\pm0.057}$ | $66.7_{\pm1.8}$ | $0.635_{\pm0.038}$ |
| Non-specific ischemic | $70.8_{\pm1.6}$ | $0.837_{\pm0.006}$ | $69.6_{\pm0.8}$ | $0.848_{\pm0.001}$ | $60.4_{\pm3.6}$ | $0.776_{\pm0.013}$ |
| Non-specific st changes | $76.9_{\pm0.6}$ | $0.683_{\pm0.025}$ | $74.1_{\pm2.6}$ | $0.713_{\pm0.022}$ | $75.5_{\pm1.4}$ | $0.662_{\pm0.059}$ |
| Non-specific st depression | $71.7_{\pm1.4}$ | $0.706_{\pm0.018}$ | $71.5_{\pm2.1}$ | $0.729_{\pm0.013}$ | $69.2_{\pm1.4}$ | $0.671_{\pm0.010}$ |
| Non-specific st elevation | $83.3_{\pm0.0}$ | $0.583_{\pm0.059}$ | $83.3_{\pm0.0}$ | $0.642_{\pm0.061}$ | $83.3_{\pm0.0}$ | $0.537_{\pm0.050}$ |
| Non-specific t-wave changes | $71.9_{\pm1.2}$ | $0.688_{\pm0.045}$ | $70.5_{\pm0.7}$ | $0.715_{\pm0.041}$ | $69.0_{\pm3.1}$ | $0.647_{\pm0.014}$ |
| Normal ecg | $72.2_{\pm4.7}$ | $0.819_{\pm0.021}$ | $68.9_{\pm7.6}$ | $0.752_{\pm0.047}$ | $70.0_{\pm5.0}$ | $0.748_{\pm0.046}$ |
| Normal functioning artificial pacemaker | $98.6_{\pm2.7}$ | $0.997_{\pm0.006}$ | $98.6_{\pm1.4}$ | $0.970_{\pm0.030}$ | $97.9_{\pm2.4}$ | $0.973_{\pm0.026}$ |
| Normal heart axis | $93.9_{\pm2.2}$ | $0.981_{\pm0.019}$ | $88.3_{\pm0.0}$ | $0.938_{\pm0.034}$ | $88.3_{\pm1.9}$ | $0.891_{\pm0.015}$ |
| Prolonged pr interval | $91.7_{\pm5.7}$ | $0.920_{\pm0.027}$ | $83.3_{\pm1.9}$ | $0.900_{\pm0.057}$ | $75.6_{\pm7.6}$ | $0.839_{\pm0.061}$ |
| Q waves present | $73.9_{\pm0.5}$ | $0.685_{\pm0.012}$ | $69.7_{\pm2.7}$ | $0.696_{\pm0.016}$ | $72.3_{\pm3.7}$ | $0.606_{\pm0.043}$ |
| Right atrial overload/enlargement | $82.2_{\pm2.3}$ | $0.764_{\pm0.075}$ | $78.7_{\pm3.0}$ | $0.817_{\pm0.020}$ | $72.4_{\pm5.2}$ | $0.758_{\pm0.017}$ |
| Right axis deviation | $91.1_{\pm2.9}$ | $0.957_{\pm0.018}$ | $89.4_{\pm2.2}$ | $0.928_{\pm0.024}$ | $88.9_{\pm2.2}$ | $0.937_{\pm0.011}$ |
| Sinus arrhythmia | $81.7_{\pm7.5}$ | $0.835_{\pm0.060}$ | $79.4_{\pm2.9}$ | $0.740_{\pm0.075}$ | $72.2_{\pm2.9}$ | $0.699_{\pm0.037}$ |
| Sinus bradycardia | $85.6_{\pm4.7}$ | $0.882_{\pm0.050}$ | $87.2_{\pm5.8}$ | $0.920_{\pm0.041}$ | $86.7_{\pm3.3}$ | $0.902_{\pm0.046}$ |
| Sinus rhythm | $71.1_{\pm2.9}$ | $0.786_{\pm0.035}$ | $76.7_{\pm5.7}$ | $0.852_{\pm0.017}$ | $73.9_{\pm4.7}$ | $0.814_{\pm0.049}$ |
| Sinus tachycardia | $96.1_{\pm2.2}$ | $0.987_{\pm0.000}$ | $95.0_{\pm1.9}$ | $0.969_{\pm0.023}$ | $94.4_{\pm1.1}$ | $0.962_{\pm0.002}$ |
| St/t change | $79.5_{\pm2.9}$ | $0.831_{\pm0.019}$ | $76.1_{\pm1.1}$ | $0.805_{\pm0.017}$ | $73.3_{\pm1.9}$ | $0.833_{\pm0.019}$ |
| Static noise | $66.7_{\pm0.5}$ | $0.580_{\pm0.010}$ | $66.5_{\pm1.1}$ | $0.625_{\pm0.007}$ | $67.9_{\pm0.8}$ | $0.647_{\pm0.004}$ |
| Subendocardial injury | $86.2_{\pm0.9}$ | $0.903_{\pm0.019}$ | $80.7_{\pm1.2}$ | $0.831_{\pm0.038}$ | $71.1_{\pm2.5}$ | $0.718_{\pm0.026}$ |
| Supraventricular extrasystoles | $85.0_{\pm1.9}$ | $0.845_{\pm0.015}$ | $84.4_{\pm5.8}$ | $0.877_{\pm0.026}$ | $74.4_{\pm3.9}$ | $0.741_{\pm0.042}$ |
| Supraventricular tachycardia | $94.4_{\pm0.0}$ | $0.876_{\pm0.069}$ | $92.6_{\pm7.3}$ | $0.893_{\pm0.137}$ | $83.3_{\pm0.0}$ | $0.598_{\pm0.204}$ |
| T-wave abnormality | $83.5_{\pm0.4}$ | $0.498_{\pm0.088}$ | $83.3_{\pm0.0}$ | $0.553_{\pm0.055}$ | $83.3_{\pm0.0}$ | $0.508_{\pm0.057}$ |
| Ventricular extrasystoles | $86.1_{\pm2.9}$ | $0.878_{\pm0.028}$ | $87.2_{\pm1.1}$ | $0.899_{\pm0.040}$ | $80.6_{\pm2.2}$ | $0.824_{\pm0.011}$ |
| Ventricular premature complex | $95.0_{\pm1.9}$ | $0.956_{\pm0.010}$ | $88.9_{\pm2.2}$ | $0.923_{\pm0.024}$ | $81.1_{\pm1.1}$ | $0.841_{\pm0.012}$ |
| Voltage criteria (qrs) for left ventricular hypertrophy | $69.6_{\pm1.5}$ | $0.655_{\pm0.029}$ | $71.5_{\pm2.0}$ | $0.729_{\pm0.031}$ | $68.5_{\pm3.2}$ | $0.674_{\pm0.038}$ |
| Within the normal range of p duration | $67.8_{\pm7.1}$ | $0.691_{\pm0.050}$ | $68.3_{\pm9.8}$ | $0.643_{\pm0.078}$ | $63.3_{\pm1.9}$ | $0.658_{\pm0.063}$ |
| Within the normal range of pr interval | $71.1_{\pm4.4}$ | $0.739_{\pm0.033}$ | $66.1_{\pm4.7}$ | $0.690_{\pm0.056}$ | $67.2_{\pm1.1}$ | $0.700_{\pm0.052}$ |
| Within the normal range of qrs duration | $76.7_{\pm5.0}$ | $0.792_{\pm0.029}$ | $71.7_{\pm8.6}$ | $0.769_{\pm0.102}$ | $74.4_{\pm3.9}$ | $0.781_{\pm0.046}$ |
| Within the normal range of qt corrected | $78.9_{\pm3.9}$ | $0.811_{\pm0.013}$ | $83.9_{\pm1.1}$ | $0.838_{\pm0.003}$ | $72.8_{\pm4.4}$ | $0.815_{\pm0.025}$ |
| Within the normal range of qt interval | $81.3_{\pm7.0}$ | $0.834_{\pm0.063}$ | $78.3_{\pm1.9}$ | $0.758_{\pm0.019}$ | $75.0_{\pm1.9}$ | $0.783_{\pm0.025}$ |
| Within the normal range of rr interval | $96.1_{\pm1.1}$ | $0.978_{\pm0.016}$ | $97.2_{\pm2.9}$ | $0.990_{\pm0.018}$ | $97.2_{\pm2.9}$ | $0.990_{\pm0.012}$ |

Table 16: Regular expressions for placeholders used for parsing lead positions of various form-related SCP codes. These placeholders are used to find lead statements from the ECG report, where the corresponding lead positions for each lead statement that can be parsed from the placeholders are described in Table 17.

| Placeholder | Regular Expression |
|---|---|
| [**lead_position**] | `v leads|chest|limb|anterior|antero\-?\s?lateral|antero\-?\s?septal|inferior|infero\-?\s?lateral|infero\-?\s?septal|lateral|high\-?\s?lateral|precordial|peripheral|standard|lateral\-?\s?chest` |
| [**lead**] | `(((v?\d)|(lead)?\s?(?<=[^A-Za-z])iii(?=[^A-Za-z])|(lead)?\s?(?<=[^A-Za-z])ii(?=[^A-Za-z])|(lead)?\s?(?<=[^A-Za-z])i(?=[^A-Za-z])|(lead)?\s?avr|(lead)?\s?avl|(lead)?\s?avf|v leads|all leads|chest( lead(s)?)?|limb( lead(s)?)?|anterior( lead(s)?)?|antero\-?\s?lateral( lead(s)?)?|antero\-?\s?septal( lead(s)?)?|inferior( lead(s)?)?|infero\-?\s?lateral( lead(s)?)?|infero\-?\s?septal( lead(s)?)?|lateral( lead(s)?)?|high\-?\s?lateral( lead(s)?)?|precordial( lead(s)?)?|peripheral( lead(s)?)?|standard( lead(s)?)?|infero\-lateral( lead(s)?)?|lateral\-?\s?chest( lead(s)?)?)\s?([/\),\.\-\&\s])?\s?(and)?\s?)+` |

Table 17: Corresponding lead positions for each lead statement.

| Statement | Lead Position |
|---|---|
| chest | V1, V2, V3, V4, V5, V6 |
| v leads | V1, V2, V3, V4, V5, V6 |
| limb | I, II, III, aVR, aVL, aVF |
| anterior | V3, V4 |
| antero-lateral | I, aVL, V3, V4, V5, V6 |
| antero-septal | V1, V2, V3, V4 |
| inferior | II, III, aVF |
| infero-lateral | I, II, III, aVL, aVF, V5, V6 |
| infero-septal | II, III, aVF, V1, V2 |
| lateral | I, aVL, V5, V6 |
| high-lateral | I, aVL |
| precordial | V1, V2, V3, V4, V5, V6 |
| peripheral | I, II, III, aVR, aVL, aVF |
| standard | I, II, III, aVR, aVL, aVF, V1, V2, V3, V4, V5, V6 |
| lateral-chest | I, aVL, V1, V2, V3, V4, V5, V6 |

Table 18: Regular expression parser for retrieving lead positions for each form-related SCP code. We provide example target statements that can be parsed from the corresponding regular expression. The placeholders such as [**lead**] or [**lead_position**] are described in Table 16.

| SCP Code | Example Target Statements | Regular Expression |
|---|---|---|
| NDT | t-change neg in [**lead**]
t wave flattening in [**lead**]
t waves flat in [**lead**]
t wave changes in [**lead**]
t wave inversion in [**lead**]
t waves are low or flat in [**lead**]
t waves are low in [**leads**]
t waves low in [**lead**]
t waves are inverted in [**lead**]
t waves inverted in [**lead**]
t waves are biphasic in [**lead**]
t abnormal in [**lead**]
t abnormality in [**lead**]
t flat in [**lead**]
flat t in [**lead**] | `(t wave(s)?|t waves are|t wve|t-change(s)?(:)?|`
`t change(s)?(:)?|flat t|biphasic t|neg t|high t`
`)\s*(are)?\s*(neg in|flattening in|flattening o`
`r slight inversion in|flat in|changes in|invers`
`ion in|low or flat in|generally low and are fla`
`t in|low in|flattened in|now generally flatter`
`and are slightly inverted in |inverted in|sligh`
`tly inverted in|low or flat in|flat or slightly`
`inverted in|now slightly inverted in|now inver`
`ted in|biphasic in|abnormal in|abnormality in|s`
`inus rhythm abnormal in|flat|flach in|in|biphas`
`.|neg\. in|biphasic|negative in|neg in|term. neg`
`in|high in|)\s*([**lead**])` |
| NST_ | st-t wave changes in [**lead**]
st-t changes in [**lead**]
st-t wave changes are more marked in [**lead**]
st-t wave changes persist in [**lead**]
t changes in [**lead**]
[**lead_position**] st-t changes
[**lead_position**] st-t wave changes | `((st-t wave changes|st-t changes|t changes)\s*(`
`are marked in|persist in|in|)\s*([**lead**]))|(`
`([**lead_position**])\s*(st-t wave changes|st-t`
`changes))` |
| DIG | digitalis t pointed in [**lead**]
digitalis change in [**lead**]
digitalis changes [**lead**]
digitalis change trough [**lead**]
digitalis change r trough [**lead**] | `(digitalis)\s*(t pointed in|change in|changes|c`
`hange trough|change r trough|in|)\s*([**lead**])` |
| STD_ | st segment depression in [**lead**]
st segments are depressed in [**lead**]
st depression in [**lead**]
st depression [**lead**]
st depression above [**lead**]
st lowering in [**lead**]
st-lowering [**lead**]
st reduction in [**lead**]
st reduction discrete in [**lead**]
st-senkung in [**lead**]
st-thinking in [**lead**] | `(st segment(s)?|st(\-)?st-senkung|st-thinking)\s`
`*(are)?\s*(depression in|depression and t wave f`
`lattening in|depressed in|depressed and t wave.{`
`0,15} in|depression|depression above|lowering in`
`|lowering|reduction in|reduction discrete in|in)`
`\s*([**lead**])` |
| VCLVH | voltages are high in [**lead**] suggesting lvh
[**lead**] voltages suggest possible lv hypertrophy
left ventricular hypertrophy are satisfied in [**lead**]
voltages in [**lead**] are at upper limit
voltages in [**lead**] of left ventricular hypertrophy | `(voltages are high in ([**lead**]) suggesting lv`
`h)|((([**lead**]) voltages suggest possible lv hy`
`pertrophy)|(left ventricular hypertrophy are sat`
`isfied in ([**lead**]))|(voltages in ([**lead**]`
`) are at upper limit)|(voltages in ([**lead**])`
`of left ventricular hypertrophy)|(r wave height`
`in ([**lead**]) suggests the possibility of left`
`ventricular hypertrophy)` |
| QWAVE | q waves in [**lead**]
q wave in [**lead**]
q wave present in [**lead**]
q in [**lead**]
q wave and small r wave in [**lead**]
q waves and t wave inversion in [**lead**]
[**lead_position**] q waves noted | `((q wave(s)?|q)\s*(are)?\s*(present in|in|(and|,`
`).{0,55} in)\s*([**lead**]))|(([**lead_position*`
`*]) q waves noted)` |
| LOWT | t waves are low in [**lead**]
t waves are low or flat in [**lead**] | `(t waves are)\s*(low or flat in|low in)\s*([**le`
`ad**])` |
| NT_ | t wave flattening in [**lead**]
t wave flattening persists in [**lead**]
t wave flattening or inversion in [**lead**]
t wave changes in [**lead**] | `(t wave)\s*(flattening in|flattening persists in`
`|flattening or slight inversion in|flattening or`
`inversion in|changes in)\s*([**lead**])` |
| INVT | t waves are inverted in [**lead**]
t wave inversion in [**lead**]
t waves inverted in [**lead**] | `(t wave(s)?)\s*(are)?\s*(inverted in|inversion i`
`n|flattening in.{0,30} and inverted in)\s*([**le`
`ad**]` |
| LVOLT | low limb lead voltage
peripheral low voltage
low voltage in [**lead**] | `(low limb lead voltage|peripheral low voltage|pe`
`ripheral low-voltage|peripheral low tension|low`
`voltage in ([**lead**]))` |
| HVOLT | high v lead voltages
voltages in [**lead**] are at upper limit
voltages are high in [**lead**] | `(high v lead voltages)|(voltages in ([**lead**])`
`\s*are at upper limit)|(voltages are high in ([*`
`*lead**]))` |

Table 18: Regular expression parser for retrieving lead positions for each form-related SCP code (Continued).

| SCP Code | Example Target Statements | Regular Expression |
|---|---|---|
| TAB_ | t changes negative in [**lead**]
t-changes in [**lead**]
t-changes [**lead**]
t-changes neg in [**lead**]
t-changes neg [**lead**]
t-changes neg. t in [**lead**]
t changes high in [**lead**]
t changes biphas. in [**lead**]
t abnormal in [**lead**] | `(t(\-)?\s?change(s)?(:)?\|t abnormal)\s*(negativ`
`e in\|neg in\|neg t\|neg\. t in\|neg\. in\|high in\|hi`
`gh t in\|flat in\|biphas\|biphas\.\|biphas\. in\|exc`
`essive\|in\|)\s*([**lead**])` |
| STE_ | st elevation in [**lead**]
st elevation discrete in [**lead**]
st elevation over [**lead**]
st elevation [**lead**]
st elevation discrete [**lead**]
st-hebung in [**lead**] | `(st\-?\s?elevation\|st-hebung)\s*(in\|discrete in`
`\|over\|discrete\|)\s*([**lead**])` |

Table 19: Full list of question templates.

| ID | Question Type | Attribute Type | Template |
|---|---|---|---|
| 1 | S-Verify | SCP code | Does this ECG show symptoms of ${scp_code}? |
| 2 | S-Verify | SCP code | Is this a normal ECG? |
| 3 | S-Choose | SCP code | Which symptom does this ECG show, ${scp_code1} or ${scp_code2}? |
| 4 | S-Choose | SCP code | Which diagnostic symptom does this ECG show, ${scp_code1} or ${scp_code2}, including uncertain symptoms? |
| 5 | S-Choose | SCP code | Which diagnostic symptom does this ECG show, ${scp_code1} or ${scp_code2}, excluding uncertain symptoms? |
| 6 | S-Choose | SCP code | Which form-related symptom does this ECG show, ${scp_code1} or ${scp_code2}? |
| 7 | S-Choose | SCP code | Which rhythm-related symptom does this ECG show, ${scp_code1} or ${scp_code2}? |
| 8 | S-Query | SCP code | What diagnostic symptoms does this ECG show, including uncertain symptoms? |
| 9 | S-Query | SCP code | What diagnostic symptoms does this ECG show, excluding uncertain symptoms? |
| 10 | S-Query | SCP code | What rhythm-related symptoms does this ECG show? |
| 11 | S-Query | SCP code | What form-related symptoms does this ECG show? |
| 12 | S-Verify | SCP code | Does this ECG show any kind of abnormal symptoms? |
| 13 | S-Verify | SCP code | Does this ECG show any diagnostic symptoms, including uncertain symptoms? |
| 14 | S-Verify | SCP code | Does this ECG show any diagnostic symptoms, excluding uncertain symptoms? |
| 15 | S-Verify | SCP code | Does this ECG show any form-related symptoms? |
| 16 | S-Verify | SCP code | Does this ECG show any rhythm-related symptoms? |
| 17 | S-Verify | SCP code | Does this ECG show symptoms of ${scp_code} in ${lead}? |
| 18 | S-Query | SCP code | What form-related symptoms does this ECG show in ${lead}? |
| 19 | S-Query | SCP code | What leads are showing symptoms of ${scp_code} in this ECG? |
| 20 | S-Verify | Heart axis | Does this ECG show ${heart_axis}? |
| 21 | S-Choose | Heart axis | Which cardiac axis does this ECG show, ${heart_axis1} or ${heart_axis2}? |
| 22 | S-Query | Heart axis | What direction is this ECG deviated to? |
| 23 | S-Verify | Stage of infarction | Does this ECG show ${stage_of_infarction}? |
| 24 | S-Choose | Stage of infarction | Which stage of infarction is this ECG at, ${stage_of_infarction1} or ${stage_of_infarction2}? |
| 25 | S-Query | Stage of infarction | What stage of infarction is this ECG at? |
| 26 | S-Verify | Noise | Does this ECG show ${noise} in ${lead}? |
| 27 | S-Verify | Noise | Does this ECG show ${noise}? |
| 28 | S-Verify | Noise | Does th is ECG show any kind of noises in ${lead}? |
| 29 | S-Verify | Noise | Does this ECG show any kind of noises? |
| 30 | S-Choose | Noise | Which noise does this ECG show, ${noise1} or ${noise2}? |
| 31 | S-Choose | Noise | Which noise does this ECG show in ${lead}, ${noise1} or ${noise2}? |
| 32 | S-Query | Noise | What kind of noises does this ECG show? |
| 33 | S-Query | Noise | What kind of noises does this ECG show in ${lead}? |
| 34 | S-Query | Noise | What leads are showing ${noise} in this ECG? |
| 35 | S-Verify | Extra systole | Does this ECG show any kind of extra systoles? |
| 36 | S-Verify | Extra systole | Does this ECG show ${extra_systole}? |
| 37 | S-Choose | Extra systole | Which kind of extra systoles does this ECG show, ${extra_systole1} or ${extra_systole2}? |
| 38 | S-Query | Extra systole | What kind of extra systole does this ECG show? |
| 39 | S-Verify | Numeric feature | Does the ${numeric_feature} of this ECG fall ${numeric_range}? |
| 40 | S-Choose | Numeric feature | Which range does the ${numeric_feature} of this ECG fall in, ${numeric_range1} or ${numeric_range2}? |
| 41 | S-Query | Numeric feature | What range does the ${numeric_feature} of this ECG fall in? |
| 42 | S-Query | Numeric feature | What numeric features of this ECG fall ${numeric_range}? |
| 43 | CC-Verify | SCP code | Compared to the previous tracing, has ${scp_code} been resolved in the recent tracing? |
| 44 | CC-Verify | SCP code | Compared to the previous tracing, has ${scp_code} been newly detected in the recent tracing? |
| 45 | CC-Verify | SCP code | Compared to the previous tracing, does ${scp_code} still remain in the recent tracing? |
| 46 | CC-Verify | SCP code | Compared to the previous tracing, is ${scp_code} still not found in the recent tracing? |
| 47 | CC-Query | SCP code | What symptoms have been resolved in the recent tracing as compared to the previous one? |
| 48 | CC-Query | SCP code | What symptoms are newly detected in the recent tracing as compared to the previous one? |

Table 19: Full list of question templates (Continued).

| ID | Question Type | Attribute Type | Template |
|---|---|---|---|
| 49 | CC-Query | SCP code | What symptoms still remain in the recent tracing as compared to the previous one? |
| 50 | CC-Verify | Numeric feature | Compared to the previous tracing, has the ${numeric_feature} of the recent tracing become normal? |
| 51 | CC-Verify | Numeric feature | Compared to the previous tracing, has the ${numeric_feature} of the recent tracing changed to an abnormal value? |
| 52 | CC-Verify | Numeric feature | Compared to the previous tracing, does the ${numeric_feature} still show an abnormal value in the recent tracing? |
| 53 | CC-Verify | Numeric feature | Compared to the previous tracing, is the ${numeric_feature} still considered normal in the recent tracing? |
| 54 | CC-Query | Numeric feature | What numeric features of the recent tracing now have become normal compared to the previous one? |
| 55 | CC-Query | Numeric feature | What numeric features of the recent tracing are now considered abnormal values compared to the previous one? |
| 56 | CC-Query | Numeric feature | What numeric features of the recent tracing are still abnormal compared to the previous one? |
| 57 | CI-Verify | SCP code | Compared to the first ECG, has ${scp_code} been resolved in the second ECG? |
| 58 | CI-Verify | SCP code | Compared to the first ECG, has ${scp_code} been newly detected in the second ECG? |
| 59 | CI-Verify | SCP code | Compared to the first ECG, does ${scp_code} still remain in the second ECG? |
| 60 | CI-Verify | SCP code | Compared to the first ECG, is ${scp_code} still not found in the second ECG? |
| 61 | CI-Query | SCP code | What symptoms have been resolved in the second ECG as compared to the first ECG? |
| 62 | CI-Query | SCP code | What symptoms are newly detected in the second ECG as compared to the first ECG? |
| 63 | CI-Query | SCP code | What symptoms still remain in the second ECG as compared to the first ECG? |
| 64 | CI-Verify | Numeric feature | Compared to the first ECG, has the ${numeric_feature} of the second ECG become normal? |
| 65 | CI-Verify | Numeric feature | Compared to the first ECG, has the ${numeirc_feature} of the second ECG changed to an abnormal value? |
| 66 | CI-Verify | Numeric feature | Compared to the first ECG, does the ${numeric_feature} still show an abnormal value in the second ECG? |
| 67 | CI-Verify | Numeric feature | Compared to the first ECG, is the ${numeric_feature} still normal in the second ECG? |
| 68 | CI-Query | Numeric feature | What numeric features of the second ECG now have become normal as compared to the first ECG? |
| 69 | CI-Query | Numeric feature | What numeric features of the second ECG are now considered abnormal values as compared to the first ECG? |
| 70 | CI-Query | Numeric feature | What numeric features of the second ECG are still abnormal as compared to the first ECG? |

Table 20: Full list of paraphrases for each question template (1/10).

| ID | Train & validation paraphrases | Test paraphrases |
|---|---|---|
| 1 | Is ${scp_code} present in this ECG?
Does this ECG reveal any signs of ${scp_code}?
Is there evidence of ${scp_code} on this ECG?
Are there any manifestations of ${scp_code} on this ECG?
Can the presence of ${scp_code} be identified from this ECG?
Is ${scp_code} detectable from this ECG?
Is ${scp_code} indicated by this ECG? | Are there any indications of ${scp_code} in this ECG?
Can ${scp_code} be detected from this ECG?
Does this ECG display any features of ${scp_code}? |
| 2 | Does this ECG display normal results?
Is the ECG result within the standard range of a healthy person?
Is this ECG pattern indicative of normal cardiac function?
Is the ECG tracing normal?
Does the ECG result suggest a healthy heart function?
Is this ECG tracing considered normal in medical terms?
Does the ECG report indicate normal cardiac activity? | Is this electrocardiogram (ECG) reading within normal parameters?
Is the ECG reading typical for a healthy individual?
Are the results of this ECG normal? |
| 3 | What are the symptoms suggested by this ECG, ${scp_code1} or ${scp_code2}?
Which symptom does this ECG illustrate, ${scp_code1} or ${scp_code2}?
What is the symptom displayed in this ECG, ${scp_code1} or ${scp_code2}?
Which one does this ECG point to, ${scp_code1} or ${scp_code2} as a symptom?
Which of the following symptoms is displayed by this ECG, ${scp_code1} or ${scp_code2}?
What symptoms are evident from this ECG, ${scp_code1} or ${scp_code2}?
Which one can this ECG be used to diagnose, ${scp_code1} or ${scp_code2}? | Which one is this ECG indicating, ${scp_code1} or ${scp_code2} symptoms?
What are the symptoms on this ECG, ${scp_code1} or ${scp_code2}?
Which symptom does this ECG show, ${scp_code1} or ${scp_code2}? |
| 4 | Can you identify which diagnostic symptom is indicated in this ECG, whether it is ${scp_code1} or ${scp_code2}, even if there are symptoms that are not entirely clear?
Regarding this ECG, which diagnostic symptom is being exhibited, specifically ${scp_code1} or ${scp_code2}, including any ambiguous symptoms?
What is the diagnostic symptom displayed in this ECG, ${scp_code1} or ${scp_code2}, even if there are uncertain symptoms?
Could you identify the diagnostic symptom depicted in this ECG, specifically ${scp_code1} or ${scp_code2}, including any indeterminate symptoms?
What diagnostic symptom is being manifested in this ECG, ${scp_code1} or ${scp_code2}, even though there are unclear symptoms?
Regarding this ECG, can you identify which diagnostic symptom is being shown, whether it is ${scp_code1} or ${scp_code2}, including symptoms that are uncertain?
What is the diagnostic symptom indicated by this ECG, specifically ${scp_code1} or ${scp_code2}, including any uncertain symptoms? | In reference to this ECG, can you determine which diagnostic symptom is being shown, ${scp_code1} or ${scp_code2}, even if they are not entirely clear?
Which diagnostic symptom is being demonstrated by this ECG, ${scp_code1} or ${scp_code2}, even if there are symptoms that are uncertain?
Can you indicate which diagnostic symptom is being demonstrated in this ECG, ${scp_code1} or ${scp_code2}, including unclear symptoms? |
| 5 | Which diagnostic symptom does this ECG show, ${scp_code1} or ${scp_code2}, excluding uncertain symptoms?
What is the diagnostic symptom that can be identified from this ECG, excluding any symptoms that are unclear, ${scp_code1} or ${scp_code2}?
By excluding uncertain symptoms, which diagnostic symptom is apparent in this ECG, ${scp_code1} or ${scp_code2}?
Excluding any uncertain symptoms, what is the diagnostic symptom that this ECG displays, ${scp_code1} or ${scp_code2}?
Which diagnostic symptom is clearly indicated in this ECG, ${scp_code1} or ${scp_code2}, with ambiguous symptoms removed from consideration?
What is the diagnostic symptom that can be identified from this ECG, ${scp_code1} or ${scp_code2}, when we ignore any symptoms that are uncertain?
Excluding ambiguous symptoms, which diagnostic symptom can be identified from this ECG, ${scp_code1} or ${scp_code2}? | Can you determine from this ECG which diagnostic symptom is present, ${scp_code1} or ${scp_code2}, without including ambiguous symptoms?
Which diagnostic symptom is evident in this ECG, ${scp_code1} or ${scp_code2}, but only if we discount ambiguous symptoms?
Can you determine which diagnostic symptom is present in this ECG, ${scp_code1} or ${scp_code2}, by excluding any symptoms that are unclear? |
| 6 | Which one is the ECG indicating, ${scp_code1} or ${scp_code2} as a form-related symptom?
Which type of form-related symptom is illustrated on this ECG, ${scp_code1} or ${scp_code2}?
Which one of ${scp_code1} or ${scp_code2} is indicated as a form-related symptom on this ECG?
What is the form-related symptom displayed on this ECG, ${scp_code1} or ${scp_code2}
What form-related symptom is shown on this ECG, ${scp_code1} or ${scp_code2}?
What is the form-related symptom present on this ECG, specifically ${scp_code1} or ${scp_code2}?
What form-related symptom is being presented on this ECG, ${scp_code1} or ${scp_code2}? | Which one does this ECG display, ${scp_code1} or ${scp_code2} in terms of form-related symptoms?
Which of the form-related symptoms, ${scp_code1} or ${scp_code2} is evident on this ECG?
What symptoms does this ECG show, ${scp_code1} or ${scp_code2} as a symptom related to form? |

Table 20: Full list of paraphrases for each question template (2/10).

| ID | Train & validation paraphrases | Test paraphrases |
|---|---|---|
| 7 | Which one is the ECG indicating, ${scp_code1} or ${scp_code2} as a rhythm-related symptom? What does the ECG demonstrate, ${scp_code1} or ${scp_code2} as a rhythm-related feature? What is the rhythm-related symptom displayed by the ECG, ${scp_code1} or ${scp_code2}? Which rhythm-related symptom is highlighted in this ECG, ${scp_code1} or ${scp_code2}? Which rhythm-related element is being presented on this ECG, ${scp_code1} or ${scp_code2}? Which one of ${scp_code1} or ${scp_code2} is being indicated as a rhythm-related issue in this ECG? What rhythm-related symptom can be inferred from this ECG, ${scp_code1} or ${scp_code2}? | Which one does this ECG display, ${scp_code1} or ${scp_code2} as a rhythm-based manifestation? What is the rhythm-related issue displayed on this ECG, ${scp_code1} or ${scp_code2}? Which rhythm-related characteristic is shown by the ECG, ${scp_code1} or ${scp_code2}? |
| 8 | Can you point out the diagnostic symptoms shown in this ECG, including those that are uncertain? Could you explain the diagnostic symptoms in this ECG, both certain and uncertain? Can you highlight the diagnostic indications seen in this ECG, including those that are not conclusive? Can you specify the diagnostic symptoms shown in this ECG, including those that are inconclusive? What are the diagnostic features of this ECG, including the uncertain ones? Could you elaborate on the diagnostic symptoms present in this ECG, including those that are uncertain? What does this ECG reveal in terms of diagnostic symptoms, both certain and questionable? | Could you identify the diagnostic symptoms of this ECG, including those that are unclear? What are the diagnostic indications that this ECG presents, including those that are not definitive? What are the symptoms of diagnosis present in this ECG, including those that are uncertain? |
| 9 | Can you identify the diagnostic symptoms present in this ECG, disregarding any uncertain symptoms? Which symptoms in this ECG are considered diagnostic, leaving out any symptoms that are uncertain? What are the distinguishable diagnostic symptoms displayed in this ECG, leaving aside any uncertain symptoms? Could you point out the certain diagnostic symptoms seen in this ECG, omitting any uncertain symptoms? Excluding any uncertain symptoms, what are the diagnostic features evident in this ECG? Can you identify the diagnostic indicators present in this ECG, leaving out any uncertain symptoms? What are the evident diagnostic features in this ECG, disregarding any uncertain symptoms? | What are the specific diagnostic symptoms shown in this ECG, excluding any uncertain symptoms? What are the conclusive diagnostic symptoms exhibited in this ECG, excluding any uncertain symptoms? Excluding any uncertain symptoms, what are the definitive diagnostic findings in this ECG? |
| 10 | Which rhythm-based indications are evident in this ECG? What are the rhythmical indications that can be observed on this ECG? What signs of a rhythm-related disorder can be found in this ECG recording? What rhythmical symptoms are displayed on this ECG? What indications of a rhythm-related issue can be found in this ECG? What rhythm-based signs are present in this ECG? What are the rhythm-related indications that can be seen on this ECG? | What are the rhythm-based symptoms displayed in this ECG? What are the rhythm-related indications exhibited by this ECG? Which rhythm-based symptoms are evident on this ECG reading? |
| 11 | Can you identify the form-related symptoms on this ECG? What are the form-related indicators displayed on this ECG? Could you point out the ECG symptoms that are related to form? What are the form-related features displayed on this ECG? Which form-related abnormalities can be observed in this ECG? Which symptoms on this ECG are associated with form-related problems? Which form-related characteristics are visible in this ECG? | What form-related symptoms are evident in this ECG? What does the ECG reveal in terms of form-related symptoms? What are the form-related manifestations apparent in this ECG? |
| 12 | Is there anything unusual in this ECG? Are there any abnormalities evident in the ECG? Are there any abnormal findings on the ECG results? Are there any indications of irregular cardiac activity on the ECG? Are there any signs of abnormalities in the ECG tracing? Are there any atypical features in the ECG? Does this ECG show any indications of abnormal cardiac activity? | Are there any signs of abnormalities on this ECG? Is there any indication of irregularities on this ECG? Does the ECG display any atypical patterns or symptoms? |
| 13 | Are there any discernible diagnostic indications, including ambiguous ones, in this ECG? Does this ECG exhibit any indicative symptoms, whether or not they are ambiguous? Are there any signs or symptoms that may aid in the diagnosis, even if they are unclear, in this ECG? Does this ECG show any potential diagnostic symptoms, including uncertain ones? Are there any identifiable diagnostic signs, even if they are uncertain, in this ECG? Are there any diagnostic symptoms, including those that are indeterminate, in this ECG? Are there any observable diagnostic indications or symptoms, even if they are uncertain, in this ECG? | Does this ECG display any diagnostic clues, including uncertain ones? Are there any observable diagnostic features, even if they are uncertain, in this ECG? Are there any signs or symptoms that may point to a diagnosis, even if they are unclear, in this ECG? |

Table 20: Full list of paraphrases for each question template (3/10).

| ID | Train & validation paraphrases | Test paraphrases |
|---|---|---|
| 14 | Can this ECG provide any conclusive evidence of diagnosis, except for symptoms that are uncertain?
Is there any evident diagnostic manifestation in this ECG, with uncertain symptoms being excluded?
Can this ECG exhibit any decisive diagnostic symptoms, with ambiguous symptoms being ruled out?
Is there any conclusive diagnostic evidence in this ECG, with uncertain symptoms being eliminated?
Does this ECG present any unmistakable diagnostic signs, leaving out any uncertain symptoms?
Is there any irrefutable diagnostic indication in this ECG, without including any unclear symptoms?
Are there any conspicuous diagnostic features displayed in this ECG, disregarding any uncertain symptoms? | Are there any clear diagnostic symptoms present in this ECG, with uncertain symptoms being excluded?
Are there any unmistakable diagnostic symptoms visible in this ECG, disregarding any unclear symptoms?
Are there any noticeable diagnostic characteristics in this ECG, with ambiguous symptoms being excluded? |
| 15 | Can any form-related symptoms be detected in this ECG?
Are there any observable form-related symptoms in this ECG?
Is there evidence of any form-related symptoms in this ECG?
Does this ECG exhibit any form-related symptoms?
Are there any apparent form-related symptoms in this ECG?
Are there any observable form-related issues in this ECG result?
Do the results of this ECG suggest any form-related symptoms? | Are there any signs of form-related symptoms in this ECG?
Does this ECG reveal any form-related symptoms?
Is there any indication of form-related issues in this ECG reading? |
| 16 | Are there any signs of rhythm-related symptoms on this ECG?
Is there evidence of a rhythm-related condition in this ECG?
Are there any rhythm-related symptoms displayed on this ECG?
Does the ECG suggest any rhythm-related symptoms?
Are there any rhythm-related signs apparent on this ECG?
Can any rhythm-related symptoms be identified on this ECG?
Does this ECG point towards any rhythm-related symptoms? | Is there any indication of a rhythm-related condition on this ECG?
Are there any signs of a rhythm-related ailment in this ECG?
Is there any evidence of rhythm-related symptoms shown in this ECG? |
| 17 | Is ${lead} showing indications of ${scp_code} in this ECG?
Are there any signs of ${scp_code} present in $lead on this ECG?
Can ${lead} provide evidence of ${scp_code} in this ECG?
Are there any indications of ${scp_code} in ${lead} on this ECG reading?
Is there any suggestion of ${scp_code} in ${lead} on this ECG?
Is there any evidence of ${scp_code} in ${lead} on this ECG recording?
Is there any indication of ${scp_code} in ${lead} on this ECG tracing? | Is ${scp_code} identifiable in ${lead} on this ECG?
Is there evidence of ${scp_code} in ${lead} on this ECG?
Does this ECG reveal symptoms of ${scp_code} in ${lead}? |
| 18 | What are the signs related to the form that this ECG shows in ${lead}?
What form-related traits are exhibited by this ECG in ${lead}?
In ${lead}, what signs of form does this ECG indicate?
What are the form-related indicators that are manifested in ${lead} in this ECG?
What are the ECG form-related indications that are noticeable in ${lead}?
What form-related manifestations are apparent in ${lead} in this ECG?
What are the ECG form-related symptoms displayed in ${lead}? | In ${lead}, what are the symptoms related to the form that is shown by this ECG?
In ${lead}, what form-related features does this ECG display?
Could you identify the form-related features shown by this ECG in ${lead}? |
| 19 | What specific leads are showing any indications of ${scp_code} on this ECG?
Which leads on this ECG are exhibiting symptoms of ${scp_code}?
Which leads in this ECG display characteristics of ${scp_code} symptoms?
In what leads of this ECG do we see signs of ${scp_code}?
What are the leads that demonstrate symptoms of ${scp_code} in this ECG?
Which leads of this ECG exhibit ${scp_code} symptoms?
Can you tell me which leads in this ECG show features of ${scp_code} symptoms? | Can you identify which leads in this ECG display signs of ${scp_code} symptoms?
Can you point out the leads in this ECG that show manifestations of ${scp_code}?
In which leads of this ECG can we identify characteristics of ${scp_code}? |
| 20 | Is the ${heart_axis} discernible in this ECG?
Is there evidence of ${heart_axis} in this ECG?
Does the ECG tracing suggest ${heart_axis}?
Can you identify the ${heart_axis} from the ECG tracing?
Does this ECG show ${heart_axis}?
Is there any indication of ${heart_axis} in this ECG?
Is there any suggestion of ${heart_axis} in this ECG? | From the ECG, can we infer the ${heart_axis}?
Does the ECG exhibit any indications of ${heart_axis}?
Does the ECG display any patterns indicative of ${heart_axis}? |
| 21 | What is the cardiac axis depicted in this ECG pointing towards, ${heart_axis1} or ${heart_axis2}?
Which cardiac axis does this ECG illustrate, ${heart_axis1} or ${heart_axis2}?
Which direction is the cardiac axis pointing in this ECG, ${heart_axis1} or ${heart_axis2}?
What is the cardiac axis indicated in this ECG directed towards, ${heart_axis1} or ${heart_axis2}?
Which of the two directions, ${heart_axis1} or ${heart_axis2}, does the cardiac axis on this ECG represent?
What does the cardiac axis displayed in this ECG correspond to, ${heart_axis1} or ${heart_axis2}?
On this ECG, which direction is the cardiac axis positioned in, ${heart_axis1} or ${heart_axis2}? | What is the cardiac axis orientation consistent with this ECG, ${heart_axis1} or ${heart_axis2}?
Which direction of the cardiac axis does this ECG tracing exhibit, ${heart_axis1} or ${heart_axis2}?
In this ECG, is the cardiac axis directed towards ${heart_axis1} or ${heart_axis2}? |

Continued on next page.

Table 20: Full list of paraphrases for each question template (4/10).

| ID | Train & validation paraphrases | Test paraphrases |
|----|--------------------------------|------------------|
| 22 | To what direction is this ECG showing a deviation?
What is the deviation direction of this ECG?
Which direction is indicated by the deviation in this ECG?
To which side is this ECG deviating?
What is the direction of the deviation illustrated in this ECG?
What is the direction of the deviation in the ECG tracing?
In which direction is the ECG recording deviating from the norm? | What is the direction of deviation shown by this ECG?
What direction is the deviation in this ECG taking place?
What direction is this ECG recording deviating towards? |
| 23 | Is ${stage_of_infarction} present based on this ECG result?
Can the presence of ${stage_of_infarction} be confirmed through this ECG?
Is there any indication of ${stage_of_infarction} from this ECG recording?
Does this ECG suggest the occurrence of ${stage_of_infarction}?
Is ${stage_of_infarction} identifiable by this ECG analysis?
Can this ECG be used to diagnose ${stage_of_infarction}?
Is there any possibility of ${stage_of_infarction} based on this ECG reading? | Can ${stage_of_infarction} be identified through this ECG reading?
Does this ECG reveal signs of ${stage_of_infarction}?
Does this ECG indicate the presence of ${stage_of_infarction}? |
| 24 | At what stage of infarction is this ECG, ${stage_of_infarction1} or ${stage_of_infarction2}?
Which infarction stage is this ECG displaying, ${stage_of_infarction1} or ${stage_of_infarction2}?
At which stage of infarction is this ECG at, ${stage_of_infarction1} or ${stage_of_infarction2}?
Is this ECG at ${stage_of_infarction1} or ${stage_of_infarction2}?
Which stage of myocardial infarction is this ECG at, ${stage_of_infarction1} or ${stage_of_infarction2}?
Which stage of infarction is represented by this ECG, ${stage_of_infarction1} or ${stage_of_infarction2}?
Is this ECG showing ${stage_of_infarction1} or ${stage_of_infarction2}? | Does this ECG indicate ${stage_of_infarction1} or ${stage_of_infarction2}?
At what stage of myocardial infarction is this ECG, ${stage_of_infarction1} or ${stage_of_infarction2}?
Which one does this ECG correspond to, ${stage_of_infarction1} or ${stage_of_infarction2}? |
| 25 | Can you determine the infarction stage from this ECG?
Which stage of myocardial infarction is indicated by this ECG?
What is the infarction stage depicted in this ECG?
Which phase of infarction is this ECG reflecting?
At what point of infarction is this ECG showing?
From this ECG, what can you infer about the stage of infarction?
What is the infarction phase that this ECG corresponds to? | What is the phase of infarction represented in this ECG?
Can you recognize the stage of infarction from this ECG tracing?
What is the stage of myocardial infarction that can be deduced from this ECG? |
| 26 | Does ${lead} show any ${noise} on this ECG?
Is there ${noise} in ${lead} as displayed in this ECG?
Is there any ${noise} in ${lead} according to this ECG?
Does the ECG reveal any ${noise} in ${lead}?
Is ${noise} present in ${lead} based on the ECG?
Can ${noise} be detected in ${lead} based no the ECG results?
Does the ECG demonstrate any ${noise} in this ${lead}? | Does ${lead} exhibit any ${noise} on this ECG?
Is ${lead} showing any ${noise} on this ECG?
Can ${noise} be observed in ${lead} on this ECG? |
| 27 | Is this ECG reading being affected by any ${noise}?
Is ${noise} visible in this ECG?
Is there any indication of ${noise} on this ECG?
Is there any ${noise} present on this ECG?
Is the ECG recording showing any signs of ${noise} interference?
Is ${noise} disrupting the ECG reading?
Is there any interference from ${noise} in this ECG recording? | Are there any signs of ${noise} in this ECG reading?
Is this ECG affected by ${noise}?
Does this ECG exhibit any ${noise}? |
| 28 | Is there any interference present in ${lead} on this ECG?
Is there any interference or distortion present in ${lead} on this ECG?
Does ${lead} on this ECG have any noises?
Is ${lead} on this ECG showing any signs of interference or distortion?
Does ${lead} on this ECG have any noises that can be detected?
Does this ECG show any kind of signal distortions in ${lead}?
Are there any noises detected in ${lead} on this ECG? | Are there any abnormal noises detected in ${lead} on this ECG?
Are there any disruptive noises present in ${lead} on this ECG?
Is ${lead} on this ECG showing any indication of interference or noises? |
| 29 | Are there any disturbances present in this ECG reading?
Is there any interference on this ECG?
Can you detect any noise in this ECG?
Does the ECG display any noise or disturbance?
Are there any distortions visible on this ECG?
Is the ECG recording affected by any kind of interference or noise?
Does the ECG signal contain any kind of disturbances? | Is there any noise in this ECG reading?
Is there any kind of distortion or interference present in this ECG?
Does the ECG exhibit any signs of interference or noise? |
| 30 | Which type of noise is present in this ECG, ${noise1} or ${noise2}?
Which one of the ${noise1} or ${noise2} is evident in this ECG?
What is causing disturbance in this ECG, ${noise1} or ${noise2}?
What does this ECG contain as a signal disturbance, ${noise1} or ${noise2}?
What type of interference is visible in this ECG, ${noise1} or ${noise2}?
Which noise source is present in this ECG, ${noise1} or ${noise2}?
Which type of noise is affecting the ECG signal, ${noise1} or ${noise2}? | Can you identify whether the ECG displays ${noise1} or ${noise2} interference?
Which noise is causing a disruption in this ECG, ${noise1} or ${noise2}?
What type of noise is present in this ECG, ${noise1} or ${noise2}? |
| 31 | What does ${lead} of the ECG display, ${noise1} or ${noise2}?
What does ${lead} of the ECG exhibit, ${noise1} or ${noise2}?
Which type of noise, ${noise1} or ${noise2}, is present in ${lead} on this ECG?
Is ${lead} of the ECG affected by ${noise1} or ${noise2}?
Which interferences does ${lead} of the ECG contain, ${noise1} or ${noise2}?
Which one of ${noise1} or ${noise2} is detected in ${lead} on the ECG?
Is ${noise1} or ${noise2} present in ${lead} of the ECG? | Is ${lead} of the ECG showing ${noise1} or ${noise2}?
What noises can be observed in ${lead} of the ECG, ${noise1} or ${noise2}?
Which type of noise, ${noise1} or ${noise2}, is visible in ${lead} of the ECG? |
| 32 | What are the noises represented in this ECG?
What noises are being displayed in this ECG?
What noises are being registered by this ECG?
What noises are visible in this ECG?
What kind of noise is being captured by this ECG reading?
What kind of noise can be seen in this ECG reading?
What is the type of noise that is shown on this ECG? | What types of noise are indicated by this ECG reading?
What kind of noise can be observed in this ECG?
What kind of noise is visible in this ECG? |
| 33 | Which noises are evident in ${lead} on this ECG recording?
What type of noises are displayed in ${lead} in this ECG waveform?
What kind of noises are present in ${lead} on this ECG tracing?
What kind of noises does this ECG show in ${lead}?
What type of noise can be seen on this ECG in ${lead}?
What types of noise can be observed in ${lead} on this ECG?
What noise is present in ${lead} on this ECG? | In ${lead}, what are the noises that can be identified in this ECG?
Which noises are shown in ${lead} on this ECG recording?
What is the noise pattern that is visible in ${lead} on this ECG? |

Continued on next page.

Table 20: Full list of paraphrases for each question template (5/10).

| ID | Train & validation paraphrases | Test paraphrases |
|---|---|---|
| 34 | Which ECG leads are affected by ${noise}? Which ECG leads are demonstrating ${noise} on the tracing? What are the leads on the ECG that are manifesting ${noise}? Which ECG leads are exhibiting ${noise} that is noticeable? What are the leads on the ECG where ${noise} is present? Which leads on the ECG tracing are showing ${noise} that needs to be addressed? What specific leads on the ECG are affected by ${noise}? | In this ECG, which leads are exhibiting ${noise}? What leads are causing ${noise} in this ECG? Which leads on the ECG are producing ${noise}? |
| 35 | Are there any extra systoles present on this ECG? Is there any indication of extra systoles on this ECG recording? Does this ECG display evidence of any extra systoles? Can any extra systoles be observed from this ECG result? Does this ECG show any evidence of extra heart rhythms? Are there any indications of extra beats on this ECG? Does this ECG reveal any evidence of extra contractions? | Is there any indication of extra systoles on this ECG? Is there any indication of extra heartbeats on this ECG? Can any extra heart contractions be detected from this ECG? |
| 36 | Does this ECG display any signs of ${extra_systole}? Is there evidence of ${extra_systole} in this ECG? Is ${extra_systole} detectable from this ECG trace? Does this ECG indicate the presence of ${extra_systole}? Is there any indication of ${extra_systole} in this ECG? Can ${extra_systole} be detected in this ECG? Is there any sign of ${extra_systole} in this ECG record? | Can ${extra_systole} be observed from this ECG result? Does this ECG suggest the existence of ${extra_systole}? Does this ECG reveal any evidence of ${extra_systole}? |
| 37 | Can you identify whether the ECG shows ${extra_systole1} or ${extra_systole2}? Could you determine whether the ECG reveals ${extra_systole1} or ${extra_systole2}? Can you differentiate if the ECG displays ${extra_systole1} or ${extra_systole2}? Which extra systoles are visible on this ECG, ${extra_systole1} or ${extra_systole2}? Could you specify if the ECG presents ${extra_systole1} or ${extra_systole2}? Which type of extra systoles does this ECG indicate, ${extra_systole1} or ${extra_systole2}? What type of extra systoles are identifiable on this ECG, specifically ${extra_systole1} or ${extra_systole2}? | What type of extra systoles is evident on this ECG, ${extra_systole1} or ${extra_systole2}? Which specific type of extra systoles is exhibited on this ECG, ${extra_systole1} or ${extra_systole2}? Can you determine the type of extra systoles displayed on this ECG, specifically ${extra_systole1} or ${extra_systole2}? |
| 38 | What is the specific extra systole pattern displayed on this ECG? Could you identify the extra systole type from this ECG reading? What type of extra systole is observed in this ECG? Which type of extra systole is evident from this ECG? What extra systole is present in this ECG recording? What is the particular type of irregular heartbeat shown in this ECG reading? What extra systole is indicated on this ECG? | What kind of extra systole is visible in this ECG? Could you specify the extra heartbeat pattern shown in this ECG? What type of extra heartbeat is evident in this ECG tracing? |
| 39 | Does the ${numeric_feature} of this ECG fall ${numeric_range}? Does the ${numeric_feature} displayed on this ECG fall ${numeric_range}? Does the ${numeric_feature} shown on this ECG fall ${numeric_range}? Is the ${numeric_feature} seen on this ECG ${numeric_range}? Is the ${numeric_feature} in this ECG ${numeric_range}? Is the ${numeric_feature} presented in this ECG ${numeric_range}? Does the ${numeric_feature} of this ECG lie ${numeric_range}? | Is the ${numeric_feature} exhibited on this ECG ${numeric_range}? Does the ${numeric_feature} in this ECG fall ${numeric_range}? Does the ${numeric_feature} of this ECG lie ${numeric_range}? |
| 40 | Within which numeric range does the ${numeric_feature} of this ECG fall, ${numeric_range1} or ${numeric_range2}? Can you tell me if the ${numeric_feature} of this ECG falls ${numeric_range1} or ${numeric_range2}? Where does the ${numeric_feature} of this ECG fall, ${numeric_range1} or ${numeric_range2}? Is the ${numeric_feature} of this ECG situated ${numeric_range1} or ${numeric_range2}? What is the range in which the ${numeric_feature} of this ECG is situated, ${numeric_range1} or ${numeric_range2}? Which one is the ${numeric_feature} on the ECG be considered, ${numeric_range1} or ${numeric_range2}? In what range does the ECG's ${numeric_feature} fall, ${numeric_range1} or ${numeric_range2}? | In which range does the ${numeric_feature} of this ECG lie, ${numeric_range1} or ${numeric_range2}? Where is the ${numeric_feature} of this ECG situated, ${numeric_range1} or ${numeric_range2}? Can you specify whether the ${numeric_feature} of this ECG falls ${numeric_range1} or ${numeric_range2}? |
| 41 | Within what range does the ECG's ${numeric_feature} occur? What is the range that the ${numeric_feature} on this ECG falls into? At what range is the ${numeric_feature} of this ECG located? What is the range of the ${numeric_feature} seen in this ECG? What is the ${numeric_feature} range detected in this ECG? What range encompasses the ${numeric_feature} shown on this ECG? Within what range does the ${numeric_feature} of this ECG lie? | What is the ${numeric_feature} range observed on this ECG? What range does the ECG's ${numeric_feature} fall into? In what interval range does the ${numeric_feature} of this ECG exist? |
| 42 | Which numerical values of this ECG fall ${numeric_range}? What are the numeric attributes of this ECG that are ${numeric_range}? What are the numerical measurements of this ECG that fall ${numeric_range}? What ECG values are present ${numeric_range}? Which ECG measurements fall ${numeric_range}? Which numerical features of this ECG are found ${numeric_range}? What are the numerical attributes of this ECG that lie ${numeric_range}? | Which numeric features of this ECG are ${numeric_range}? What numerical values on this ECG fall ${numeric_range}? Which ECG measurements have values ${numeric_range}? |

Continued on next page.

| ID | Train & validation paraphrases | Test paraphrases |
|---|---|---|
| 43 | Compared to the previous tracing, has ${scp_code} been resolved in the recent tracing? 
 When comparing the recent tracing to the previous one, has the ${scp_code} been resolved? 
 Has ${scp_code} been resolved in the recent tracing compared to the previous one? 
 Has ${scp_code} been eliminated in the recent tracing in comparison to the previous one? 
 Has ${scp_code} been removed from the recent tracing in contrast to the previous one? 
 Does the recent tracing no longer indicate ${scp_code} when compared to the previous one? 
 Does the recent tracing show no evidence of ${scp_code} when compared to the previous tracing? | Has ${scp_code} been eliminated in the recent tracing in comparison to the previous one? 
 Is ${scp_code} no longer visible in the recent tracing in comparison to the previous one? 
 Does the recent tracing show the absence of ${scp_code} compared to the previous one? |
| 44 | Has ${scp_code} been identified in the recent tracing, as opposed to the previous one? 
 Has the recent tracing revealed any new instances of ${scp_code} in comparison to the previous tracing? 
 Has ${scp_code} been found in the recent tracing, which was not identified in the previous tracing? 
 Has the recent tracing detected any new cases of ${scp_code}, as compared to the previous one? 
 Are there any instances of ${scp_code} that have been newly detected in the recent tracing, as opposed to the previous tracing? 
 Has any new presence of ${scp_code} been observed in the recent tracing, as opposed to the previous one? 
 Are there any new instances of ${scp_code} identified in the recent tracing, as compared to the previous one? | Are there any new occurrences of ${scp_code} in the recent tracing compared to the previous one? 
 Does the recent tracing show any newly detected ${scp_code}, unlike the previous tracing? 
 Has there been any newly detected presence of ${scp_code} in the recent tracing, which was not found in the previous one? |
| 45 | Is ${scp_code} still included in the recent tracing as compared to the previous one? 
 Has ${scp_code} been retained in the recent tracing as compared to the previous tracing? 
 Compared to the previous tracing, does ${scp_code} still remain in the recent tracing? 
 Is ${scp_code} still accounted for in the recent tracing, in comparison to the previous one? 
 Are ${scp_code} still being tracked in the recent tracing, as compared to the previous tracing? 
 Does ${scp_code} continue to be part of the recent tracing as compared to the previous one? 
 Are ${scp_code} still detected in the recent tracing as compared to the previous one? | Does the recent tracing still contain ${scp_code} as compared to the previous one? 
 Are ${scp_code} still present in the recent tracing, as compared to the previous tracing? 
 Does the recent tracing still involve ${scp_code} as compared to the previous one? |
| 46 | Compared to the previous tracing, is ${scp_code} still not found in the recent tracing? 
 When compared to the previous tracing, is ${scp_code} still not located in the recent tracing? 
 Is ${scp_code} still absent in the recent tracing when compared to the previous one? 
 When compared to the previous tracing, is ${scp_code} still undiscovered in the recent tracing? 
 Has ${scp_code} been undiscovered in the recent tracing when compared to the previous tracing? 
 Is ${scp_code} still unlocated in the recent tracing when compared to the previous tracing? 
 In the recent tracing, is ${scp_code} still not present when compared to the previous tracing? | When compared to the previous tracing, is ${scp_code} still not discovered in the recent tracing? 
 Is ${scp_code} still not detected in the recent tracing when compared to the previous one? 
 Has ${scp_code} still not been identified in the recent tracing when compared to the previous tracing? |
| 47 | What symptoms have been resolved in the recent tracing as compared to the previous one? 
 Which symptoms have been removed during the recent tracing compared to the previous one? 
 What symptoms have been eliminated in the recent tracing that were present in the previous one? 
 In comparison to the previous tracing, what symptoms have been resolved in the recent tracing? 
 What are the symptoms that were present in the previous tracing that have been resolved in the previous one? 
 What symptoms were present in the previous tracing that are no longer present in the recent one? 
 Which symptoms have been cured during the recent tracing in contrast to the previous one? | Which symptoms have been alleviated in the recent tracing as opposed to the previous one? 
 What are the resolved symptoms during the recent tracing in contrast to the previous one? 
 What symptoms have disappeared in the recent tracing as compared to the previous one? |
| 48 | What symptoms are newly detected in the recent tracing as compared to the previous one? 
 In contrast to the previous tracing, what symptoms have been newly detected in the recent one? 
 What new symptoms have been detected in the recent tracing in comparison with the previous one? 
 What are the newly identified symptoms in the recent tracing compared to the previous one? 
 What are the newly discovered symptoms in the recent tracing as compared to the previous one? 
 What are the symptoms that are newly discovered in the recent tracing that were not detected in the previous one? 
 What new symptoms have emerged in the recent tracing that were not present in the previous one? | In the recent tracing, what new symptoms have been identified in contrast to the previous tracing? 
 What are the new symptoms detected in the recent tracing in comparison to the previous one? 
 As compared to the previous tracing, what new symptoms have been detected in the recent one? |

Table 20: Full list of paraphrases for each question template (7/10).

| ID | Train & validation paraphrases | Test paraphrases |
|---|---|---|
| 49 | What symptoms still remain in the recent tracing as compared to the previous one?
Which symptoms remain present in the latest tracing in contrast to the previous one?
What symptoms are still observable in the recent tracing, as compared to the earlier one?
In the recent tracing, what symptoms are still present in contrast to the preceding one?
Which symptoms are still evident in the recent tracing, in comparison to the preceding one?
What symptoms continue to be present in the recent tracing, relative to the earlier one?
In comparison to the previous tracing, which symptoms still remain in the recent tracing? | In the recent tracing, what symptoms persist in comparison to the previous tracing?
What symptoms continue to exist in the recent tracing, relative to the prior one?
What symptoms remain in the recent tracing as compared to the prior one? |
| 50 | Has the recent tracing's ${numeric_feature} become normal when compared to the previous tracing?
When compared to the previous tracing, has the ${numeric_feature} of the recent tracing returned to its normal state?
In contrast to the previous tracing, is the ${numeric_feature} of the recent tracing now normal?
Is the ${numeric_feature} of the recent tracing back to normal when compared to the previous tracing?
Compared to the previous tracing, has the ${numeric_feature} in the recent tracing become typical?
Has the ${numeric_feature} of the recent tracing normalized compared to the previous tracing?
Is the ${numeric_feature} of the recent tracing now typical, in comparison to the previous tracing? | Has the ${numeric_feature} of the recent tracing returned to normal after being compared to the previous tracing?
Is the ${numeric_feature} of the recent tracing now normal, as opposed to the previous tracing?
Has the ${numeric_feature} of the recent tracing returned to typical levels compared to the previous tracing? |
| 51 | Is the ${numeric_feature} in the recent tracing considered abnormal when compared to the previous tracing?
Is the ${numeric_feature} of the recent tracing deemed abnormal, while it was not in the previous tracing?
Does the recent tracing indicate an abnormal ${numeric_feature}, unlike the previous tracing?
Has an abnormal ${numeric_feature} been detected in the recent tracing but not in the previous tracing?
Does the recent tracing display an abnormal ${numeric_feature} when compared to the previous tracing?
In comparison to the previous tracing, is the ${numeric_feature} in the recent tracing considered abnormal?
Does the recent tracing exhibit an abnormal ${numeric_feature} in contrast to the previous tracing? | Has the ${numeric_feature} in the recent tracing been identified as abnormal, whereas it was not in the previous tracing?
Is the ${numeric_feature} in the recent tracing abnormal compared to the previous tracing?
Is the ${numeric_feature} in the recent tracing regarded as abnormal, unlike the previous tracing? |
| 52 | Has the value of ${numeric_feature} continued to be abnormal in the current tracing, when compared to the previous one?
Does the recent tracing still exhibit an unusual value for ${numeric_feature} in comparison to the previous tracing?
Is ${numeric_feature} still displaying an unusual value in the recent tracing compared to the previous one?
Has the abnormality of ${numeric_feature} persisted in the current tracing as compared to the previous tracing?
Is the current tracing still showing an abnormal value for ${numeric_feature} when compared to the previous tracing?
Has the abnormal value of ${numeric_feature} persisted in the current tracing, in comparison to the previous one?
Does the recent tracing still indicate an abnormal value for ${numeric_feature} in comparison to the previous tracing? | Is the ${numeric_feature} still showing an abnormal value in the recent tracing, when compared to the previous one?
Is ${numeric_feature} still displaying an abnormal value in the current tracing, in comparison to the previous one?
Does the recent tracing show an abnormal value for ${numeric_feature} similar to the previous tracing? |
| 53 | Compared to the previous tracing, is the ${numeric_feature} still considered normal in the recent tracing?
Does the recent tracing confirm the normalcy of the ${numeric_feature} as shown in the previous tracing?
Is the ${numeric_feature} still in the normal range on the recent tracing, as it was in the previous one?
Does the ${numeric_feature} value remain within normal limits on the recent tracing, similar to the previous tracing?
Is the ${numeric_feature} still considered to be within the normal limits on the recent tracing, as it was on the previous tracing?
Is the ${numeric_feature} value on the recent tracing consistent with the previous tracing, indicating normality? | Is the ${numeric_feature} still within the normal range on the recent tracing when compared to the previous tracing?
Has the ${numeric_feature} remained normal in the recent tracing compared to the previous tracing?
Is the ${numeric_feature} still considered normal in the latest tracing when compared to the previous tracing? |
| 54 | Which numeric attributes in the recent tracing have returned to the normal range from the previous one?
What numeric features in the recent trace are now considered normal, relative to the previous one?
What numeric factors in the recent tracing have now fallen within the normal range when compared to the prior one?
What numerical elements of the latest tracing have become normal compared to the previous one?
Which numeric values in the recent tracing are now in the standard range, in contrast to the prior one?
Which numeric attributes in the recent tracing are now considered typical when compared to the previous one?
What numeric indicators in the recent trace have returned to the standard range compared to the previous one? | What numeric features in the recent trace have returned to the normal range compared to the previous one?
What numerical values in the recent tracing have returned to the standard range from the previous one?
In comparison to the previous one, which numeric features in the recent tracing have become normal? |

Table 20: Full list of paraphrases for each question template (8/10).

| ID | Train & validation paraphrases | Test paraphrases |
|---|---|---|
| 55 | What numeric attributes in the latest tracing are currently identified as abnormal when compared to the previous one? Which numerical features of the latest tracing are deemed abnormal in contrast to the previous one? What numeric values in the recent tracing are now considered abnormal compared to the previous one? Which numerical features of the recent tracing are now abnormal values when compared to the previous one? What numerical features of the recent tracing are currently abnormal compared to the previous one? Which numeric values in the latest tracing are now considered abnormal in comparison to the previous one? Which numerical features of the latest tracing are regarded as unusual values compared to the previous one? | Which numerical features in the recent tracing are considered abnormal values in comparison to the previous one? What are the newly abnormal numeric features of the recent tracing when compared to the previous one? What numeric attributes of the recent tracing are deemed abnormal values in contrast to the previous one? |
| 56 | What numerical attributes of the recent tracing remain atypical when compared to the previous one? Which numerical values of the recent tracing remain abnormal relative to the previous one? What numeric indicators in the recent tracing still stand out as abnormal compared to the previous one? What kinds of numeric features in the recent tracing are still considered abnormal when compared to the previous one? Which specific numerical values in the recent tracing are still considered abnormal when compared to the previous one? Which numerical factors of the recent tracing remain abnormal relative to the previous one? What are the abnormal numerical measurements that still exist in the recent tracing compared to the previous one? | What numeric features of the recent tracing are still abnormal compared to the previous one? What numerical attributes of the recent tracing remain abnormal as compared to the previous tracing? What numerical values of the recent tracing are still atypical compared to the previous tracing? |
| 57 | Has ${scp_code} been eliminated in the second ECG as compared to the first ECG? Is ${scp_code} absent in the second ECG as opposed to the first ECG? Does the second ECG demonstrate the removal of ${scp_code} in contrast to the first ECG? Has the second ECG shown the removal of ${scp_code} in relation to the first ECG? Does the second ECG show the elimination of ${scp_code}, in relation to the first ECG? Has the second ECG shown that there is no ${scp_code} in comparison to the first ECG? Has ${scp_code} been ruled out in the second ECG, compared to the first ECG? | Has the second ECG shown the absence of ${scp_code} in comparison to the first one? Does the second ECG show removal of ${scp_code} in comparison to the first ECG? Has ${scp_code} been removed in the second ECG, when compared to the first ECG? |
| 58 | Is ${scp_code} a newly observed finding in the second ECG when compared to the first one? Is ${scp_code} a newly discovered feature in the second ECG compared to the first one? Has the second ECG indicated the presence of ${scp_code} that was not previously detected in the first ECG? Has ${scp_code} been newly identified in the second ECG when compared to the initial ECG? Does the second ECG reveal the presence of ${scp_code} that was not previously detected in the first ECG? Has ${scp_code} been newly detected in the second ECG in comparison to the initial ECG? Does the second ECG show the presence of ${scp_code} that was not previously detected in the first ECG? | Have we discovered an occurrence of ${scp_code} on the second ECG that was not apparent on the first? Has ${scp_code} been newly diagnoses on the second ECG compared to the first? Is there any evidence of a newly detected ${scp_code} in the second ECG that was not present in the first? |
| 59 | Is ${scp_code} still present in the second ECG as compared to the first one? Is ${scp_code} still detectable in the second ECG in comparison to the first one? Does the second ECG still show the presence of ${scp_code} like the first ECG? Is ${scp_code} still observable in the second ECG in comparison to the first ECG? Has the presence of ${scp_code} remained consistent in the second ECG when compared to the first ECG? Does ${scp_code} still show up in the second ECG like it did in the first ECG? Has the presence of ${scp_code} been sustained in the second ECG when compared to the first ECG? | Does the second ECG still indicate the presence of ${scp_code} as compared to the first ECG? Has the presence of ${scp_code} persisted in the second ECG as compared to the first ECG? Does the second ECG still reflect the presence of ${scp_code} that was observed in the first ECG? |
| 60 | Has the absence of ${scp_code} persisted in the second ECG when compared to the first ECG? Does the second ECG still indicate the absence of ${scp_code} as compared to the first ECG? Is ${scp_code} still not observable in the second ECG when compared to the first ECG? Does the second ECG still exhibit the absence of ${scp_code} in comparison to the first ECG? Is ${scp_code} still not apparent in the second ECG as compared to the first ECG? Does the second ECG still demonstrate the absence of ${scp_code} in relation to the first ECG? Is ${scp_code} still not detected in the second ECG when compared to the first ECG? | Does the second ECG still show the absence of ${scp_code} when compared to the first ECG? Does the second ECG continue to show the absence of ${scp_code} in comparison to the first ECG? Is ${scp_code} still not present in the second ECG when compared to the first ECG? |
| 61 | Which diagnoses are no longer evident in the second ECG as compared to the first ECG? Which diagnoses are no longer visible in the second ECG compared to the first ECG? What symptoms are present in the first ECG but absent in the second ECG? What symptoms have been resolved in the second ECG as compared to the first ECG? Can you identify any diagnoses that were resolved in the second ECG as compared to the first ECG? What are the symptoms that are present in the first ECG but absent in the second ECG? Which symptoms have been alleviated in the second ECG that were present in the first ECG? | Can you identify which symptoms were present in the first ECG but absent in the second ECG? What diagnoses were ruled out in the second ECG compared to the first ECG? What diagnoses were removed or resolved in the second ECG as opposed to the first ECG? |

Table 20: Full list of paraphrases for each question template (9/10).

| ID | Train & validation paraphrases | Test paraphrases |
|---|---|---|
| 62 | What new symptoms are visible in the second ECG that were not present in the first one?
What symptoms have been detected in the second ECG that were not present in the first?
What symptoms have emerged on the second ECG as opposed to the first?
Which symptoms have been newly identified in the second ECG in comparison to the first?
Which new symptoms have been detected in the second ECG that were not present in the first?
What symptoms are new on the second ECG that were not present in the first ECG?
Can you describe any new symptoms that have appeared on the second ECG but not on the first? | Can you identify any new symptoms in the second ECG compared to the first one?
What new symptoms can be seen on the second ECG compared to the first one?
What are the newly discovered symptoms on the second ECG when compared to the first? |
| 63 | What are the symptoms that are still evident in the second ECG when compared to the first ECG?
Which symptoms remain unchanged in the second ECG as compared to the first ECG?
Which symptoms have persisted in the second ECG as compared to the first ECG?
Which symptoms persist in the second ECG compared to the first ECG?
What are the symptoms that are still present in the second ECG that were also present in the first ECG?
Which symptoms continue to exist in the second ECG when compared to the first ECG?
Which symptoms are still evident in the second ECG that were present in the first ECG? | What are the symptoms that have not resolved in the second ECG compared to the first ECG?
Which symptoms are still detectable in the second ECG as compared to the first ECG?
Which symptoms remain unchanged between the first and second ECG? |
| 64 | Has the ${numeric_feature} in the second ECG returned to a normal state compared to the first ECG?
Is the ${numeric_feature} value on the second ECG now within the normal range when compared to the first ECG?
Has the ${numeric_feature} returned to a normal range on the second ECG compared to the first ECG?
Is the ${numeric_feature} measurement on the second ECG now within the normal range compared to the first ECG?
Does the ${numeric_feature} value on the second ECG return to a normal state compared to the first ECG?
Is the ${numeric_feature} reading on the second ECG indicative of a normal state as opposed to the first ECG?
Is the ${numeric_feature} on the second ECG within the normal range as opposed to the first ECG? | Is the ${numeric_feature} reading on the second ECG now considered normal compared to the first ECG?
Has the ${numeric_feature} reached a normal level on the second ECG as opposed to the first ECG?
Has the ${numeric_feature} returned to normal levels on the second ECG in comparison to the first ECG? |
| 65 | Has the ${numeric_feature} value on the second ECG shifted from the normal range in comparison to the first ECG?
Is the ${numeric_feature} on the second ECG now considered abnormal when compared to the first ECG?
Does the ${numeric_feature} value on the second ECG fall outside the normal range as opposed to the first ECG?
Is the ${numeric_feature} on the second ECG indicating abnormality as opposed to the first ECG result?
Does the ${numeric_feature} value on the second ECG indicate abnormality compared to the first ECG result?
Has the ${numeric_feature} result on the second ECG moved away from the normal range relative to the first ECG result?
Is the ${numeric_feature} result on the second ECG now indicative of abnormality when compared to the first ECG reading? | Is the ${numeric_feature} value on the second ECG indicating abnormality as opposed to the first ECG?
Is the ${numeric_feature} result on the second ECG indicating deviation from normal as opposed to the first ECG?
Has the ${numeric_feature} on the second ECG moved out of the normal range compared to the first ECG? |
| 66 | Does the ${numeric_feature} on the second ECG still demonstrate an abnormal value compared to the first one?
Is the ${numeric_feature} still abnormal on the second ECG as compared to the first one?
Does the second ECG exhibit an abnormal value for the ${numeric_feature}, similar to the first ECG?
Is the ${numeric_feature} in the second ECG still outside the normal range seen in the first ECG?
Does the second ECG still show the same abnormality in the ${numeric_feature} as the first ECG?
Is there still an abnormality in the second ECG with respect to the ${numeric_feature}?
Does the ${numeric_feature} on the second ECG still indicate an abnormality similar to the first ECG? | Does the ${numeric_feature} of the second ECG still indicate an abnormal value compared to the first one?
Is there still an abnormality in the ${numeric_feature} on the second ECG when compared to the first one?
Has the abnormal value in the ${numeric_feature} persisted in the second ECG? |
| 67 | Does the second ECG still show a normal reading for the ${numeric_feature} compared to the first ECG?
Does the second ECG continue to show a normal ${numeric_feature} reading when compared to the first ECG?
Is the ${numeric_feature} reading still normal on the second ECG as compared to the first ECG?
Does the second ECG still demonstrate a normal ${numeric_feature} reading in comparison to the first ECG?
Is the ${numeric_feature} reading still normal on the second ECG in comparison to the first ECG?
Is the ${numeric_feature} reading on the second ECG still normal as compared to the first ECG?
Does the second ECG still indicate a normal reading for the ${numeric_feature} as compared to the first ECG? | Is the ${numeric_feature} still in the normal range on the second ECG as compared to the first ECG?
Is the ${numeric_feature} still within the normal range on the second ECG when compared with the first ECG?
Does the second ECG still present a normal ${numeric_feature} reading as compared to the first ECG? |
| 68 | Which numerical features in the second ECG have returned to a normal state in comparison to the first ECG?
What are the numeric features that are now considered normal in the second ECG, as opposed to the first one?
What numerical values in the second ECG have returned to a standard state in comparison to the first ECG?
Which numeric features of the second ECG have now become typical in contrast to the first ECG?
What are the numerical features of the second ECG that are now within a normal range in contrast to the first ECG?
What numeric measurements of the second ECG have changed from abnormal to normal compared to the first ECG?
Which numeric measurements in the second ECG are now considered stable compared to the first ECG? | What are the numerical features of the second ECG that have reverted to normal compared to the first ECG?
Which numeric features in the second ECG are now within the normal range in contrast to the first ECG?
Which numeric measurements in the second ECG have become normal as compared to the first ECG? |

Continued on next page.

Table 20: Full list of paraphrases for each question template (10/10).

| ID | Train & validation paraphrases | Test paraphrases |
|---|---|---|
| 69 | Which numerical features of the second ECG are considered unusual in contrast to the first ECG? Which numeric values in the second ECG are newly considered abnormal when compared to the first ECG? Which numerical attributes of the second ECG are now abnormal values when compared to the first ECG? What numeric factors in the second ECG are now considered irregular values compared to the first ECG? Which numeric measurements in the second ECG are newly considered abnormal values compared to the first ECG? What are the numeric values of the second ECG that are regarded as abnormal in contrast to the first ECG? What are the numerical measurements in the second ECG that deviate from the normal range established by the first ECG? | Which numeric features in the second ECG are now considered abnormal in relation to the first ECG? Which numerical values of the second ECG are considered irregular values as opposed to the first ECG? What are the numeric measurements in the second ECG that are newly considered abnormal compared to the first ECG? |
| 70 | Which numerical features of the second ECG remain irregular compared to the first ECG? Which numerical features of the second ECG are not still normal when compared to the first ECG? Which numeric features of the second ECG are still irregular compared to the first ECG? What are the numeric measurements of the second ECG that remain atypical compared to the first ECG? Which numeric attributes of the second ECG are still irregular relative to the first ECG? What are the numerical measurements of the second ECG that are still not normal compared to the first ECG? What numeric features of the second ECG are still abnormal as compared to the first ECG? | What are the numerical attributes of the second ECG that are still abnormal relative to the first ECG? Which numerical factors of the second ECG continue to be abnormal as compared to the first ECG? What are the numeric factors of the second ECG that are still abnormal as compared to the first ECG? |