# OpenReview forum: "ECG-QA: A Comprehensive Question Answering Dataset Combined With Electrocardiogram"
_NeurIPS.cc/2023/Track/Datasets_and_Benchmarks — NeurIPS 2023 Datasets and Benchmarks Poster_

### Official Review · Reviewer_qtCQ · 2023-07-10
**The paper is solid, but lacks some impact.**

**Rating:** 7
**Confidence:** 3
**Correctness:** The dataset construction process appe…
**Clarity:** The paper is well written and the pur…

**Strengths:**

* A QA dataset with a variety of questions focused on ECG is rare and has potential clinical significance.
* As a data set based on expert review, it is expected to be of high quality.
* The chapter on "question templates" is particularly well written and would be useful for researchers wishing to undertake similar initiatives.
* Extensive experimental results based on several state-of-the-art models are presented.
* The usefulness of the proposed method is also demonstrated in comparison with LLM.

**Additional Feedback:**

I have no additional feedback.

**Documentation:**

The content of the dataset is thoroughly described.

**Ethics:**

As it is built upon existing datasets, there seem to be no ethical issues.

**Limitations:**

* It seemed that many of the questions were designed to be answered with a simple "Yes" or "No" or with a class classification. If this is the case, what is the qualitative difference between these easy-to-answer questions and simply solving a classification problem?

**Opportunities For Improvement:**

* It was unclear what clinical criteria the term "stage of myocardial infarction" refers to.
* Regarding extra systole, it was difficult to understand from a clinical perspective what origins exist other than ventricular and supraventricular.

**Relation To Prior Work:**

The relation to prior research is also detailed.

**Summary And Contributions:**

The paper constructed a dataset for question answering (QA) in electrocardiograms (ECGs). The paper has constructed a QA dataset that covers a wide range of clinical significance using a total of 70 question templates, while receiving reviews from specialists in ECGs. In addition, it was designed to be more clinically useful by including data based on the difference between two ECGs.

---

> ### Author Response · Authors · 2023-08-19
> **Response to the reviewer qtCQ**
>
> We appreciate your time and effort for reviewing our manuscript. We address your comments below.
>
> **Q: It was unclear what clinical criteria the term "stage of myocardial infarction" refers to.**
>
> In the original ECG dataset, PTB-XL, the stages of infarction for samples belonging to Myocardial Infarction (MI) were labeled by extracting keywords from their respective reports.
> Specifically, as outlined in Table (A) (see below), when a report contained specific measurement keywords, the corresponding stage was assigned as the label.
> To simplify these stages, in ECG-QA, intermediate stages such as I-II and II-III were considered as I and II, respectively.
>
> |Table (A)|Keyword|
> |-|-|
> |Stage I|acut, early|
> |Stage I-II|acut/subacut, ablaufend|
> |Stage II|recent, subacut, bereits abgelaufen|
> |Stage II-III|subacut/chronisch|
> |Stage III|old, abgelaufen, chronisch|
> |Unknown|uncertain, unknown, unbekannt|
>
>
> **Q: Regarding extra systole, it was difficult to understand from a clinical perspective what origins other than ventricular and supraventricular.**
>
> In the PTB-XL dataset, extra beats were categorized into three categories: ES, VES, and SVES.
> However, the original dataset paper does not mention the exact criteria for defining these three classes, especially for the ES category where the origin is not specified unlike VES (ventricular ES) or SVES (supraventricular ES).
> Based on the medical expert analysis of the ECG samples that belong to the ES category, it seems ES category is used for labeling ECG samples that are difficult to label as either of ventricular origin or supraventricular origin, or do not distinctly belong to either (e.g., junctional beats).
> Considering that it is often challenging to differentiate the origin of extrasystoles by observing only a 10-second segment of the 12-lead ECG, we speculate that the ES class is used for such cases.
>
> **Q: It seemed that many of the questions were designed to be answered with a simple "Yes" or "No" with a class classification. If this is the case, what is the qualitative difference between these easy-to-answer questions and simply solving a classification problem?**
>
> Thank you for pointing out a very important aspect of our dataset.
> We would like to address this question in two aspects: Single type questions and Comparison type questions.
> * As you pointed out, if only Single-Verify questions are considered, it could be seen as a simple binary classification problem. However, the proportion of Single-Verify questions out of the total Single questions is only approximately 37%. The remaining 63% are composed of Choose questions, where the selection is made from the given two options (A, B), which yields four possible answers (A, B, Both, or Neither), and Query questions, which require retrieving all the associated attributes. As mentioned in Section 4.4, Verify questions can be considered as samples that test the model's basic perception of ECG. In fact, even in the case of Visual Question Answering (VQA) dataset [\[1\]](https://arxiv.org/abs/1505.00468), Yes/No answers account for around 38% of the total answers (See Appendix VIII in that paper).
> * When looking at Comparison questions, which involve comparing two ECGs, the ratio of Verify questions is about 45%, which is relatively higher compared to the case of Single questions. However, in this context, Verify questions are much more complex than simple binary classification problems because the model needs to determine whether to answer based on the differences or commonalities between the two analyzed results, as indicated by the question. For example, to answer the question "Compared to the previous tracing, does atrial fibrillation still remain in the recent tracing?", the model must first detect whether atrial fibrillation is present in both the previous and the recent tracings, and then integrate these results with the posed question to provide the correct answer—in this case, determining whether atrial fibrillation is present in both ECGs, which requires the model to learn logical operations such as AND. This is evidently a more challenging task than a simple binary classification problem, and it ultimately highlights why the ECG-QA dataset can be referred to as a comprehensive dataset.
>
> We hope that our response sufficiently addresses your comments.
>
> **Reference**
>
> [1]: Antol, Stanislaw, et al. "Vqa: Visual question answering." Proceedings of the IEEE international conference on computer vision. 2015.

---

> > ### Comment · Reviewer_qtCQ · 2023-08-28
> > **Response to authors**
> >
> > Thank you very much for answering my questions so detailed and courteously. In particular, I understood well that building a VQA dataset for the time-series electrocardiogram could be useful for developing a more comprehensive and practical AI system.

---

### Official Review · Reviewer_4ESf · 2023-07-21
**Review for ECG-QA: A Comprehensive Question Answering Dataset Combined With Electrocardiogram**

**Rating:** 7
**Confidence:** 3
**Clarity:** The paper is well-written.

**Strengths:**

1. The authors provide one of the first QA datasets for ECG analysis, and the dataset is of a reasonable size and richness.
2. Having QA questions where 2 ECGs are compared is crucially important and in the reviewer's opinion a major highlight of the provided dataset.
3. The dataset repository is well-documented and there are clear instructions on how to reproduce and run experiments using the generated dataset.
4. The experiments are well-designed and executed, and the writing is generally clear.

**Additional Feedback:**

None

**Correctness:**

The dataset is formulated in a sound way as far as the reviewer can check. The benchmark evaluation methods appear to follow standard practices.

**Documentation:**

Yes, the extensive documentation available is a highlight of this paper.

**Ethics:**

No. It is based on an open-source dataset.

**Limitations:**

The authors did not include a limitations section for their dataset. The authors should detail the limitations of their extraction of query questions and answering from the original dataset (if any), the limitations regarding human labels (which the article touched upon in Section 4.2), the limitations regarding the SCP-ECG v0.4 standard, as it has now been superseded, and any other limitations the authors have realized in the construction of the dataset and benchmarking.

**Opportunities For Improvement:**

1. Is the regular expression parser for extracting the form-related symptoms from the ECG reports perfectly accurate? If not, what procedures did the authors use to improve/check its accuracy?
2. The authors have not adequately addressed the limitations of this dataset and their benchmark (discussed below).
3. It could be interesting to evaluate LLMs in a free-text manner, especially for the query questions, to see not only if they get the correct answer, but if the explanation makes sense to an expert cardiologist.

**Relation To Prior Work:**

Relation to prior work is clearly discussed.

**Summary And Contributions:**

In this paper, authors present the first QA dataset specifically designed for ECG analysis, named ECG-QA. The ECG-QA dataset comprises 70 question templates covering a broad spectrum of clinically relevant ECG topics. The dataset includes a range of ECG interpretation questions, including those that necessitate comparative analysis between two different ECGs. The authors conducted numerous experiments with machine learning and LLM models.

---

> ### Author Response · Authors · 2023-08-19
> **Response to the reviewer 4ESf**
>
> We appreciate your time and effort for reviewing our manuscript. We address your comments below.
>
> **Q: Is the regular expression parser for extracting the form-related symptoms from the ECG reports perfectly accurate? If not, what procedures did the authors use to improve/check its accuracy?**
>
> For your reference, we added in Supplementary B.1 of the revised manuscript the regular expression parser that was used to extract lead positions of form-related symptoms.
> After developing the parser by carefully examining ECG reports, we consulted with medical experts to check the validity of the parser.
> During the validation, however, we could not measure the full accuracy for all samples due to the absence of ground-truth lead positions for each symptom.
> Therefore, we manually labeled 20 random samples for each symptom and then compared these labels with the parser's output to determine exact-match accuracy.
> The results are presented in Table (A) (see below).
> As a baseline for comparing the parser's performance, we utilized ChatGPT (gpt-3.5-turbo), using the following prompt:
>
> > *What leads are showing ${scp_code} based on this report?*
> *The possible options are [lead I, lead II, lead III, lead aVR, lead aVL, lead aVF, lead V1, lead V2, lead V3, lead V4, lead V5, lead V6, none]*
> *Only output the answer without any explanation.*
> *report:*
> *${report}*
>
> |Table (A)|Our parser|gpt-3.5-turbo|
> |:-:|:-:|:-:|
> |Num. of exactly matched samples / Total samples|240 / 260|126 / 260|
> |Percentage (exact-match accuracy)|92.31%|48.46%|
>
> As shown in Table (A), the regular expression parser demonstrates significantly superior exact-match accuracy compared to gpt-3.5-turbo, achieving a performance level of approximately 92.3%.
> Throughout this analysis, we have noticed that the inaccurately parsed samples are primarily caused from the variances in vocabulary that specify the lead positions.
> For example, a string of a single letter such as "r", "l", or "f" should have been interpreted as "lead aVR", "lead aVL", or "lead aVF", respectively.
> In our next release, we plan to enhance the parser to incorporate these specific vocabularies, expecting improved accuracy beyond its current performance (92.31%).
>
> **Q: The authors have not adequately addressed the limitations of this dataset and their benchmark.**
>
> Due to the page limit, we initially included the limitations in the Conclusion section.
> However, in the revised version, as you suggested, we have added a dedicated limitation section in Section 6.
> This section includes both the original limitations and those you highlighted as follows:
> * Due to the limited number of ECGs in the original PTB-XL dataset, questions involving too rare symptoms (e.g., Wolf-Parkinson-White syndrome) could not be included. With the utilization of larger datasets (e.g., MIMIC-IV-ECG) in the future, it is expected that questions regarding very rare attributes could be incorporated.
> * As mentioned in Section 4.2, given the intricacies of the medical field, even medical experts cannot provide 100% accurate diagnoses for all questions. Thus, the upper-bound of the dataset itself is not expected to be 100%. To address this, we conducted experiments demonstrating the estimated upper-bound for each question type and attribute (see Section 4.2).
> * Despite SCP-ECG v3.0 being the latest version, the metadata of the original dataset, PTB-XL, follows the SCP-ECG v0.4 standard. Consequently, in ECG-QA, we were constrained to categorize various symptoms based on the SCP-ECG v0.4 standard. However, after investigating how the SCP codes in SCP-ECG v3.0 are categorized, we found that there is only a little difference between SCP-ECG v0.4 and v3.0 regarding the SCP codes used in PTB-XL. Among the SCP codes used in PTB-XL, only one SCP code (BIGU, bigeminal pattern - unknown origin, SV or Ventricular) has a different representation, which has changed to "SVBIG" (supraventricular bigeminy BIGU bigeminal pattern - unknown origin, SV or Ventricular) in SCP-ECG v3.0. The rest of the SCP codes have maintained their codes and definitions intact in SCP-ECG v3.0. Therefore, we believe that the impact of the differences between the two versions will not be significant in the ECG-QA dataset.

---

> > ### Author Response · Authors · 2023-08-19
> > **Response to the reviewer 4ESf (Continued)**
> >
> > **Q: It could be interesting to evaluate LLMs in a free-text manner, especially for the query questions, to see not only if they get the correct answer, but if the explanation makes sense to an expert cardiologist.**
> >
> > In our experimental setup, the LLM does not directly comprehend the ECG itself.
> > Instead, it uses results analyzed by the ECG classifier that are converted into a natural language format for QA.
> > Given this setup, evaluating outputs in a free-text format that includes explanations can be challenging.
> > One potential solution might be to analyze the answers generated by a multi-modal LLM that can process ECG signals along with natural language.
> > Given the emergence of multi-modal LLMs like Med-PaLM M [\[1\]](https://arxiv.org/abs/2307.14334) recently, which can simultaneously process images and text in the clinical domain, we also plan to explore using such models in our future work.
> >
> > We hope that our response sufficiently addresses your comments.
> >
> > **Reference**
> >
> > \[1\]: Tu, Tao, et al. "Towards Generalist Biomedical AI." arXiv preprint arXiv:2307.14334 (2023).

---

> > > ### Comment · Reviewer_4ESf · 2023-08-28
> > >
> > > Thank you for addressing my comments.

---

### Official Review · Reviewer_28Uj · 2023-07-28
**well-designed QA dataset on ECG**

**Rating:** 7
**Confidence:** 2
**Correctness:** Yes
**Clarity:** Yes

**Strengths:**

- The paper is well structured and the ideas are well presented.
- The Dataset Construction section is detailed and authors provide the question generation pipeline clearly.
- The paper provides replicable baselines. The experiments are supported by detailed Appendix (hyper-parameters, LLM prompting,...)
- Authors propose an interesting Upper bound experiments

**Additional Feedback:**

/

**Documentation:**

Yes

**Ethics:**

Yes

**Limitations:**

The limitations have been stated clearly by the authors:
- Due to the limited number of ECGs in the original PTB-XL dataset, our dataset was built using a relatively small number of ECGs (approximately 16k).
- While the paraphrases were manually curated, the initial candidates were automatically generated by ChatGPT, which might not be the best approach. Paraphrases could have been more diverse if medical practitioners were involved in manually generating them.

**Opportunities For Improvement:**

- It would be helpful for the reader that the authors provide interesting future research directions with the datasets.
- It would be helpful to have a lower bound experiment: training a model without the ECG, just the Q/A pairs. This experiments could detect important bias in the dataset.


**Relation To Prior Work:**

Yes

**Summary And Contributions:**

- The ECG-QA dataset is a diverse collection of questions focusing on ECG interpretation and analysis.
- This dataset introduces a novel concept by incorporating question answering into ECG analysis. This addition brings a new level of complexity as it extends beyond the standard scope of ECG analysis.
- The paper reports a benchmark for QA models, including large language models.
- The dataset addresses real-world needs of medical professionals and expands the potential applications of machine learning in ECG analysis.

---

> ### Author Response · Authors · 2023-08-19
> **Response to the reviewer 28Uj**
>
> We appreciate your time and effort for reviewing our manuscript. We address your comments below.
>
> **Q: It would be helpful for the reader that authors provide interesting future research directions with the datasets.**
>
> As you suggested, we have added the following future research directions in the Conclusion section (see Section 5) of the revised manuscript.
> There are a variety of future research directions with ECG-QA.
> One of the promising avenues is the exploration of multi-modal LLMs that can simultaneously process both ECG signals and natural language.
> While there is extensive work on LLMs that combine vision and language, there has been limited research on models that integrate signal processing with natural language.
> We believe our dataset can serve as an excellent testbed for such models.
>
> **Q: It would be helpful to have a lower bound experiment: training a model without the ECG, just the Q/A pairs. These experiments could detect important bias in the dataset.**
>
> The experimental results for models trained solely on QA pairs without ECG (Blind Transformer) and models trained solely on ECG without QA pairs (Deaf Transformer) are already presented in Table 2 to demonstrate the lower bounds.
> Both the Blind Transformer and Deaf Transformer exhibit significantly lower performance compared to the Fusion Transformer, which is the model trained on both ECG and QA pairs.
> Moreover, they even perform worse than the prior model, per Q-type majority, which always outputs only the most frequent answer for each question type.
> This clearly indicates that our dataset consists of questions that cannot be answered without considering both the question and ECG signal simultaneously.
>
> We hope that our response sufficiently addresses your comments.

---

> > ### Comment · Reviewer_28Uj · 2023-08-22
> > **Response to authors**
> >
> > Thank you for addressing my comments,
> >
> > Best,

---

### Decision · Program_Chairs · 2023-09-22

**Decision:**

Accept (Poster)

**Comment:**

The reviewers were overall positive and did not highlight any critical flaws in the dataset. On that basis I am recommending acceptance.